# Convergence Behavior of an Adversarial Weak Supervision Method

**Steven An**[1]

**Sanjoy Dasgupta**[1]

[1]Computer Science Department,
University of California, San Diego,
La Jolla, CA 92093, USA

## Abstract

Labeling data via rules-of-thumb and minimal label supervision is central to Weak Supervision, a paradigm subsuming subareas of machine learning such as crowdsourced learning and semi-supervised ensemble learning. By using this labeled data to train modern machine learning methods, the cost of acquiring large amounts of hand labeled data can be ameliorated. Approaches to combining the rules-of-thumb falls into two camps, reflecting different ideologies of statistical estimation. The most common approach, exemplified by the Dawid-Skene model, is based on probabilistic modeling. The other, developed in the work of Balsubramani-Freund and others, is adversarial and game-theoretic. We provide a variety of statistical results for the adversarial approach under log-loss: we characterize the form of the solution, relate it to logistic regression, demonstrate consistency, and give rates of convergence. On the other hand, we find that probabilistic approaches for the same model class can fail to be consistent. Experimental results are provided to corroborate the theoretical results.

## 1 INTRODUCTION

We consider a common setting found in Weak Supervision (WS): suppose we have a fixed set of data points $X = \{x_1, \ldots, x_n\}$ whose labels, in $\mathcal{Y} = \{1, 2, \ldots, k\}$, are not known. We are also given $p$ *rules-of-thumb* (sometimes called *labeling functions*) $h^{(1)}, \ldots, h^{(p)} : X \to \mathcal{Y} \cup \{?\}$, where "?" means "abstain". Given rough estimates of the accuracies of these rules, how can we use them to make inferences about the labels of $X$?

The WS umbrella contains work from several lines of machine learning including crowdsourced learning, semi-

supervised learning, and programmatic weak supervision. Methods range from unsupervised to semi-supervised.

In *crowdsourced learning*, several workers are asked to label $X$. They might abstain on some of the points, and thus each worker's labeling corresponds to a rule-of-thumb. These rules can be combined using a purely unsupervised process, or using a small amount of expertly-labeled data to help estimate the accuracy of each rule (person).

The field of *semi-supervised learning* has a long history of using simple human- or machine-generated rules like

document contains `goalie` $\implies$ label = `sports`.

In this example, the rule abstains on any document not containing the word `goalie` and is only somewhat accurate. The hope is for a collection of such rules, together with a little labeled data, to be turned into a good classifier. Early instances of this idea include work in information retrieval [Croft and Das, 1990], Yarowsky's method for word sense disambiguation [Yarowsky, 1995], and co-training [Blum and Mitchell, 1998]. More recently, Balsubramani and Freund [2015b] have suggested using highly accurate "specialists" that predict on small parts of input space, and then combining them.

*Programmatic weak supervision*, exemplified by the `snorkel` framework [Ratner et al., 2020], uses user-defined or automatically generated computer programs which serve as rules-of-thumb. The combination is usually unsupervised, but a small amount of labeled data can help.

Methods from these lines of work can make it easier to produce large labeled data sets, which are key to supervised learning. There are two broad approaches to combining rules-of-thumb. The well-studied *probabilistic* approach assumes that the labeling process conforms to a generative model and uses this model to determine the most likely label of each point. We take the Dawid-Skene estimator [1979] as a representative of this approach. On the other hand, the *game-theoretic* or *adversarial* approach, as developed in the work of Balsubramani and Freund [2015a] and others,

uses estimates of rule accuracies to generate a plausible set of labelings and chooses predictions that minimize the maximum possible error under this constraint.

Each approach has potential pitfalls. Probabilistic approaches can produce poor predictions if the rules-of-thumb violate the generative assumptions: i.e. under misspecification. The adversarial approach, on the other hand, can be too pessimistic given its preoccupation with mitigating the worst case. In this paper, we do a statistical analysis of the adversarial approach and prove various results for it including convergence. We find that similar properties do not hold for the Dawid-Skene probabilistic approach. The results are discussed through the lens of model and approximation uncertainty (to be defined later), providing a clearer image of the approaches. Empirical results corroborate our analysis.

To set the background, suppose that the label $y$ of any point $x \in \mathcal{X}$ is given by the conditional probability function $\eta(\ell \mid x) = \Pr(y = \ell \mid x)$, where $\ell \in \mathcal{Y} = \{1, 2, \dots, k\}$. In some applications, such as object detection, $\eta(\ell \mid x)$ will place almost all its mass on the single true label. In other settings, like predicting the course of a disease, there is inherent uncertainty and $\eta(\ell \mid x)$ will be spread over several labels $\ell$. We will look at methods that estimate the probabilities of different labels for the given data points $X = \{x_1, \dots, x_n\}$. That is, the probabilities $\eta = (\eta(\ell \mid x_i) : 1 \le i \le n, 1 \le \ell \le k)$. Generative approaches such as Dawid-Skene yield this readily. The solution space is $\Delta_k^n$, where $\Delta_k$ denotes the $k$-probability simplex: we select a distribution over $k$ labels for each of the $n$ data points.

For the adversarial approach, [Balsubramani and Freund, 2016] provide a framework that accommodates different loss functions for classification. Although their work focuses primarily on 0-1 loss, we use the log loss, which is more appropriate when label probabilities are sought. Given rules-of-thumb $h^{(1)}, \dots, h^{(p)}$, and estimates of their accuracies, the adversarial approach first defines a set $P \subset \Delta_k^n$ of plausible labelings; this takes estimation error into account and thus includes the true $\eta$. The goal is then to choose a model $g \in \Delta_k^n$ whose log-likelihood is maximized even for the worst-case "true" labeling $z \in P$:

$$\max_{g \in \Delta_k^n} \min_{z \in P} z \cdot \log g.$$

We show that the solution $g$ has several favorable properties.

1. (Maximum entropy) $g$ is the maximum entropy distribution in $P$.
2. (Form of solution) $g$ belongs to an exponential family of distributions $\mathcal{G}$ that can be defined in terms of the given rules-of-thumb.
3. (Logistic Regression) The minimax game is shown to be an instance of regularized logistic regression.
4. (Consistency) As the estimation error for rule accuracies goes to zero, $g$ converges to the model $g^* \in \mathcal{G}$ that

is closest to $\eta$ in KL-divergence.

5. (Rates of convergence) We bound the rate at which $g$ approaches $g^*$.
6. (Dawid-Skene comparison) For sufficiently good rule accuracy estimates, $g$ is guaranteed to be closer to $\eta$ in KL-divergence than the Dawid-Skene prediction.
7. (Empirical Results) Consistency is demonstrated on synthetic data and $g$ is compared to the Dawid-Skene prediction/other SOTA methods on real data.

Interestingly, the Dawid-Skene prediction is in the same family $\mathcal{G}$. However, we show it's not always consistent.

## 2 RELATED WORK

The study of WS is not only about constructing a classifier from rules-of-thumb, but encompasses all aspects of the process from start to end. Zhang et al. [2022] provide a good survey discussing the various aspects of a WS pipeline. The pipeline involves the creation of *labeling functions* (rules-of-thumb), creating a *label model* (classifier) to aggregate the rule predictions, and an *end model* trained on the label model's labeling of the data. These components can be separate, but can also be trained end-to-end, e.g. [Rühling Cachay et al., 2021b]. In our setting, the rules-of-thumb are fixed, but adding more rules has been studied, e.g. [Varma and Ré, 2018].

Since rules-of-thumb abstract the feature space, the domains to which WS is applicable varies widely. E.g. computer vision [Fu et al., 2019], natural language processing [Yu et al., 2021], medical applications [Wang et al., 2019].

Our representative for the probabilistic approach, created by Dawid and Skene [1979], has spawned of a myriad of models. Indeed, the recent work of Ratner et al. [2016] can be viewed as a generalization of the Dawid-Skene model where inter-rule dependencies are modeled. Dawid-Skene type estimators are well studied theoretically too, e.g. Gao and Zhou [2013] study the convergence of EM, Li and Yu [2014] provide finite sample error bounds, and Zhang et al. [2016] provide a provably good EM initialization.

A good survey of semi-supervised learning can be found in [Zhu and Goldberg, 2009]; the approaches taken are mostly probabilistic. A very different, game-theoretic/adversarial optimization, approach was introduced by [Balsubramani and Freund, 2015a] for binary classification and complete (non-abstaining) rules. Their work was generalized to accommodate partial rules in [Balsubramani and Freund, 2015b] and to a variety of different losses [Balsubramani and Freund, 2016]. In Arachie and Huang [2021], a similar optimization problem is considered with a focus on experiments. The work of Mazuelas et al. [2020] Mazzetto et al. [2021] and Mazuelas et al. [2022] give finite sample generalization bounds for rules learned under a similar adversarial

framework. In contrast, our bounds are in the transductive setting rather than the inductive one.

## 3 SETUP

Our goal is to label $n$ datapoints $X = \{x_1, \ldots, x_n\}$ whose labels lie in $\mathcal{Y} = [k]$ using $p$ rules of thumb $h^{(1)}, \ldots, h^{(p)}$ where $h^{(j)}\colon X \to \mathcal{Y} \cup \{?\}$ and "?" denotes an abstention. $\eta_{i\ell} = \Pr(y_i = \ell \mid x_i)$ is the true probability of class $\ell$ for $x_i$. We'll write that in vector form a la Mazuelas et al. [2020]:

$$\eta_i = (\eta_{i1}, \ldots, \eta_{ik}) \in \Delta_k \quad \text{and} \quad \eta = (\eta_1, \ldots, \eta_n) \in \Delta_k^n,$$

so that $\eta$ is a vector of length $nk$. Each rule's prediction $h^{(j)}(x_i)$ can be written as a vector in $\{0, 1\}^k$:

$$h_i^{(j)} = \begin{cases} \vec{e}_\ell & \text{if } h^{(j)}(x_i) = \ell \in [k] \\ \vec{0}_k & \text{if } h^{(j)}(x_i) = ? \end{cases} \quad (1)$$

$\vec{e}_\ell$ is the $\ell^{th}$ canonical basis vector in $k$ dimensions. Write

$$h^{(j)} = (h_1^{(j)}, \ldots, h_n^{(j)}) \in \Delta_k^n.$$

Thus $h_{i\ell}^{(j)}$ is 1 if $h^{(j)}(x_i) = \ell$ and 0 otherwise.

## 4 AN ADVERSARIAL APPROACH

Suppose we had upper and lower bounds on the accuracies of each rule $h^{(j)}$'s predictions on $X$. E.g. For instance, if these are based on $v$ labeled instances, then our estimates are accurate within $O(1/\sqrt{v})$. While there are $k^n$ possible labelings of $X$, knowing $h^{(j)}$ makes at most $v$ mistakes implies that only labelings whose Hamming distance is at most $v$ from $h^{(j)}$ are *coherent* with that piece of knowledge. This is a significant decrease. Every additional rule and the bounds for its mistakes on $X$ further constrains and shrinks the set of coherent labelings. We will soon see how this information effectively constrains the true labeling $\eta$ to lie in a specific polytope $P \subset \Delta_k^n$.

If $h^{(j)}$ makes $n_j \leq n$ predictions on $X = \{x_1, \ldots, x_n\}$, abstaining on the rest, the expected proportion of correct predictions is

$$b_j^* := \frac{1}{n_j} \sum_{i=1}^n \sum_{\ell=1}^k \eta_{i\ell} \mathbf{1}(h^{(j)}(x_i) = \ell) = \frac{1}{n_j} \eta \cdot h^{(j)}.$$

$b_j^*$ is the empirical accuracy of rule $j$. If $b_j$ is an estimate of $b_j^*$ and $\epsilon_j \geq 0$ so large that $b_j^* \in [b_j - \epsilon_j, b_j + \epsilon_j]$, we have

$$b_j - \epsilon_j \leq \frac{1}{n_j} \eta \cdot h^{(j)} \leq b_j + \epsilon_j. \quad (2)$$

For instance, $b_j$ could be an estimate from labeled data and $\epsilon_j$ could be from a binomial confidence interval.

Likewise, the empirical fraction of labels that are $\ell$ is

$$w_\ell^* = \frac{1}{n} \sum_{i=1}^n \eta_{i\ell} = \frac{1}{n} \eta \cdot \vec{e}_\ell^n$$

where $\vec{e}_\ell^n \in \{0, 1\}^{nk}$ is an $n$-fold repetition of $\vec{e}_\ell$. Like above, say $w_\ell$ is an estimate of $w_\ell^*$ and $\xi_\ell \geq 0$ so large that $w_\ell^* \in [w_\ell - \xi_\ell, w_\ell + \xi_\ell]$. We can then write

$$w_\ell - \xi_\ell \leq \frac{1}{n} \eta \cdot \vec{e}_\ell^n \leq w_\ell + \xi_\ell,$$

For brevity, take $m = p + k$, the number of constraints from rule accuracies and class frequencies. We'll abuse notation and let $b = (b_1, \ldots, b_p, w_1, \ldots, w_k)$. Similarly, we'll say $\epsilon = (\epsilon_1, \ldots, \epsilon_p, \xi_1, \ldots, \xi_k)$.

In writing the rule accuracy and class frequency bounds in matrix form, $A \in \mathbb{R}^{m \times nk}$, we construct a polytope of coherent labelings. Defined row-wise,

$$a^{(j)} = \begin{cases} h^{(j)}/n_j & \text{when} \quad 1 \leq j \leq p \\ \vec{e}_{j-p}^n/n & \text{when} \quad p+1 \leq j \leq p+k = m. \end{cases} \quad (3)$$

With element-wise inequalities, we can write the $m$ inequalities for rule accuracy and class frequency as $b - \epsilon \leq A\eta \leq b + \epsilon$ for $b, \epsilon \in \mathbb{R}^m$. The polytope of coherent labelings is defined by those inequalities for $b^*, w^*$:

$$P = \{z \in \Delta_k^n : b - \epsilon \leq Az \leq b + \epsilon\}. \quad (4)$$

For any $\epsilon \geq \vec{0}_m$, we require the interval $[b - \epsilon, b + \epsilon]$ to contain $b^*$, i.e. $\epsilon \to \vec{0}_m$ implies $b \to b^*$, because our adversarial approach requires that the underlying labeling $\eta$ be in $P$.

So, given information about the rule accuracies and class frequencies, the adversary can only choose labelings $z \in P$, i.e. labelings coherent with the information. Balsubramani and Freund [2015a] propose a two player zero-sum minimax game where the adversary seeks to maximize loss with their choice of labeling $z \in P$, while the learner attempts to minimize it with their prediction $g \in \Delta_k^n$. While they consider 0-1 loss, we use log loss/cross entropy as the game's objective. Said another way, the learner wishes to maximize log-likelihood while the adversary seeks to minimize it. The Balsubramani-Freund (BF) model's game can be written

$$V = \max_{g \in \Delta_k^n} \min_{z \in P} z \cdot \log g. \quad (5)$$

We will see that this is equivalent to a max-entropy type problem, and can be optimized either via gradient descent (as proposed by Balsubramani and Freund [2015a]) or via an off-the-shelf convex program solver.

# 5 STATISTICAL ANALYSIS OF BF

## 5.1 LEARNER'S PREDICTION IS MAXIMUM ENTROPY MODEL IN $P$

The learner's optimal prediction $g^{bf}$ from the minimax game in Equation 5 turns out to be the maximum entropy distribution in the polytope $P$ of labelings consistent with the accuracy and class frequency bounds. (Strictly speaking objects in $\Delta_k^n$ contain $n$ distributions, each over $k$ objects, but we call such objects distributions for brevity.)

**Theorem 1.** *The minimax game in Equation 5 can be equivalently written as follows.*

$$V = \max_{g \in \Delta_k^n} \min_{z \in P} z \cdot \log g = \min_{z \in P} \max_{g \in \Delta_k^n} z \cdot \log g = \min_{z \in P} z \cdot \log z$$

*The first expression defines a learner prediction $g^{bf}$, the second defines an adversarial labeling $z^*$. Then, $g^{bf} = z^*$ and they are the maximum entropy distribution in $P$, the optimal solution to the right-most expression.*

The general steps are to commute the min and max via Von Neumann's minimax theorem, apply Gibb's inequality repeatedly, and show the second and third problems have the same Lagrange dual. All proofs are found in the Appendix. For a more general treatment of minimax games and maximum entropy, see [Grünwald and Dawid, 2004].

## 5.2 CHARACTERIZING THE BF SOLUTION

We now show an easily optimizable dual of the sum of max entropies problem from Theorem 1. This exposes the functional form of $g^{bf}$. To be terse, for $\theta \in \mathbb{R}^m$, we will use $a^{(\theta)}$ as shorthand for $A^\top \theta = \theta_1 a^{(1)} + \cdots + \theta_m a^{(m)}$ where $a^{(j)}$ is row $j$ of $A$. Also, recall $g_{i\ell}$ is the learner's prediction for class $\ell$ on $x_i$ and $A, b, \epsilon$ together fully specify polytope $P$ (Equation 4).

**Theorem 2.** *The learner's optimal prediction $g$ for the game*

$$V = \max_{g \in \Delta_k^n} \min_{z \in P} z \cdot \log g \quad is \quad g_{i\ell} = \frac{\exp(a_{i\ell}^{(\sigma'-\sigma)})}{\sum_{\ell'} \exp(a_{i\ell'}^{(\sigma'-\sigma)})}.$$

*$\sigma, \sigma'$ are gotten from optimizing the dual problem*

$$V = V(b, \epsilon) = \max_{\sigma, \sigma' \geq \bar{0}_m} \left[ (\sigma' - \sigma) \cdot b - (\sigma' + \sigma) \cdot \epsilon \right.$$
$$\left. - \sum_{i=1}^n \log \left( \sum_{\ell=1}^k \exp(a_{i\ell}^{(\sigma'-\sigma)}) \right) \right].$$

The dual problem is concave in $2p + 2k$ variables $\sigma, \sigma'$ and can easily be solved with gradient descent or a convex program solver.

## 5.3 BF SOLUTION LIES IN AN EXPONENTIAL FAMILY

With matrix $A \in \mathbb{R}^{m \times nk}$, we can define a family of conditional probability distributions parameterized by $\theta \in \mathbb{R}^m$:

$$g_{i\ell}^{(\theta)} \propto \exp \left( \sum_{j=1}^m \theta_j a_{i\ell}^{(j)} \right) = \exp(a_{i\ell}^{(\theta)}). \qquad (6)$$

The family $\mathcal{G} = \{g^{(\theta)} : \theta \in \mathbb{R}^m\}$ has the characteristic exponential family form. We can treat $g^{(\theta)}$ as a vector in $\Delta_k^n$. If we take the optimal $\sigma', \sigma$ from Theorem 2 and define $\theta^{bf} = \sigma' - \sigma$, the learner's best-play $g^{bf}$ is $g^{(\theta^{bf})}$.

## 5.4 BF IS A FORM OF LOGISTIC REGRESSION

Since BF is a maximum entropy problem, we can relate it to multi-class logistic regression (MLR) with $\ell_1$ regularization. This connection was previously observed by Mohri et al. [2018] (Chapter 13 Section 7) and Mazuelas et al. [2022].

To start, we briefly review the formulation for MLR. Suppose we have datapoint features $x_i \in \mathbb{R}^d$ and their label distributions $\eta_i \in \Delta_k$ for $i \in [n]$. For each $x_i$, the goal is to predict each class' probability (elements of $\eta_i$) by using different weighted combinations of $x_i$. Formally, we wish to learn $w_\ell \in \mathbb{R}^d$ for every $\ell \in [k]$ such that

$$g_{i\ell}^{lr} = \frac{\exp(w_\ell^\top x_i)}{\sum_{\ell'=1}^k \exp(w_{\ell'}^\top x_i)} \quad \text{approximates} \quad \eta_{i\ell}.$$

If $w_\ell$ serves as row $\ell$ of a weight matrix $W \in \mathbb{R}^{k \times d}$, the prediction for datapoint $x_i$ is the softmax of elements in the vector $W x_i$. To learn $W$, one minimizes cross entropy, which can be regularized with coefficient $C$:

$$\min_{W \in \mathbb{R}^{k \times d}} \left[ -\eta^\top \log g^{lr} + C \sum_{\ell=1}^k \|w_\ell\|_1 \right]$$

To connect the BF problem to the above, it suffices to show two things. First is that the prediction $g^{bf}$ can be written as the softmax of a weight matrix times datapoint features. Second, the dual objective in Theorem 2 can be turned into a cross entropy term plus $\ell_1$ type regularization.

For the first point, consider $g = g^{(\theta)} \in \mathcal{G}$ where $\theta \in \mathbb{R}^m$. Looking at the $i^{th}$ datapoint, define $A_i \in \mathbb{R}^{m \times k}$ to be columns $k(i-1) + 1, \ldots, ki$ of matrix $A$. $A_i$ contains the one-hot encoding of the rule predictions on $x_i$ and all canonical basis vectors in $k$ dimensions (Equation 3). Then, $g_i$ is the softmax of $A_i^\top \theta$ (Equation 6). Observe that the weights $\theta$ are in vector rather than matrix form. We show in the appendix how to rewrite $A_i^\top \theta$ so that it is equal to a weight *matrix* $T_\theta \in \mathbb{R}^{k \times mk}$ times feature *vector* $\hat{x}_i \in \mathbb{R}^{km}$ (taking elements from $A_i$).

Now, rather than have one regularization coefficient $C$, BF actually has $2m$, two for each weight in $\theta \in \mathbb{R}^m$. This is because weight $\theta_j$ will have a different regularization coefficient depending on whether it's positive or negative. So, let $\theta_j = \sigma'_j - \sigma_j$ with $\sigma'_j, \sigma_j$ being the positive and negative parts of $\theta_j$ respectively. We'll now see what those regularization coefficients are.

**Lemma 3.** *BF is a logistic regression type classifier. Fix $A, \eta, b, \epsilon$, which fixes $b^* = A\eta$, $P$, and the BF prediction $g^{bf} = g^{(\theta^{bf})}$. The BF dual problem from Theorem 2 can be rewritten as*

$$- V(b, \epsilon) = \min_{\theta \in \mathbb{R}^m} \Big[ -\eta^\top \log g^{(\theta)} + \underbrace{(b^* - (b - \epsilon)) \cdot \sigma' + (b + \epsilon - b^*) \cdot \sigma}_{regularization} \Big].$$

*The optimal weights $\theta$ from above equals $\theta^{bf}$. Moreover, for every $i \in [n]$, BF's prediction $g_i^{bf}$ for datapoint $i$ is the softmax of $T_\theta \widehat{x}_i$, a weight matrix times a "feature" vector.*

The proof is deferred to the Appendix where we'll also show how to convert any MLR problem into an instance of BF.

For regular $\ell_1$ regularization, we would see the term

$$C\|\theta\|_1 = C\sum_{j=1}^m |\theta_j| = \sum_{j=1}^m \big[ C\sigma'_j + C\sigma_j \big]$$

in the objective because $|\theta_j| = \sigma'_j + \sigma_j$. Since $b^* \in [b - \epsilon, b + \epsilon]$ by assumption, our $2m$ regularization coefficients are all non-negative. In the Appendix, we'll see $\sigma_j$ is associated with constraint $a^{(j)} z \leq b_j + \epsilon_j$, the $j^{th}$ upper bound constraint in $P$ (Equation 4). $\sigma_j$'s regularization coefficient $b_j + \epsilon_j - b^*_j$ depends on how poor the upper bound $b_j + \epsilon_j$ is for the empirical accuracy $b^*_j$. Similarly, $\sigma'_j$ is associated with the constraint $b_j - \epsilon_j \leq a^{(j)} z$ and its regularization coefficient $b^*_j - (b_j - \epsilon_j)$ measures how poorly $b_j - \epsilon_j$ lower bounds $b^*_j$.

To finish this section, note that for MLR, if the datapoint features number in $d$, i.e. $x_i \in \mathbb{R}^d$, one learns $kd$ weights since $W \in \mathbb{R}^{k \times d}$. For BF, since $\widehat{x}_i \in \mathbb{R}^{km}$, there will be $km = k(p + k)$ so called "datapoint features". However, only $m = p + k$ weights will be learned as $T_\theta \in \mathbb{R}^{k \times mk}$ is completely characterized by $m$ values.

## 5.5 MODEL AND APPROXIMATION UNCERTAINTY

To eventually draw the link between the loss of the BF and Dawid-Skene (DS) prediction, we will need a more granular notion of loss or *uncertainty*. While the ultimate goal is to infer the labels of $X$, the best one can in general hope for is to infer $\eta$, the label distribution for each datapoint. The

loss one incurs from predicting $\eta$ rather than the true labels is irreducible and called *aleatoric uncertainty*. On the other hand, the loss in estimating $\eta$ is nominally reducible and called *epistemic uncertainty*, denoted $\mathcal{E}$.

The sources of $\mathcal{E}$ are twofold and depend on the method chosen to estimate $\eta$. When a method is chosen, the set of predictions for that method is fixed. For BF as the chosen method, its set of predictions is $\mathcal{G}$. If $\eta \notin \mathcal{G}$, then one immediately incurs loss for choosing a method that cannot predict $\eta$. That quantity is the *model uncertainty* or $\mathcal{E}^{mod}$. If $d(\cdot, \cdot)$ measures distance between $\mu, \nu \in \Delta_k^n$, the model uncertainty is $\mathcal{E}^{mod} = \min_{g \in \mathcal{G}} d(\eta, g)$. That distance is defined later. If we suppose the best approximator to $\eta$ with respect to $d(\cdot, \cdot)$ is unique, the failure of a method to produce the best approximator $g^* := \arg\min_{g \in \mathcal{G}} d(\eta, g)$ for $\eta$ is regarded as *approximation uncertainty $\mathcal{E}^{appr}$*. Approximation uncertainty can be from poor estimates of rule accuracies/class frequencies or be pathological to the method. If the model produces prediction $g$, then $\mathcal{E}^{appr} = d(g^*, g)$. To summarize, $\mathcal{E} = \mathcal{E}^{mod} + \mathcal{E}^{appr}$ and for appropriately chosen $d(\cdot, \cdot)$,

$$\mathcal{E} = d(\eta, g), \quad \mathcal{E}^{mod} = d(\eta, g^*), \quad \mathcal{E}^{appr} = d(g^*, g).$$

For example, a sufficiently deep neural network is a universal approximator, i.e. $\mathcal{E}^{mod} = 0$ because it can predict any continuous function. Any epistemic uncertainty would be from the approximation uncertainty, e.g. from small training sets, optimization difficulties. See the Appendix and especially [Hüllermeier and Waegeman, 2021] for a more subtle and complete discussion of these concepts.

## 5.6 A PYTHAGOREAN THEOREM FOR $\mathcal{G}$

In the previous section, we considered $d(\eta, g^*), d(g^*, g)$, corresponding to model and approximation uncertainty respectively. By using KL divergence as our distance, a sufficiently nice $g^*$ can simplify the sum of those terms by way of a Pythagorean theorem. For $\mu, \nu \in \Delta_k^n$, we define

$$d(\mu, \nu) = \sum_{i=1}^n KL(\mu_i, \nu_i).$$

If $\mu = \eta$ and $\nu = g$, $d(\eta, g)$ sums the KL divergence of our prediction $g_i \in \Delta_k$ for point $x_i$ against the underlying conditional label distribution $\eta_i \in \Delta_k$ for each datapoint in $X$. Now, if $g^*$ as a labeling produces the same rule accuracies and class frequencies as $\eta$ (namely $Ag^* = A\eta$), then $d(\eta, g^*) + d(g^*, g)$ equals a single $d(\cdot, \cdot)$ term.

**Lemma 4.** *Pick any $\theta, \theta' \in \mathbb{R}^m$ and write $g = g^{(\theta)}$, $g' = g^{(\theta')}$. If $g$ satisfies $Ag = A\eta$, then*

$$d(\eta, g') = d(\eta, g) + d(g, g').$$

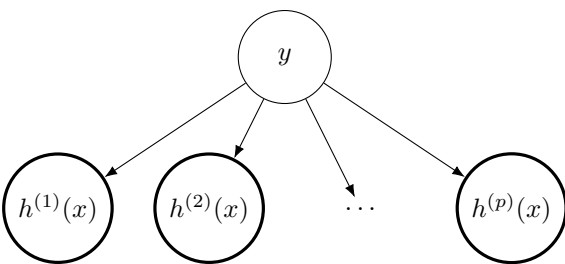

Figure 1: Dawid-Skene Graphical Model

## 5.7 BF'S CONSISTENCY AND ITS RATE OF CONVERGENCE

The Pythagorean theorem just shown allows us to decompose the BF prediction's (or learner's best play's) loss $d(\eta, g^{bf})$ into model and approximation uncertainty. We will see that the best approximator to $\eta$ in $\mathcal{G}$ with respect to KL divergence, i.e. the minimizer to $d(\eta, \cdot)$, is unique. Hence, we can define $g^* = g^{(\theta^*)} := \arg\min_{g \in \mathcal{G}} d(\eta, g)$. We can show that $Ag^* = A\eta$ (see Appendix) and use Lemma 4 to get the following.

**Lemma 5.**

$$\underbrace{d(\eta, g^{bf})}_{\mathcal{E}_{bf}} = \underbrace{d(\eta, g^*)}_{\mathcal{E}_{bf}^{mod}} + \underbrace{d(g^*, g^{bf})}_{\mathcal{E}_{bf}^{appr}}.$$

In the following, we upper-bound $\mathcal{E}_{bf}^{appr}$ by a linear function of $\epsilon$, the widths of the interval containing $b^*$. Moreover, we will see that $\mathcal{E}_{bf}^{appr} \to 0$ as $\epsilon \to \vec{0}_m$, meaning the loss of BF's prediction $g^{bf}$ tends to the smallest possible loss, i.e. the model uncertainty. If a method's approximation uncertainty can be brought down to 0 for every problem, we call it *consistent*. Therefore, we conclude BF is consistent.

**Theorem 6.** *The $g \in \mathcal{G}$ minimizing $d(\eta, g)$ is unique. Call it $g^*$. When $\epsilon = \vec{0}_m$, the learner's prediction gotten from solving $V(b^*, \vec{0}_m)$ is exactly $g^*$.*

In bounding $\mathcal{E}_{bf}^{appr}$, we have the rate of convergence for $g^{bf} \to g^*$ as $\epsilon \to \vec{0}_m$ in terms of $\epsilon$ and $g^*$'s weights $\theta^*$.

**Theorem 7.** *For fixed $g^*$ and $|\cdot|$ acting element-wise,*

$$d(g^{(\theta^*)}, g^{bf}) \leq 2\epsilon^\top |\theta^*| \leq 2\|\epsilon\|_\infty \|\theta^*\|_1 = O(\|\epsilon\|_\infty)$$

## 6 A PROBABILISTIC APPROACH

To estimate the probability of labels, the DS model posits an underlying generative model. By fitting the parameters of this model, one is able to obtain estimates of label probabilities for every datapoint. The model assumed is simple: for a datapoint and its label $(x, y) \in \mathcal{X} \times \mathcal{Y}$, rule predictions $h^{(1)}(x), \ldots, h^{(p)}(x)$ are conditionally independent given label $y$ (see Figure 1).

The parameters of the DS model are the underlying class frequencies $\Pr(y = \ell)$ and $p$ confusion matrices. Each rule $h^{(j)}$ is parameterized by its underlying row stochastic confusion matrix $B_j = b_{j\ell\ell'} \in [0, 1]^{k \times k}$ where

$$b_{j\ell\ell'} = \Pr(h^{(j)}(x) = \ell' \mid y = \ell).$$

Since the rule predictions are conditionally independent given the label, the true posterior label probability is

$$\Pr(y \mid h^{(j)}(x), j \in [p]) = \frac{\Pr(y, h^{(j)}(x), j \in [p])}{\Pr(h^{(j)}(x), j \in [p])}$$

$$= \frac{\Pr(y) \prod_{j=1}^p \Pr(h^{(j)}(x) \mid y)}{\sum_{\ell=1}^k \Pr(y = \ell) \prod_{j=1}^p \Pr(h^{(j)}(x) \mid y = \ell)}. \quad (7)$$

Note that when anything other than the underlying probabilities are substituted in the bottom expression, the last equality does not hold. In practice, one estimates the confusion matrices and class frequencies, e.g. by EM, and plugs them in the right hand side in lieu of their empirical counterparts.

We present results involving the one-coin DS model (OCDS), where each rule is a biased coin. While $b_j$ is an estimate of $b_j^*$ in other parts of the paper, take $b_j$ to mean $\Pr(h^{(j)}(x) = y)$ in this section only. I.e. $b_j$ is the underlying bias (resp. accuracy) of coin (resp. rule) $j$. Diagonal elements of the confusion matrix $b_{j\ell\ell}$ equal $b_j$, while all off diagonal elements $b_{j\ell\ell'}$ equal $(1 - b_j)/(k - 1)$.

While OCDS is one of the simplest generative models, it is representative. The more complex probabilistic models have similar characteristics in terms of consistency. The comparison of those more complex probabilistic models to BF is taken up in the experiments.

## 7 RELATION AND COMPARISON TO ADVERSARIAL APPROACH

### 7.1 DS PREDICTION IN SAME EXPONENTIAL FAMILY

If $g^{ds}$ is a OCDS prediction, then Equation 7 gives its form:

$$g_{i\ell}^{ds} \propto w_\ell \prod_{j=1}^p b_j^{\mathbf{1}(h^{(j)}(x_i)=y_i)} \left(\frac{1 - b_j}{k - 1}\right)^{\mathbf{1}(h^{(j)}(x_i)\notin\{y_i,?\})}.$$

This is the result of running OCDS' E step from its EM algorithm with class frequencies $w \in \Delta_k$ and rule accuracies $b \in [0, 1]^m$. Note that if a rule $j$ abstains on $x_i$, i.e. $h^{(j)}(x_i) = ?$, then it makes no contribution to the prediction on $x_i$. Now, we can exhibit weights $\theta^{ds}$ such that $g^{(\theta^{ds})} = g^{ds}$ from above. This means that $g^{ds} \in \mathcal{G}$. The next fact was observed by Li and Yu [2014] in Corollary 9.

**Lemma 8** (OCDS Weights)**.** *Suppose $0 < w_\ell, b_j < 1$ for $\ell \in [k], j \in [p]$ and $g^{ds}$ is the OCDS prediction as above. If*

$\theta^{ds} \in \mathbb{R}^m$ *is defined as*

$$\theta_j^{ds} = \log\left(e^{n_j} \frac{b_j(k-1)}{1-b_j}\right) \quad and \quad \theta_{p+\ell}^{ds} = \log(e^n w_\ell)$$

*for $j \in [p]$ and $\ell \in [k]$, then $g^{(\theta^{ds})} = g^{ds}$.*

Since $\mathcal{G}$ is essentially parameterized by all real weights, we have to avoid weights that are infinite – hence the restriction on $w, b$. Therefore, $\mathcal{G}$ doesn't contain all OCDS predictions.

**Lemma 9** (Informal). *$\mathcal{G}$ contains all one-coin DS predictions constructed by class frequencies $w_\ell$ and rule accuracies $b_j$ each not equal to $0$ or $1$.*

### 7.2 COMPARING BF WITH DS

Suppose we took an arbitrary OCDS prediction $g^{ds}$. This could be from EM after convergence, a single E step given estimates of the accuracies/class frequencies, etc. To continue our discussion, we need to know what OCDS estimates the rule accuracies and class frequencies to be. For $g^{ds}$, performing the M step (essentially $Ag^{ds}$) gives those quantities. Call those estimates $b^{ds} = (b_1^{ds}, \ldots, b_p^{ds})$ and $w^{ds} = (w_1^{ds}, \ldots, w_k^{ds})$ respectively. Also, recall that the rule accuracy/class frequency estimates given to BF are $b$, $w$ respectively while $b^*$, $w^*$ denote the empirical values.

To compare BF to OCDS, we will also decompose OCDS' epistemic uncertainty into model and approximation uncertainty. Let $g^{ds*}$ be the OCDS prediction formed by using the empirical rule accuracies and class frequencies (i.e. do one E step with $b^*$, $w^*$). While $\mathcal{G}$ doesn't contain all OCDS predictions, it's sufficiently large so that BF and OCDS have the same model uncertainty.

**Lemma 10.**

$$\underbrace{d(\eta, g^{ds})}_{\mathcal{E}_{ds}} = \underbrace{d(\eta, g^*)}_{\mathcal{E}_{ds}^{mod}} + \underbrace{d(g^*, g^{ds})}_{\mathcal{E}_{ds}^{appr}}$$

$$\mathcal{E}_{ds}^{appr} = \underbrace{d(\eta, g^{ds*}) - d(\eta, g^*)}_{\mathcal{E}_{ds,1}^{appr}} + \underbrace{d(\eta, g^{ds}) - d(\eta, g^{ds*})}_{\mathcal{E}_{ds,2}^{appr}}$$

The first equality follows from our Pythagorean theorem (Lemma 5). One easily sees by substitution that the second equality is true. By definition, $\mathcal{E}_{ds,1}^{appr} \geq 0$. Moreover, it only depends on $A, \eta$ (because $b^*$, $w^*$ are from $A\eta$). Thus, as soon as the rules predictions and true label distribution are fixed (i.e. $A$ and $\eta$ are fixed), $\mathcal{E}_{ds,1}^{appr}$ is fixed. $\mathcal{E}_{ds,2}^{appr}$ essentially measures how well $b^{ds}$, $w^{ds}$ estimate $b^*$, $w^*$. While possibly negative, it tends to 0 as $b^{ds} \to b^*$ and $w^{ds} \to w^*$. (Its exact form in terms of those quantities is shown in the Appendix.) In the experiments, we show how the individual contributions of $\mathcal{E}_{ds,1}^{appr}$ and $\mathcal{E}_{ds,2}^{appr}$ can vary for real datasets.

Before comparing BF and OCDS' predictions, we first discuss OCDS' consistency. For OCDS to be consistent, we need to be able to bring $\mathcal{E}_{ds}^{appr}$ down to 0 for every problem. To be concrete, we cannot talk about OCDS' consistency in a vacuum. The OCDS generative assumption only defines the functional form of the posterior label probability (in terms of the underlying class frequencies/rule accuracies, Equation 7). In other words, the OCDS generative assumption only defines the model uncertainty. To get a prediction, one has to estimate those underlying quantities, e.g. by running EM. Thus, we have to talk about the consistency of OCDS *alongside* an algorithm. In this paper, we focus our attention on the consistency of OCDS paired with EM.

OCDS with EM is not consistent because we exhibit a problem $(A, \eta)$ in the Appendix where EM never converges to $g^*$. Note that since $g^*$ is unique (Theorem 6), we only have to check convergence at that point. Essentially, if EM starts at $g^*$ for that problem, then an M step followed by an E step results in $g^{ds*} \neq g^*$. This is because applying the M step to $g^*$ gives $b^*, w^*$ – we show in the appendix that $Ag^* = A\eta = b^*$. The E step with those quantities is $g^{ds*}$ by definition.

**Lemma 11.** *OCDS with EM is inconsistent because there exists a problem where EM doesn't converge at $g^*$.*

For the BF prediction to be better than OCDS', we need $d(\eta, g^{bf}) \leq d(\eta, g^{ds})$. Using our loss decompositions, we want to know when $\mathcal{E}_{bf}^{mod} + \mathcal{E}_{bf}^{appr} \leq \mathcal{E}_{ds}^{mod} + \mathcal{E}_{ds}^{appr}$. Since Lemmata 5, 10 show $\mathcal{E}_{bf}^{mod} = \mathcal{E}_{ds}^{mod}$, we present a sufficient condition for $\mathcal{E}_{bf}^{appr} \leq \mathcal{E}_{ds}^{appr}$ to hold.

**Lemma 12.** *Fix OCDS prediction $g^{ds}$. If $\epsilon$ (for BF) is s.t.*

$$\|\epsilon\|_\infty \leq \frac{d(\eta, g^{ds*}) - d(\eta, g^*) + d(\eta, g^{ds}) - d(\eta, g^{ds*})}{2\|\theta^*\|_1},$$

*then $d(\eta, g^{bf}) \leq d(\eta, g^{ds})$.*

*Proof Sketch.* By Theorem 7, $\mathcal{E}_{bf}^{appr} \leq 2\|\epsilon\|_\infty \|\theta^*\|_1$. By Lemma 10, $\mathcal{E}_{ds}^{appr} = d(\eta, g^{ds*}) - d(\eta, g^*) + d(\eta, g^{ds}) - d(\eta, g^{ds*})$. Substitute into $\mathcal{E}_{bf}^{appr} \leq \mathcal{E}_{ds}^{appr}$ and rearrange. $\square$

Because $\mathcal{E}_{ds}^{appr} = d(g^*, g^{ds}) \geq 0$, the upper bound is non-negative, showing the existence of a ball that $\epsilon$ has to lie in for BF to have a better prediction than OCDS. We note that this bound is very conservative and is not useful in practice except for very small $\epsilon$.

## 8 EXPERIMENTAL RESULTS

We now provide experimental results comparing BF to one coin DS, but also other SOTA Weak Supervision (WS) methods on 10 real datasets. Visualiza-

Table 1: Dataset Statistics

| Name, Dataset Source, Rule Source | # Rules ($p$) | # Train ($n$) | # Valid |
|---|---|---|---|
| AwA [Xian et al., 2019], [Mazzetto et al., 2021] | 36 | 1372 | 172 |
| Basketball [Fu et al., 2020], [Fu et al., 2020] | 4 | 17970 | 1064 |
| Cancer [Wolberg et al., 1995], [Arachie and Huang, 2018] | 3 | 171 | 227 |
| Cardio [de Campos and Bernardes, 2010], [Arachie and Huang, 2018] | 3 | 289 | 385 |
| Domain [Peng et al., 2019], [Mazzetto et al., 2021] | 5 | 2587 | 323 |
| IMDB [Maas et al., 2011], [Ren et al., 2020] | 8 | 20000 | 2500 |
| OBS [Rajab, 2017], [Arachie and Huang, 2018] | 3 | 239 | 317 |
| SMS [Almeida and Hidalgo, 2012], [Awasthi et al., 2020] | 73 | 4571 | 500 |
| Yelp [Zhang et al., 2015], [Ren et al., 2020] | 8 | 30400 | 3800 |
| Youtube [Alberto and Lochter, 2017], [Snorkel AI Inc., 2022] | 10 | 1586 | 120 |

Table 2: Comparison of BF Against Other WS Methods Using Average Log Loss

| Method | AwA | Basketball | Cancer | Cardio | Domain | IMDB | OBS | SMS | Yelp | Youtube |
|---|---|---|---|---|---|---|---|---|---|---|
| MV | 0.31 | 2.40 | 14.87 | 0.66 | 5.48 | 6.39 | 8.73 | 0.79 | 5.90 | 1.27 |
| OCDS | 0.24 | 3.75 | 4.46 | 13.74 | 22.32 | 2.91 | 6.28 | 0.78 | 1.73 | 17.63 |
| DP | 0.42 | 1.31 | 6.14 | 7.01 | 9.21 | 0.68 | 3.98 | 0.53 | 2.61 | 0.72 |
| EBCC | **0.13** | 0.45 | 4.25 | 0.90 | 1.80 | 0.73 | 2.23 | 0.43 | 0.81 | 0.69 |
| HyperLM | 0.21 | 1.31 | 6.93 | 0.60 | 1.29 | 0.62 | 2.66 | 0.68 | **0.60** | **0.42** |
| AMCL CC | **0.14** | 1.26 | 14.86 | 0.42 | 5.42 | 1.46 | 8.73 | 0.69 | 0.85 | 0.70 |
| BF | **0.13** | **0.39** | **0.68** | **0.20** | **1.12** | **0.59** | **0.61** | **0.42** | 0.64 | 0.50 |
| $\frac{1}{n}d(\eta, g^*)$ | 0.01 | 0.32 | 0.65 | 0.13 | 1.01 | 0.57 | 0.59 | 0.25 | 0.54 | 0.21 |

Table 3: Comparison of BF Against Other WS Methods Using Average 0-1 Loss and Average Brier Score

| Method | AwA | | Basketball | | Cancer | | Cardio | | Domain | | IMDB | | OBS | | SMS | | Yelp | | Youtube | |
|---|---|---|---|---|---|---|---|---|---|---|---|---|---|---|---|---|---|---|---|---|
| | 0-1 | BS | 0-1 | BS | 0-1 | BS | 0-1 | BS | 0-1 | BS | 0-1 | BS | 0-1 | BS | 0-1 | BS | 0-1 | BS | 0-1 | BS |
| MV | **1.31** | 0.15 | 24.54 | 0.31 | 52.05 | 0.95 | 34.95 | 0.35 | 45.73 | 0.62 | 29.40 | 0.47 | **27.62** | 0.54 | 31.92 | 0.32 | **31.84** | 0.49 | **18.79** | **0.23** |
| OCDS | 2.11 | 0.04 | **11.29** | **0.23** | 52.05 | 1.02 | 39.79 | 0.80 | 80.17 | 1.60 | 49.81 | 0.95 | **27.62** | 0.55 | 9.67 | **0.18** | 46.74 | 0.72 | 52.40 | 1.05 |
| DP | 3.15 | 0.06 | **11.29** | **0.23** | 50.88 | 1.01 | 39.79 | 0.80 | 72.51 | 1.36 | 30.48 | 0.45 | **27.62** | 0.55 | 32.19 | 0.36 | 46.78 | 0.71 | 34.75 | 0.40 |
| EBCC | **1.57** | **0.03** | 36.33 | 0.29 | 52.05 | 1.03 | 39.79 | 0.62 | 48.23 | 0.74 | 28.26 | 0.45 | **27.62** | 0.55 | **8.16** | 0.25 | 36.02 | 0.51 | 52.40 | 0.50 |
| HyperLM | 2.55 | 0.10 | 36.36 | 0.45 | 52.05 | 0.94 | 7.96 | 0.31 | 41.98 | 0.65 | **27.74** | 0.41 | **27.62** | 0.45 | 53.73 | 0.50 | 32.92 | **0.41** | 20.37 | 0.26 |
| AMCL CC | 2.00 | 0.06 | 12.14 | **0.23** | 49.18 | 0.93 | **3.11** | **0.06** | 36.82 | 0.54 | 31.74 | 0.46 | **27.62** | 0.54 | 45.04 | 0.49 | 37.39 | 0.48 | 38.88 | 0.47 |
| BF | 3.67 | 0.06 | **11.40** | 0.22 | **40.47** | **0.49** | **3.11** | 0.08 | **36.75** | 0.55 | 29.33 | **0.41** | **27.62** | **0.42** | 13.50 | 0.25 | **34.42** | 0.45 | 24.34 | 0.33 |
| $g^*$ | 0.58 | 0.01 | 11.27 | 0.19 | 36.26 | 0.46 | 3.11 | 0.06 | 37.26 | 0.51 | 28.74 | 0.38 | 27.62 | 0.40 | 8.09 | 0.14 | 26.54 | 0.36 | 7.31 | 0.12 |

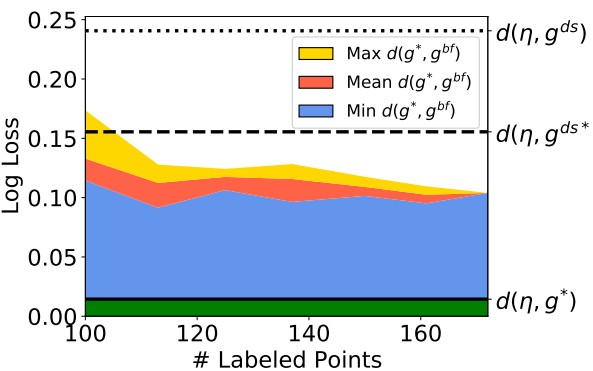

Figure 2: BF/OCDS loss breakdowns on the AwA dataset where $\mathcal{E}_{ds,1}^{appr} = d(\eta, g^{ds*}) - d(\eta, g^*)$ is large. The green section (below solid line) is loss incurred by any prediction in $\mathcal{G}$.

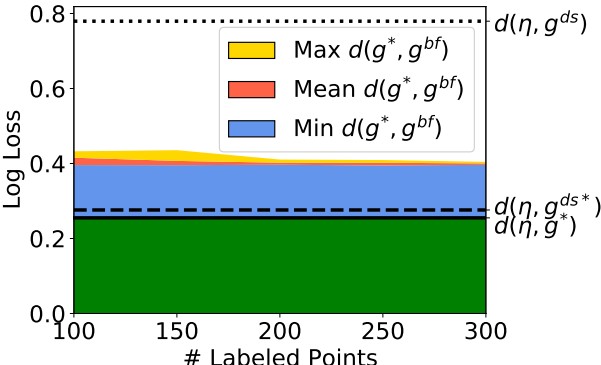

Figure 3: BF/OCDS loss breakdowns on the SMS dataset where $\mathcal{E}_{ds,1}^{appr}$ is small.

tions of BF and OCDS' model and approximation uncertainty are given, along with a synthetic experiment demonstrating BF's consistency. See the supplementary material or https://github.com/stevenan5/balsubramani-freund-uai-2024 for the code and the Appendix for the complete experimental results.

The other WS methods compared are Majority Vote (MV), Data Programming (DP) [Ratner et al., 2016], a popular generative WS method, EBCC [Li et al., 2019], a Bayesian method, HyperLM [Wu et al., 2023], a Graph NN method, and AMCL 'Convex Combination' [Mazzetto et al., 2021], another adversarial WS method. All but BF and AMCL CC are implemented in WRENCH [Zhang et al., 2021]. We consider 10 real datasets from varying domains. See Table 1 for the dataset/rule sources and some relevant statistics. Apart from Domain with $k = 5$ classes, all other datasets had $k = 2$ classes. Unless provided, validation sets are from random splits with split sizes following the cited authors. We evaluate how well each method labels the training set.

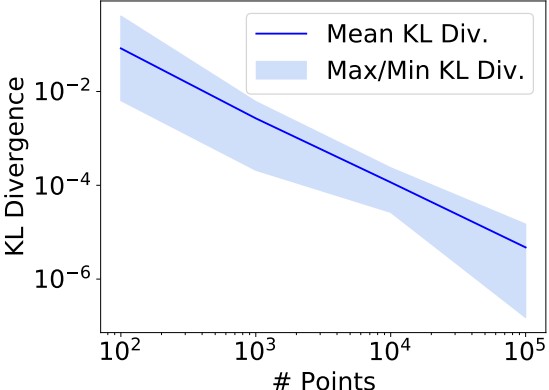

Figure 4: Convergence of $g^*$ to $\eta$ on a synthetic dataset as $n$ increases.

BF, DP, EBCC, AMCL CC are each run 10 times due to randomness in their methodology. For BF and AMCL CC, the randomness is from the sampling of 100 labeled datapoints from the validation set. For BF, the raw estimates of rule accuracies and class frequencies (our $b$) are bounded via the Wilson score interval (following Brown et al. [2001]) with confidence $0.95$ to obtain $\epsilon$. OCDS' EM algorithm is initialized with the majority vote labeling and is run until convergence, while DP and EBCC are run with the default hyperparameters supplied in WRENCH.

Table 2 shows the average log loss $\frac{1}{n}d(\eta, g)$ of each method while Table 3 shows the average 0-1 loss and average Brier score. Bold entries are indistinguishable via two-tailed paired t-test with $p = 0.05$. For Table 2, we see that BF's prediction has low log loss, beating out other methods on most datasets. Also included is the BF/OCDS model uncertainty $\frac{1}{n}d(\eta, g^*)$. We can get $g^*$ by taking $\eta$ to be the ground truth labels and solving $V(b^*, \vec{0}_m)$ a la Theorem 6. The values for $\frac{1}{n}d(\eta, g^*)$ being smaller than the values of $\frac{1}{n}d(\eta, g^{bf})$ (the row above), is evidence for BF's consistency (i.e. BF can give the best estimator $g^*$). For Table 3, we see that the loss for the BF prediction $g^{bf}$ is reasonable with respect to average 0-1 loss and average Brier score despite being chosen to guard against worst case log loss.

Figures 2, 3 show how OCDS' loss breaks down when we judge how well EM approximates the empirical rule accuracies and class frequencies. For every quantity of labeled points, BF is run 10 times and the min/mean/max BF approximation uncertainties are plotted. (Since we can compute $d(\eta, g^{bf})$ and $d(\eta, g^*)$, we know $\mathcal{E}_{bf}^{appr}$ via Lemma 5.) To compute $d(\eta, g^{ds*}) = \mathcal{E}_{ds,1}^{appr} + d(\eta, g^*)$, we plug in the empirical rule accuracies $b^*$/class frequencies $w^*$ into the OCDS prediction – i.e. do one E step with those quantities. Note that to maintain consistency with Table 2, we plot the average loss rather than absolute loss. E.g. we write $d(\eta, g^*)$ when we've actually plotted $\frac{1}{n}d(\eta, g^*)$. For AwA, $d(\eta, g^{ds*})$

(dashed line) is essentially the same as the BF loss. Compare that to SMS, where that same quantity is very close to the model uncertainty $d(\eta, g^*)$, meaning OCDS' reducible error mainly comes from $\mathcal{E}_{ds,2}^{appr}$, or poor estimates of the rule accuracies and class frequencies from EM.

Finally, we generated 10 binary label datasets with 3 rules and $10^5$ datapoints under the one-coin DS assumption. The label distribution was drawn from $Dirichlet(1,1)$ while each rule's accuracy was drawn from $Beta(2, 4/3)$. For each dataset, BF was run on the first $10^2, 10^3, 10^4, 10^5$ datapoints, being given the empirical rule accuracies and class frequencies each time. We measure $\frac{1}{n}d(\eta, g^*)$, the average KL divergence between the BF prediction and the underlying generative distribution $\eta$. This is done to empirically demonstrate our notion of consistency under a simple generative setting. Indeed, it's easy to show by inspection that $g^{ds*} \to \eta$ as $n \to \infty$. It turns out that $g^* \to \eta$ as $n \to \infty$ too. To be clear, we have abused notation – for each $n$, we can get $g^{ds*}, g^*$ via one E step/running BF respectively. We want to remind the reader that the OCDS model generates a *hard* label for each datapoint. Thus, the empirical rule accuracies/class frequencies (for fixed $n$) do not match their underlying values. Since the BF prediction $g^*$ will induce the empirical rule accuracies/class frequencies, i.e. $Ag^* = b^*$, not equal to the underlying rule accuracies/class frequencies, $g^* \neq \eta$ even though $\eta \in \mathcal{G}$. Figure 4 shows a log-log scale graph of min/average/max of $\frac{1}{n}d(\eta, g^*)$ versus the number of datapoints. We see that the average KL divergence to the underlying distribution decreases exponentially fast.

# 9 DISCUSSION

The theoretical and empirical results presented in this paper point toward the viability of adversarial weak supervision methods in general. Multiple theoretical results are presented in support while experimental results demonstrate real world performance.

First, we show the close relationship of BF with $\ell_1$ regularized multi-class logistic regression.

Second, in showing that BF and OCDS have the same model uncertainty, we not only reduce the problem of their comparison to comparing their approximation uncertainties, but we ensured a fair comparison. I.e. neither model had an advantage out the gate by being more expressive than the other. By comparing approximation uncertainties, we deduced the existence of a region of $\epsilon$'s where BF's performance is no worse than OCDS (Lemma 12). Moreover, BF's approximation uncertainty only depends on how well the empirical rule accuracies/class frequencies are estimated (Theorem 7) while OCDS' approximation uncertainty has a term that depends on the problem specification. This means BF is consistent while OCDS with EM is not (Lemma 11).

Third, for adversarial methods to be viable in practice, it's necessary (but not sufficient) that they be competitive or outperform unsupervised methods when given labeled data. With just 100 labeled points to estimate the rule accuracies and class frequencies, we observed results for BF that are promising. A natural open question is whether one can reduce the dependence of a method like BF on labeled data.

We also saw two scenarios involving BF's consistency in the experiments. First, we provided evidence of Theorem 6 in the last row of Table 2. That theorem shows how to compute $g^*$, the best approximator to $\eta$. As predicted, $d(\eta, g^*)$ was shown to be smaller than $d(\eta, g^{bf})$ for every dataset. Not only that, having $g^*$ allowed the discussion of BF and OCDS' approximation uncertainty on real datasets (via Lemmas 5, 10). Second, we saw how $g^* \to \eta$ as $n \to \infty$ in a setting with an underlying generative model. I.e. BF was not too pessimistic in a favorable scenario.

To conclude, we want to touch on model and approximation uncertainty in other WS methods, especially for the purposes of introducing clarity. For example, although BF and DP [Ratner et al., 2016] receive the same rules-of-thumb, DP can have lower model uncertainty because it considers other factors, expanding its expressivity. Moreover, the interplay between the two types of uncertainty is unclear: Rühling Cachay et al. [2021a] showed that adding too many factors for DP caused the quality of the resulting prediction to decrease, i.e. the extra approximation uncertainty overshadowed the drop in model uncertainty. These notions of uncertainty come into play when one is allowed to add rules-of-thumb (e.g. [Varma and Ré, 2018]) or acquire labeled data to better estimate model parameters. For example, the approximation uncertainty of BF with 100 labeled points on IMDB, OBS (Table 2) is very low. Thus, the main source of loss is high model uncertainty, i.e. $g^*$ far from $\eta$, so one shouldn't acquire more labeled data to lower approximation uncertainty.

### Acknowledgements

We thank the National Science Foundation for support under grant IIS-2211386, Santiago Mazuelas for bringing the connection to logistic regression to our attention, Verónica Álvarez for helping to curate the datasets used, and the anonymous reviewers for their time and their suggestions, which have undoubtedly improved the paper.

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

# Convergence Behavior of an Adversarial Weak Supervision Method (Appendix)

**Steven An**[1]          **Sanjoy Dasgupta**[1]

[1]Computer Science Department,

University of California, San Diego,

La Jolla, CA 92093, USA

## A    APPENDIX OVERVIEW

Here, we provide the missing proofs along with the remaining experimental results. We'll recap the Balsubramani-Freund (BF) model (with log loss) and the Dawid-Skene (DS) model (specifically the one-coin variant).

After the model recap, the order of results is as follows.

- Section C, statement and proof of Theorems 1 and 2, derivations of the optimal learner play and adversary labeling.

- Section D, discussion on the set of BF predictions.

- Section E, statement and proof of Lemma 3, BF's relationship to logistic regression.

- Section F, statement and proof of Lemma 4, a Pythagorean theorem.

- Section G, a discussion of model and approximation uncertainty.

- Section H, statement and proof of Lemma 5, BF's loss decomposition.

- Section I, statement and proof of an error bound for BF, with major steps fleshed out in subsequent subsections. Also the proof of Theorem 7.

  - Subsection I.1, exhibition of reference BF program, a prerequisite to sensitivity analysis result.
  - Subsection I.2, relevant background and proof of sensitivity analysis result used.
  - Subsection I.3, simplification of terms in sensitivity analysis result.

- Section J, statement and proof of Theorem 6, BF's consistency. I.e. the learner's prediction being the best approximator for $\eta$ when $\epsilon = \vec{0}_m$.

- Section K, statement and proof of Lemma 9, showing that the exponential family $\mathcal{G}$ only contains DS predictions which doesn't use parameters that are $0$ or $1$.

- Section L, statement and proof of Lemma 10, the DS loss decomposition.

- Section M, statement and proof of Lemma 12.

- Section N, construction and analysis of a set of problems where DS is in general inconsistent (Lemma 11).

- Section O, the complete experimental results.

## B    MODEL RECAP

In this section, we recap the two models we consider.

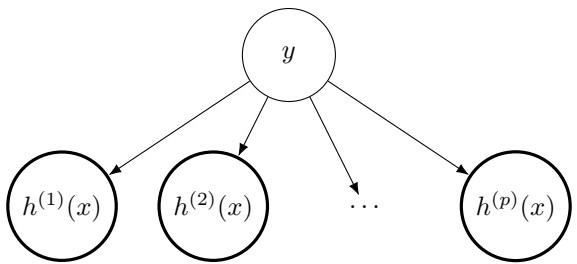

Figure 5: Dawid-Skene Graphical Model

**Input** : Predictions and unlabeled points
**Output** : Posterior distributions $g \in \Delta_k^n$
Initialize $g$ using simple majority vote;
**while** *g has not converged* **do**
    // Maximization Step
    **for** $j \in [p]$ **do**
        $b_j \leftarrow \frac{1}{n_j} \sum_{i=1}^{n} \sum_{\ell=1}^{k} g_{i\ell} \mathbf{1}(h^{(j)}(x_i) = \ell)$;
    **end**
    **for** $\ell \in [k]$ **do**
        $w_\ell \leftarrow \sum_{i=1}^{n} g_{i\ell}/n$;
    **end**
    // Expectation Step
    **for** $i \in [n], \ell \in [k]$ **do**
        $\widehat{g}_{i\ell} \leftarrow w_\ell \prod_{j=1}^{p} (b_j)^{\mathbf{1}(h^{(j)}(x_i)=\ell)} \left( \frac{1-b_j}{k-1} \right)^{\mathbf{1}(h^{(j)}(x_i)\neq\ell)}$;
    **end**
    **for** $i \in [n], \ell \in [k]$ **do**
        $g_{i\ell} \leftarrow \widehat{g}_{i\ell}/(\sum_{\ell'=1}^{k} \widehat{g}_{i\ell'})$;
    **end**
**end**
**return g**;

**Algorithm 1:** One Coin Dawid-Skene EM

## B.1 NOTATION RECAP

Before going forward, we quickly review the notation. We have $n$ datapoints $X = \{x_1, \ldots, x_n\} \subset \mathcal{X}^n$ with labels in $\mathcal{Y} = [k]$. There are $p$ rules of thumb in the ensemble, the $j^{th}$ rule is denoted $h^{(j)} \colon X \to \mathcal{Y} \cup \{?\}$. The ? denotes abstention.

In general, we will be dealing with $n$ distributions, each over $k$ elements, denoted by the set $\Delta_k^n$. The ground truth $\eta$ is a vector in $\Delta_k^n$. $\eta_i \in \Delta_k$ is a distribution over $k$ elements where

$$\eta_i = (\eta_{i1}, \ldots, \eta_{ik}) \qquad \text{where} \qquad \eta_{i\ell} = \Pr(y = \ell \mid x_i) \qquad \text{so that} \qquad \eta = (\eta_1, \ldots, \eta_n).$$

We will also consider the empirical distribution $\Pr_n(\cdot)$, which assigns probability mass $1/n$ to every datapoint $x_i$.

## B.2 DS MODEL

To estimate the probability of labels, the DS model posits an underlying generative model. By fitting the parameters of this model, one is able to obtain estimates of label probabilities for every datapoint. The model assumed is simple: for a datapoint and its label $(x, y) \in \mathcal{X} \times \mathcal{Y}$, rule predictions $h^{(1)}(x), \ldots, h^{(p)}(x)$ are conditionally independent given label $y$ (see Figure 1).

The parameters of the DS model are the underlying class frequencies $\Pr(y = \ell)$ and $p$ confusion matrices. Each rule $h^{(j)}$ is

parameterized by a row stochastic confusion matrix $B_j = b_{j\ell\ell'} \in [0,1]^{k\times k}$ where

$$b_{j\ell\ell'} = \Pr(h^{(j)}(x) = \ell' \mid y = \ell).$$

Since the rule predictions are conditionally independent given the label, the true posterior probability of the label is

$$\Pr(y \mid h^{(j)}(x), j \in [p]) = \frac{\Pr\left(y, h^{(j)}(x), j \in [p]\right)}{\Pr\left(h^{(j)}(x), j \in [p]\right)} = \frac{\Pr\left(y\right)\prod_{j=1}^{p}\Pr\left(h^{(j)}(x) \mid y\right)}{\sum_{\ell=1}^{k}\Pr\left(y = \ell\right)\prod_{j=1}^{p}\Pr\left(h^{(j)}(x) \mid y = \ell\right)}. \qquad (8)$$

In practice, one estimates the confusion matrix entries and class frequencies, e.g. by EM, and plugs them in the right hand side in lieu of their empirical counterparts.

For the one-coin DS model (OCDS), each rule is a biased coin. Whether or not it makes the correct prediction is independent of the class and is the result of a coin flip. Call the bias (or accuracy) for rule $h^{(j)}$ $b_j$. The confusion matrix can be defined as follows.

$$b_{j\ell\ell'} = \begin{cases} b_j & \text{if} \quad \ell = \ell' \\ \frac{1-b_j}{k-1} & \text{otherwise.} \end{cases}$$

We present the EM algorithm for the OCDS model in Algorithm 1. Recall that $n_j$ is the number of predictions rule $h^{(j)}$ makes on $X$.

## B.3 BF MODEL

If we could upper and lower bound the relevant parameters (class frequencies, accuracies) from the DS model, we can shrink the possible labelings for the datapoints the adversary can choose. For element-wise inequality, we mean the adversary is restricted to

$$P = \{z \in \Delta_k^n : b - \epsilon \le Az \le b + \epsilon\}$$

where $b$ is our estimate for $b^* := A\eta$ and $\epsilon$ so large that $b_j^* \in [b_j - \epsilon_j, b_j + \epsilon_j]$ for each $j$. While we consider $A$ as defined in the main paper, this could be generalized. Then, the learner and adversary can play a zero-sum minimax game, where the learner aims to maximize log-likelihood, while the adversary seeks to minimize it. That is,

$$V = \max_{g \in \Delta_k^n} \min_{z \in P} z \cdot \log g.$$

Our goal is to upper bound $-V$, and with such a bound we can prove our claimed results.

For completeness, we restate the definition of $A$. Write each rule's prediction $h^{(j)}(x_i)$ as a vector in $\{0,1\}^k$, as follows:

$$h_i^{(j)} = \begin{cases} \vec{e}_\ell & \text{if } h^{(j)}(x_i) = \ell \in [k] \\ \vec{0}_k & \text{if } h^{(j)}(x_i) = ? \end{cases}$$

Here $\vec{e}_\ell$ is the $k$-dimensional coordinate vector that is 0 except for a 1 at position $\ell$, and $\vec{0}_k$ is the $k$-dimensional zero vector. We write all of $h^{(j)}$'s predictions in vector form (abusing notation),

$$h^{(j)} = (h_1^{(j)}, \ldots, h_n^{(j)}) \in \Delta_k^n.$$

Thus $h_{i\ell}^{(j)}$ is 1 if $h^{(j)}(x_i) = \ell$ and 0 otherwise. Let $n_j$ denote the number of points (out of $n$ possible) where rule $h^{(j)}$ makes a prediction.

Letting $a^{(j)}$ be the $j^{th}$ row of matrix $A \in \mathbb{R}^{m \times kn}$ where $m = p + k$,

$$a^{(j)} = \begin{cases} h^{(j)}/n_j & \text{when} \quad 1 \le j \le p \\ \vec{e}_{j-p}^n/n & \text{when} \quad p+1 \le j \le p + k = m. \end{cases}$$

$\vec{e}_\ell^n$ is the $\ell^{th}$ canonical basis vector in $k$ dimensions, repeated $n$ times.

# C  PROOF OF THEOREMS 1 AND 2

We prove Theorem 1 by finding the dual of the two problems mentioned. Theorem 2 immediately follows from our derivations in service of Theorem 1.

**Theorem 13** (Theorem 1). *The minimax game in Equation 5 can equivalently written as follows.*

$$V = \max_{g \in \Delta_k^n} \min_{z \in P} \; z \cdot \log g = \min_{z \in P} \max_{g \in \Delta_k^n} \; z \cdot \log g = \min_{z \in P} z \cdot \log z$$

*The first expression defines a learner prediction $g^{bf}$, the second defines an adversarial labeling $z^*$. Then, $g^{bf} = z^*$ and they are the maximum entropy distribution in $P$, the optimal solution to the right-most expression.*

*Proof.* The first equality is by definition. For the second equality, Von Neumann's minimax theorem [Nikaidô, 1954] allows the commutation of the max and min for the second equality – the objective is concave in $g$ when $z$ is fixed, convex in $z$ when $g$ fixed, and the respective sets containing $g, z$ are convex and compact. The third equality follows from Gibb's inequality: for the prediction/label distribution of datapoint $x_i$, the objective is $\sum_{\ell=1}^{k} z_{i\ell} \log g_{i\ell}$, which is maximized exactly when $g_{i\ell}$ equals $z_{i\ell}$ for all $\ell \in [k]$.

To show that the learner and adversary's labelings are equal at optimality, we derive their forms in terms of Lagrange multipliers, and show that the dual problems where the Lagrange multipliers come from are one and the same. To figure out what $g^{bf}$ is, we find the dual of $\min_{z \in P} z \cdot \log g$ and then perform the maximization over $g$. This is done in Lemma 14. In going from the second to third expression in the claim, we have already performed the inner maximization over $g$. It suffices to find the dual of the sum of max entropies problem. This is done in Lemma 15. One sees that the functional form and dual problem are the same. □

**Lemma 14.** *The learner's optimal prediction $g$ for game*

$$V = \max_{g \in \Delta_k^n} \min_{z \in P} z \cdot \log g \qquad \text{is} \qquad g_{i\ell} = \frac{\exp(a_{i\ell}^{(\sigma' - \sigma)})}{\sum_{\ell'=1}^{k} \exp(a_{i\ell'}^{(\sigma' - \sigma)})}.$$

$\sigma, \sigma'$ *are gotten from optimizing the dual problem*

$$V = V(b, \epsilon) = \max_{\sigma, \sigma' \geq \vec{0}_m} \left[ (\sigma' - \sigma) \cdot b - (\sigma' + \sigma) \cdot \epsilon - \sum_{i=1}^{n} \log \left( \sum_{\ell=1}^{k} \exp(a_{i\ell}^{(\sigma' - \sigma)}) \right) \right].$$

*Proof.* We find the Lagrange dual of the inner minimization problem first, then do the outer maximization over $g$. Let's associate Lagrange multiplier $\sigma$ with $Az \leq b + \epsilon$ and $\sigma'$ with $-Az \leq -b + \epsilon$. Observe that

$$\begin{aligned}
\min_{z \in P} z \cdot \log g &= \max_{\sigma, \sigma' \geq \vec{0}_m} \min_{z \in \Delta_k^n} \left[ z \cdot \log g + \sigma \cdot (Az - b - \epsilon) + \sigma' \cdot (-Az + b - \epsilon) \right] \\
&= \max_{\sigma, \sigma' \geq \vec{0}_m} \min_{z \in \Delta_k^n} \left[ z \cdot (\log g - A^\top \sigma' + A^\top \sigma) + \sigma \cdot (-b - \epsilon) + \sigma' \cdot (b - \epsilon) \right] \\
&= \max_{\sigma, \sigma' \geq \vec{0}_m} \left[ \sum_{i=1}^{n} \min(\log g_i - a_i^{(\sigma' - \sigma)}) + (\sigma' - \sigma)^\top b - (\sigma' + \sigma)^\top \epsilon \right]
\end{aligned}$$

where the minimum and logarithm functions each act element-wise. This means that

$$\max_{g \in \Delta_k^n} \min_{z \in P} z \cdot \log g = \max_{g \in \Delta_k^n} \max_{\sigma, \sigma' \geq \vec{0}_m} \left[ \sum_{i=1}^{n} \min(\log g_i - a_i^{(\sigma' - \sigma)}) + (\sigma' - \sigma)^\top b - (\sigma' + \sigma)^\top \epsilon \right].$$

We can commute the maximums, and then consider how to maximize just one of the summands. That is sufficient because each of the summands is one out of the $n$ distributions in $g$. For simplicity, write $a_i^{(\sigma' - \sigma)}$ as $a_i$.

We claim that the optimal $g$ is such that $\log g_i - a_i = M_i \mathbf{1}_k$ for some $M_i \in \mathbb{R}$, i.e. it has the same value each of the $k$ positions. We'll show this by contradiction. Suppose $g_i'$ is optimal and is such that $\log g_i' - a_i$ has different values at different positions. The goal is to maximize the minimum element of vector $f := \log g_i' - a_i$. We'll show that if not all elements of that vector are equal, then it's possible to increase the minimum.

Without loss of generality, suppose that $f_\ell, \ell \in [k]$ is a non-decreasing sequence with the added condition that $f_1 < f_2$. The case where the minimum element of $f$ appearing multiple times is an easy generalization of the below argument. Define $\xi = 0.5(f_2 - f_1)$, which is strictly positive by assumption. Now, for $\ell' \geq 2$, define $\gamma_{\ell'}$ as the strictly positive value such that

$$\log\left(g'_{i\ell'} - \gamma_{\ell'}\right) - a_{i\ell'} = f_2 - \xi = \frac{1}{2}(f_2 + f_1) > f_1.$$

If we create a new distribution $g''_{i\ell'} = g_{i\ell'} - \gamma_{\ell'}$ (for the same $\ell'$ as above), we'll have $\sum_{\ell'=2}^{k} \gamma_{\ell'}$ extra probability mass. Thus, if we give that extra mass to the first position, we have

$$\log\left(g'_{i1} + \sum_{\ell'=1}^{k} \gamma_{\ell'}\right) - a_{i1} > \log(g'_{i\ell}) - a_{i1} = f_1.$$

Fully written out, the new distribution $g''_i \in \Delta_k$ is defined as follows:

$$g''_{i\ell} = \begin{cases} g'_{i\ell} - \gamma_\ell & \text{for} \quad \ell \geq 2 \\ g'_{i\ell} + \sum_{\ell'=1}^{k} \gamma_{\ell'} & \text{otherwise.} \end{cases}$$

From our previous argument, we know that for all $\ell \in [k]$,

$$\log(g''_{i\ell}) - a_{i\ell} > f_1$$

meaning $g'_i$ was not the optimal choice of distribution.

Now, we solve for $M_i$. For every $\ell \in [k]$,

$$\log g_i - a_i = M_i \vec{1}_k \qquad \text{implies} \qquad g_{i\ell} = \exp(M_i + a_{i\ell}).$$

Now since $g_i$ is a distribution over $k$ elements,

$$\sum_{\ell=1}^{k} g_{i\ell} = \sum_{\ell=1}^{k} \exp(M_i + a_{i\ell}) = 1 \quad \Rightarrow \quad M_i = -\log\left(\sum_{\ell=1}^{k} \exp(a_{i\ell})\right)$$

We are able to immediately derive our claims. First, lets plug $M_i$ back into our expression for $g_{i\ell}$. We have

$$g_{i\ell} = \frac{\exp(a_{i\ell}^{(\sigma'-\sigma)})}{\sum_{\ell'=1}^{k} \exp(a_{i\ell'}^{(\sigma'-\sigma)})}.$$

Now, by substituting $M_i$ for $\min(\log g_i - a_i)$, the maximization problem is

$$\max_{\sigma,\sigma' \geq \vec{0}_m} \left[(\sigma' - \sigma)^\top b - (\sigma' + \sigma)^\top \epsilon - \sum_{i=1}^{n} \log\left(\sum_{\ell=1}^{k} \exp(a_{i\ell}^{(\sigma'-\sigma)})\right)\right].$$

$\square$

**Lemma 15.** *The adversary's optimal labeling $z$ for the game*

$$V = \min_{z \in P} \max_{g \in \Delta_k^n} z \cdot \log g \qquad is \qquad z_{i\ell} = \frac{\exp(a_{i\ell}^{(\sigma'-\sigma)})}{\sum_{\ell'} \exp(a_{i\ell'}^{(\sigma'-\sigma)})}.$$

*$\sigma, \sigma'$ are gotten from optimizing the dual problem*

$$V = V(b, \epsilon) = \max_{\sigma,\sigma' \geq \vec{0}_m} \left[(\sigma'^\top - \sigma)^\top b - (\sigma' + \sigma)^\top \epsilon - \sum_{i=1}^{n} \log\left(\sum_{\ell=1}^{k} \exp(a_{i\ell}^{(\sigma'-\sigma)})\right)\right].$$

*Proof.* We have argued in the proof of Theorem 1 that the inner maximum to our problem can be dispensed with. In this case, $g = z$. Thus, we just need to find the Lagrange dual of

$$\min_{z \in P} \; z \cdot \log z \quad \text{where} \quad P = \{z \in \Delta_k^n : b - \epsilon \leq Az \leq b + \epsilon\}.$$

Define matrix $D \in \{0,1\}^{n \times nk}$ which is the block identity matrix in $n$ dimensions, except rather than having a scalar in each entry, one has a row vector of length $k$ (that's either all 0's or 1's). Namely, $Dz = \vec{1}_n$, the vector of $n$ 1's – $D$ adds up blocks of $k$ elements of the vector $z$, which are defined to be distributions. The Lagrangian is then

$$L(z, \sigma, \sigma', \xi) := z \cdot \log z + \sigma \cdot (Az - b - \epsilon) + \sigma' \cdot (-Az + b - \epsilon) + \xi \cdot (Dz - \vec{1}_n).$$

$\sigma, \sigma' \geq 0$ are for the constraints $Az \leq b + \epsilon$ and $b - \epsilon \leq Az$ respectively, while $\xi \in \mathbb{R}^n$ is for the constraint that each block of $k$ elements in $z$ adds to 1 ($Dz = \vec{1}_n$). There is no need for a constraint on the non-negativity of $z$ as the functional form of $z$ is always non-negative.

To find the Langrange dual, we need to minimize over $z$. If we gather all the $z$'s together, we get

$$\min_{z \in \mathbb{R}^{nk}} \left[ z \cdot \left( \log z + A^\top \sigma - A^\top \sigma' + D^\top \xi \right) + \sigma \cdot (-b - \epsilon) + \sigma' \cdot (b - \epsilon) - \xi \cdot \vec{1}_n \right].$$

Differentiating with respect to $z$ gives

$$\log z + \vec{1}_{nk} + A^\top \sigma - A^\top \sigma' + D^\top \xi.$$

Setting to $\vec{0}_{nk}$ gives the optimal $z^*$, which we write with exponential function acting element-wise:

$$z^* = \exp(-\vec{1}_{nk} - A^\top \sigma + A^\top \sigma' - D^\top \xi) = \exp(-\vec{1}_{nk} + a^{(\sigma' - \sigma)} - D^\top \xi)$$

where we used $a^{(\theta)} = A^\top \theta$ after grouping to get the term $A^\top (\sigma' - \sigma)$. Plugging $z^*$ in for $z$, we can write the Lagrangian as

$$L(z^*, \sigma, \sigma', \xi) = -\exp(-\vec{1}_{nk} + a^{(\sigma' - \sigma)} - D^\top \xi) \cdot \vec{1}_{nk} + \sigma \cdot (-b - \epsilon) + \sigma' \cdot (b - \epsilon) - \xi \cdot \vec{1}_n$$

In the dual program, we will need to maximize over all Lagrange multipliers. Indeed, we can maximize over $\xi$ analytically. We will solve for $\xi_1$, but the argument will work for an arbitrary element of $\xi$. First, note that $D^\top \xi$ is a vector of length $nk$ such that the first $k$ elements are $\xi_1$, the next $k$ elements are $\xi_2$, etc. Therefore, in the first $k$ elements of the vector of exponential values, $D^\top \xi$ simplifies to $\xi_1$. Also, note that in the way that our notation is defined, $a_1^{(\theta)}$ is the length $k$ vector denoting the first $k$ elements of the vector $a^{(\theta)}$. Observe then that

$$\frac{\partial}{\partial \xi_1} L(z^*, \sigma, \sigma', \xi) = \sum_{\ell=1}^{k} \exp(-1 + a_{1\ell}^{(\sigma' - \sigma)} - \xi_1) - 1.$$

Setting this equal to 0, one can see that

$$\xi_1^* = -1 + \log \left( \sum_{\ell=1}^{k} \exp \left( a_{1\ell}^{(\sigma' - \sigma)} \right) \right).$$

From what we have said, we can plug in $\xi_1^*$ to get the form of $z_1^*$ in terms of $\sigma$ and $\sigma'$.

$$z_1^* = \exp \left( -\vec{1}_k + a_1^{(\sigma' - \sigma)} - \xi_1 \vec{1}_k \right) = \frac{\exp \left( a_1^{(\sigma' - \sigma)} \right)}{\sum_{\ell=1}^{k} \exp \left( a_{1\ell}^{(\sigma' - \sigma)} \right)}$$

where the division is element-wise. This gives us our result on the prediction form of the learner, as we know its prediction matches the adversary's labeling.

If we now look at the Lagrangian (after plugging in the optimal $\xi$), we see that the first term evaluates to $-n$ because we are summing over $n$ softmaxes, i.e.

$$-\exp(-\vec{1}_{nk} + a^{(\sigma' - \sigma)} - D^\top \xi) \cdot \vec{1}_{nk} = -\sum_{i=1}^{n} \sum_{\ell=1}^{k} \frac{\exp \left( a_{i\ell}^{(\sigma' - \sigma)} \right)}{\sum_{\ell'=1}^{k} \exp \left( a_{i\ell'}^{(\sigma' - \sigma)} \right)} = -\sum_{i=1}^{n} 1.$$

One easily sees that

$$-\sum_{i=1}^{n} \xi_i = -\sum_{i=1}^{n} \left[ -1 + \log \left( \sum_{\ell=1}^{k} \exp \left( a_{i\ell}^{(\sigma'-\sigma)} \right) \right) \right] = n - \sum_{i=1}^{n} \log \left( \sum_{\ell=1}^{k} \exp \left( a_{i\ell}^{(\sigma'-\sigma)} \right) \right).$$

This means that our dual function is

$$\left[ \sigma \cdot (-b - \epsilon) + \sigma' \cdot (b - \epsilon) - \sum_{i=1}^{n} \log \left( \sum_{\ell=1}^{k} \exp \left( a_{i\ell}^{(\sigma'-\sigma)} \right) \right) \right].$$

Maximizing over $\sigma, \sigma'$ both being non-negative in every element gives us our claim. $\qquad\square$

## D   EXPONENTIAL FAMILY OF BF PREDICTIONS

The matrix $A \in \mathbb{R}^{m \times nk}$ used in the BF program defines a family of conditional probability distributions parameterized by $\theta \in \mathbb{R}^m$, a vector of real weights. We write such a distribution as

$$g_{i\ell}^{(\theta)} \propto \exp \left( \sum_{j=1}^{m} \theta_j a_{i\ell}^{(j)} \right) = \exp(a_{i\ell}^{(\theta)}).$$

The set of these predictions where $\theta$ is allowed to range is

$$\mathcal{G}_{bf} := \{ g^{(\theta)} : \theta \in \mathbb{R}^m \},$$

which is an exponential family of distributions. (Note that $g^{(\theta)} \in \Delta_k^n$.) Indeed, the BF predictions from Theorem 2 take this form – just take $\theta = \sigma' - \sigma$.

## E   BF AND LOGISTIC REGRESSION (LEMMA 3)

Now, we turn to showing the relationship between BF and logistic regression.

We consider multiclass logistic regression for $k$ classes trained on $X$, the dataset of $n$ points. Say that each datapoint $x_i \in X$ has $d$ dimensions, i.e. $x_i \in \mathbb{R}^d$. Rather than have hard labels for points in $X$, we will allow the labels to be probability distributions, $\eta_i \in \Delta_k$ specifically. To estimate $\eta_i$, each class will get a score and the softmax of those scores will be taken. To get class $\ell$'s score, the datapoint features $x_i$ will be weighted by vector $w_\ell \in \mathbb{R}^d$ and summed together via $w_\ell^\top x_i$. (We're abusing notation here because $w_\ell$ will later represent a class frequency). The logistic regression prediction for datapoint $x_i$ and class $\ell$ is

$$g_{i\ell}^{lr} = \frac{\exp(w_\ell^\top x_i)}{\sum_{\ell'=1}^{k} \exp(w_{\ell'}^\top x_i)} \quad \text{which estimates} \quad \eta_{i\ell}.$$

In vector form, the class scores are $W x_i$, where $W$ is a $k \times d$ matrix, row $\ell$ being $w_\ell$. In words, the logistic regression prediction form is the softmax of a weight matrix times feature vector. Recalling that logistic regression minimizes cross entropy loss, the optimization problem is

$$\min_{W \in \mathbb{R}^{k \times d}} \eta^\top \log g^{lr}$$

To relate BF to logistic regression, we need to do two things. First is to convert the formulation of the BF constraints so that the prediction form matches the logistic regression form. Second is to show that the dual form of the BF problem can be converted into regularized cross entropy. At the end of this section we will show how to convert an instance of a multi-class logistic regression problem into a BF problem.

Recall that $A$ is matrix in $\mathbb{R}^{m \times nk}$. Say $A_i$ is the $m \times k$ matrix for datapoint $x_i$, consisting of columns $k(i-1) + 1$ to $ki$. The $k$ scores that BF gives to datapoint $i$ with weights $\theta$ is $A_i^\top \theta$. We want to exhibit weight matrix $T_\theta$ and datapoint features $\widehat{x}_i$ such that $A_i^\top \theta = T_\theta \widehat{x}_i$.

The idea is to flatten $A_i$ into a vector while inflating $\theta$ into a matrix of the correct size. $\widehat{x}_i$ will take its elements from $A_i$. Specifically $\widehat{x}_i \in \mathbb{R}^{mk}$ will be $A_i$ flattened in row major order. Each block of $k$ elements in $\widehat{x}_i$ is a row of $A_i$. (We write $a_{i,j}$ to mean the element of $A$ at row $i$, column $j$ – no relation to $i, j$ in the rest of the appendix.)

$$\widehat{x}_i = \left(a_{1,k(i-1)+1}, a_{1,k(i-1)+2}, \ldots, a_{1,ki}, \ldots, a_{m,k(i-1)+1}, \ldots, a_{m,ki}\right)^\top$$

For the weights in matrix form, let $I_k$ be the identity matrix in $k$ dimensions. Call our weight matrix $T_\theta \in \mathbb{R}^{k \times mk}$ where

$$T_\theta = (\theta_1 I_k, \theta_2 I_k, \ldots, \theta_m I_k).$$

**Lemma 16.** *For any datapoint index $i$ and fixed weight vector $\theta$, define $\widehat{x}_i$ and $T_\theta$ as above.*

$$A_i^\top \theta = T_\theta \widehat{x}_i.$$

*This means that the BF prediction can be written as the softmax of a weight matrix times feature vector, just like in logistic regression.*

*Proof.* Say $a_{ij}$ is the $(i, j)^{th}$ element of matrix $A$. Then,

$$(A_i^\top \theta)_\ell = \sum_{j=1}^m \theta_j a_{j,k(i-1)+\ell}.$$

The new prediction for class $\ell$ on datapoint $i$ is

$$(T_\theta \widehat{x}_i)_\ell = \sum_{j=1}^m \sum_{\ell'=1}^k \mathbf{1}(\ell' = \ell)\theta_j a_{j,k(i-1)+\ell'} = \sum_{j=1}^m \theta_j a_{j,k(i-1)+\ell}$$

which matches our expression above. This means that $A_i^\top \theta = T_\theta \widehat{x}_i$. So,

$$g_{i\ell}^{(\theta)} = \frac{\exp(a_{i\ell}^{(\theta)})}{\sum_{\ell'=1}^k \exp(a_{i\ell'}^{(\theta)})} = \frac{\exp((T_\theta \widehat{x}_i)_\ell)}{\sum_{\ell'=1}^k \exp((T_\theta \widehat{x}_i)_{\ell'})}$$

$\square$

Now, we can rewrite the BF dual program so it looks like $\ell_1$ regularized logistic regression. The regularization is different in that there isn't one regularization constant, but twice the number of weights being learned. Moreover, we learn $m$ weights for $mk$ features, whereas regular multi-class logistic regression would learn $mk^2$ weights for the same amount of features.

**Lemma 17** (Lemma 3)**.** *Fix $A, \eta, b, \epsilon$ so that $b^* = A\eta$ and the BF prediction $g^{bf}$ from Lemma 14 are fixed. The BF dual from that problem is equivalent to $\ell_1$ regularized multi-class logistic regression where each weight gets two regularization coefficients, depending on its sign. Let $\theta \in \mathbb{R}^m$ and $\sigma'_j$, $\sigma_j$ be the positive and negative parts of $\theta_j$ respectively.*

$$-V(b, \epsilon) = \min_{\theta \in \mathbb{R}^m} \left[ \underbrace{-\eta^\top \log g^{(\theta)}}_{\text{cross entropy}} + \underbrace{\sigma'^\top (b^* - (b - \epsilon)) + \sigma^\top (b - b^* + \epsilon)}_{\text{regularization}} \right].$$

*Moreover, the optimal $g^{(\theta)}$ from above equals $g^{bf}$ and can be written as a logistic regression style prediction.*

*Proof.* We start by rewriting the BF program for fixed $A, b, \epsilon, \eta$ (Theorem 2).

$$V(b, \epsilon) = \max_{\sigma, \sigma' \geq \vec{0}_m} \left[ (\sigma' - \sigma) \cdot b - (\sigma' + \sigma) \cdot \epsilon - \sum_{i=1}^n \log \left( \sum_{\ell=1}^k \exp(a_{i\ell}^{(\sigma'-\sigma)}) \right) \right].$$

We will need $b^* = A\eta$, so we rewrite the (dual) function as

$$(\sigma' - \sigma)^\top b^* - \sigma'^\top (b^* - b + \epsilon) - \sigma^\top (b - b^* + \epsilon) - \sum_{i=1}^n \log \left( \sum_{\ell=1}^k \exp(a_{i\ell}^{(\sigma'-\sigma)}) \right)$$

If we take $\theta$ to mean $\sigma' - \sigma$, we recall this statement from the proof of Lemma 4:

$$\log g_{i\ell}^{(\theta)} = \log \frac{\exp(a_{i\ell}^{(\theta)})}{\sum_{\ell'=1}^{k} \exp(a_{i\ell'}^{(\theta)})}$$

By using $A\eta = b^*$ and $\theta = \sigma' - \sigma$, the BF dual function can be written as

$$\eta^\top A^\top \theta - \sum_{i=1}^{n} \log \left( \sum_{\ell=1}^{k} \exp(a_{i\ell}^{(\theta)}) \right) - \sigma'^\top (b^* - b + \epsilon) - \sigma^\top (b - b^* + \epsilon)$$

which becomes

$$\sum_{i=1}^{n} \sum_{\ell=1}^{k} \eta_{i\ell} \left( \log(\exp(a_{i\ell}^{(\theta)})) - \log \left( \sum_{\ell'=1}^{k} \exp((A^\top \theta)_{i\ell'}) \right) \right) - \sigma'^\top (b^* - b + \epsilon) - \sigma^\top (b - b^* + \epsilon)$$

because $\sum_{\ell=1}^{k} \eta_{i\ell} = 1$. Observe the term inside the outermost parentheses is just $\log g_{i\ell}^{(\theta)}$. Reintroducing the maximization,

$$V(b, \epsilon) = \max_{\theta \in \mathbb{R}^m} \left[ \eta^\top \log g^{(\theta)} - \sigma'^\top (b^* - b + \epsilon) - \sigma^\top (b - b^* + \epsilon) \right]$$

which is

$$-V(b, \epsilon) = \min_{\theta \in \mathbb{R}^m} \left[ -\eta^\top \log g^{(\theta)} + \sigma'^\top (b^* - b + \epsilon) + \sigma^\top (b - b^* + \epsilon) \right].$$

We now want to argue that this is a logistic regression problem. Clearly, we're minimizing cross entropy with $\eta$. We argue that the last two terms are equivalent to $\ell_1$ regularization. The usual regularization term would look like $C\|\theta\|_1$ for some $C \geq 0$. This is equal to $\sum_{j=1}^{m} C|\theta_j| = \sum_{j=1}^{m} C(\sigma'_j + \sigma_j)$, a weighted sum of non-negative values. This is because $|\theta_j| = \sigma'_j + \sigma_j$ as $\sigma'_j$ is the positive part of $\theta_j$ while $\sigma_j$ is the negative part. The construction of $P$ assumes $b^* \in [b - \epsilon, b + \epsilon]$ so that one can check that the coefficients for $\sigma', \sigma$ are non-negative. Therefore, the last two terms in the stated objective act like $\ell_1$ regularization. To finish, we use the fact that the BF prediction $g^{(\theta)}$ can be written in the logistic regression style, Lemma 16. □

We can also go from a $\ell_1$ regularized logistic regression loss to a BF problem. The matrix $A$ will be defined in terms of the datapoint features while $b$ will be an expected value of the features. This means that $b$ will not necessarily lie in the unit interval. This specification of constraints is very similar to the "uncertainty sets" proposed by Mazuelas et al. [2020].

**Lemma 18.** *Suppose one's goal was to minimize a $\ell_1$ regularized logistic regression loss function written as below. Then, the minimization of that loss function is equivalent to solving a BF problem.*

$$\min_{w_\ell \in \mathbb{R}^d, \ell \in [k]} \left[ -\sum_{i=1}^{n} \sum_{\ell=1}^{k} \eta_{i\ell} \log \frac{\exp(w_\ell^\top x_i)}{\sum_{\ell'=1}^{k} \exp(w_\ell'^\top x_i)} + C \sum_{\ell=1}^{k} \|w_\ell\|_1 \right]$$

*The constraint matrix $A$ will have size $dk \times nk$ where $d$ is the dimension of $x_i$. In block matrix form (where the blocks are scaled identity matrices), the block in row $c$ column $i$ of $A$ is $(x_i)_c I_k$. That is, the $c^{th}$ feature of datapoint $x_i$ times the identity matrix in $k$ dimensions. The constraints for BF are $z \in \Delta_k^n$ and*

$$b^* - C\vec{1}_{dk} \leq Az \leq b^* + C\vec{1}_{dk} \quad \text{where} \quad b^*_{k(c-1)+\ell} = \sum_{i=1}^{n} (x_i)_c \eta_{i\ell}.$$

*Proof.* To prove this, we essentially work backwards from the result above. We need to specify a matrix $A$ and we also need $\epsilon$.

We first go from the regularization coefficient $C$ to a choice of $b$ and $\epsilon$. Note that in the above proof, the regularization constants had either $b - b^*$ or $b^* - b$. These two values can be different. However, since we only have a single value $C$, we will choose $b = b^*$ and $\epsilon = C$.

We will need $A$ to be of size $dk \times nk$ because for multi-class logistic regression, one learns $dk$ weights. Like above, we define $A_i \in \mathbb{R}^{dk \times k}$, the part of the constraint matrix for datapoint $x_i$. (The $A_i$'s are concatenated horizontally to form $A$.) Then,

$$
A_i = \begin{bmatrix} (x_i)_1 I_k \\ (x_i)_2 I_k \\ \vdots \\ (x_i)_d I_k \end{bmatrix}.
$$

To see that this is correct, we write the logistic regression weights as a $dk$ length vector $w$.

$$
\widehat{w} = (w_{11}, w_{21}, \ldots, w_{k1}, w_{12}, w_{22}, \ldots, w_{k2}, \ldots, w_{1d}, \ldots, w_{kd})^\top
$$

Basically, the $k$ weight vectors are interleaved. For index $i$ of $\widehat{w}$, the element comes from the weight vector for class $i$ mod $k$. Concretely, $\widehat{w}_{k(c-1)+\ell} = w_{\ell c}$, the $c^{th}$ element of class $\ell$'s weight vector. We will now show that $(A_i^\top \widehat{w})_\ell = w_\ell^\top x_i$.

$$
(A_i^\top \widehat{w})_\ell = \sum_{c=1}^{d} \sum_{\ell'=1}^{k} \mathbf{1}(\ell' = \ell)(x_i)_c \widehat{w}_{k(c-1)+\ell'} = \sum_{c=1}^{d} (x_i)_c \widehat{w}_{k(c-1)+\ell}
$$

However, by virtue of the definition of $\widehat{w}$, we see that that last sum is exactly $w_\ell^\top x_i$.

To conclude the proof, we just need to specify $b^*$. $b^*$ will be equal to $A\eta$, so we compute element $k(c-1)+\ell$ of that vector.

$$
b^*_{k(c-1)+\ell} = (A\eta)_{k(c-1)+\ell} = \sum_{i=1}^{n} \sum_{\ell'=1}^{k} \mathbf{1}(\ell = \ell')(x_i)_c \eta_{i\ell'} = \sum_{i=1}^{n} (x_i)_c \eta_{i\ell}.
$$

$\square$

# F  PYTHAGOREAN THEOREM (LEMMA 4)

**Lemma 19** (Lemma 4).  *Pick any $\theta, \theta' \in \mathbb{R}^m$ and write $g = g^{(\theta)}$ and $g' = g^{(\theta')}$. If $Ag = A\eta$, then*

$$
d(\eta, g') = d(\eta, g) + d(g, g').
$$

*Proof.* We begin with

$$
d(\eta, g') - d(\eta, g) = \sum_{i=1}^{n} \sum_{\ell=1}^{k} \eta_{i\ell} \log \frac{\eta_{i\ell}}{g'_{i\ell}} - \sum_{i=1}^{n} \sum_{\ell=1}^{k} \eta_{i\ell} \log \frac{\eta_{i\ell}}{g_{i\ell}}
$$

$$
= \sum_{i=1}^{n} \sum_{\ell=1}^{k} \eta_{i\ell} \log \frac{g_{i\ell}}{g'_{i\ell}}.
$$

For $g = g^{(\theta)}$, we have

$$
\log g_{i\ell} = \log \frac{\exp(a_{i\ell}^{(\theta)})}{\sum_{\ell'=1}^{k} \exp(a_{i\ell'}^{(\theta)})} = a_{i\ell}^{(\theta)} - \log Z_i^{(\theta)},
$$

where $Z_i^{(\theta)}$ is a shorthand for $\sum_{\ell'=1}^{k} \exp(a_{i\ell'}^{(\theta)})$. Thus

$$
d(\eta, g') - d(\eta, g) = \sum_{i=1}^{n} \sum_{\ell=1}^{k} \eta_{i\ell} \left( a_{i\ell}^{(\theta)} - \log Z_i^{(\theta)} - a_{i\ell}^{(\theta')} + \log Z_i^{(\theta')} \right)
$$

$$
= \eta \cdot a^{(\theta)} - \eta \cdot a^{(\theta')} + \sum_{i=1}^{n} (\log Z_i^{(\theta')} - \log Z_i^{(\theta)})
$$

Now, by assumption $Ag = A\eta$. This means that for any $\theta \in \mathbb{R}^m$, $\theta \cdot Ag = \theta \cdot A\eta$. Written another way, for the same $\theta$, $\eta \cdot a^{(\theta)} = g \cdot a^{(\theta)}$, whereupon

$$d(\eta, g') - d(\eta, g) = g \cdot a^{(\theta)} - g \cdot a^{(\theta')} + \sum_{i=1}^{n} (\log Z_i^{(\theta')} - \log Z_i^{(\theta)})$$

$$= \sum_{i=1}^{n} \sum_{\ell=1}^{k} g_{i\ell} \left( a_{i\ell}^{(\theta)} - \log Z_i^{(\theta)} - a_{i\ell}^{(\theta')} + \log Z_i^{(\theta')} \right)$$

$$= \sum_{i=1}^{n} \sum_{\ell=1}^{k} g_{i\ell} \log \frac{g_{i\ell}}{g'_{i\ell}} = d(g, g').$$

$\square$

# G   MODEL AND APPROXIMATION UNCERTAINTY

For both BF and one-coin DS (with EM), their respective models use quantities that are estimated. For BF, it's given estimates of the rule accuracies and class frequencies, while for OCDS, EM estimates those same quantities. We would like to know how much of each method's error results from the fact that it is receiving estimated quantities in comparison to error that's unavoidable to the method. We will say the set of unlabeled datapoints $X$ is fixed, along with the rules $h^{(1)}, \ldots, h^{(p)}$, their predictions on $X$, and true label probabilities $\eta$.

Formally, abuse notation and say that all possible predictions a method can make lies in the set $\mathcal{G} \subseteq \Delta_k^n$. By choosing a method (read: by choosing BF/OCDS), one is choosing $\mathcal{G}$. Thus, the unavoidable error from choosing that method is $\min_{g \in \mathcal{G}} d(\eta, g)$. That quantity is referred to as the *model uncertainty* or $\mathcal{E}^{mod}$. Now, in general, the method doesn't output the best approximator from the set of possible predictions to $\eta$ due to a variety of factors. These could be how the method is initialized, how the hyperparameters are set, etc. Suppose that the best approximator to $\eta$ is unique. This will be what happens with our choice of $d(\cdot, \cdot)$ (Theorem 27). If the method predicts $g$, then the excess error from predicting $g$ over $g^* = \arg\min_{g \in \mathcal{G}} d(\eta, g)$ is $d(g^*, g)$. We call this the *approximation uncertainty* or $\mathcal{E}^{appr}$.

Taken together, these form a learner's *epistemic uncertainty*. Epistemic as in these sources of uncertainty are from our lack of knowledge. For model uncertainty, it would be sufficient to choose a method such that it can exactly predict $\eta$. Then, one would have no model uncertainty. Of course this is extremely non-trivial if you wish to choose a method with limited expressivity. Indeed, one could pick a method that was a universal approximator. For approximation uncertainty, one can in theory give the method the perfect hyperparameters, initialization, or stop it early. For example, the method might do a form of gradient descent and only stops after convergence. However, it may be optimal to take the method's prediction before convergence. In other words, one has approximation uncertainty when they do not know the best way to use the method to make it output $g^*$. If we call the epistemic uncertainty $\mathcal{E}$, the model uncertainty $\mathcal{E}^{mod}$, and the approximation uncertainty $\mathcal{E}^{appr}$, we have

$$\mathcal{E} = \underbrace{\mathcal{E}^{mod}}_{d(\eta, g^*)} + \underbrace{\mathcal{E}^{appr}}_{d(g^*, g)}.$$

Just predicting $\eta$ is not the end all. In the end, we are interested in the labels of points $x_1, \ldots, x_n$. However, if the true label for a datapoint is a distribution there is uncertainty on what the actual label is. For example, we may have some information about someone who smokes. Since not all smokers get cancer, there is uncertainty about whether they will get cancer. In that sense there is a distribution over the two outcomes representing whether they get cancer. This kind of uncertainty about the label is *aleatoric*.

Aleatoric uncertainty is not reducible and does not concern us. However, epistemic uncertainty is (in theory) reducible and we are concerned with this quantity. Indeed, our discussion here is in line with the ideas found in [Hüllermeier and Waegeman, 2021]. The main difference is that they suggest that the hypothesis space (our $\mathcal{G}$) should be fixed to be able to discuss aleatoric and epistemic uncertainty without ambiguity. However, we are allowing the hypothesis space ($\mathcal{G}$) to change with the model. So, if the reader is unsatisfied with how we use the terms "aleatoric", "epistemic", "model", "approximation" uncertainty, then they may refer to the mathematical quantities we have associated each of those terms with.

We now go through two examples and explain what the model and approximation uncertainties are. A method with all its epistemic uncertainty coming from the model uncertainty is majority vote. It only ever makes 1 prediction $g^{mv}$, i.e. $\mathcal{G}$ contains 1 point. Therefore, $g^{mv}$ is the best approximator for $\eta$. There is no approximation uncertainty because the method does not estimate anything. Now, suppose we considered weighted majority vote. For simplicity, say that none of the rules abstain and that the weights form a distribution. In other words, we take a convex combination of the rule predictions. Here, one can compute the optimal convex combination such that the resulting prediction $g^{wmv*}$ is closest to $\eta$. The distance between the two is the model uncertainty. However, one may not be able to compute the optimal combination. The approximation uncertainty is the distance from some weighted majority vote prediction $g^{wmv}$ to the optimal weighted majority vote prediction $g^{wmv*}$.

We now want to disambiguate and be very clear. Because we have chosen $d(\cdot, \cdot)$ to be KL divergence (of the sum thereof), the Pythagorean theorem we showed in Section F makes it so $\mathcal{E} = d(\eta, g)$ when $\mathcal{E}^{mod} = d(\eta, g^*)$ and $\mathcal{E}^{appr} = d(g^*, g)$. If one wants to choose another $d(\mu, \nu) : \Delta_k^n \times \Delta_k^n \to \mathbb{R}$, one may need to write

$$\mathcal{E}^{appr} = d(\eta, g) - d(\eta, g^*).$$

Now, because we decompose the error of BF and OCDS into model and approximation uncertainty, we are able to easily compare the two. We'll see that BF and OCDS have the same model uncertainty, meaning the comparison between the two models is fair. I.e. the unavoidable prediction loss for predictions from each model is the same. Contrast that to recent work in the literature with models that add rules-of-thumb in the process of creating labels for datapoints in $X$, e.g. Varma and Ré [2018]. There, the model uncertainty is being reduced and a naive comparison to a method that doesn't add rules-of-thumb would be unfair.

## H  BF LOSS DECOMPOSITION (LEMMA 5)

We're ready now to decompose the loss of any BF prediction. We'll show that the optimal approximator $g^*$ to $\eta$ satisfies the property needed to apply the Pythagorean theorem we just proved (Lemma 4) later.

**Lemma 20** (Lemma 5).
$$\underbrace{d(\eta, g^{bf})}_{\mathcal{E}_{bf}} = \underbrace{d(\eta, g^*)}_{\mathcal{E}_{bf}^{mod}} + \underbrace{d(g^*, g^{bf})}_{\mathcal{E}_{bf}^{appr}}.$$

*Proof.* By Corollary 28, we know that $A\eta = Ag^*$. So, in Lemma 4, choose $g'$ to be the BF prediction $g^{bf}$ and $g$ to be $g^*$. The result is immediate. □

## I  THE BF BOUND AND PROOF OF THEOREM 7

We now present an error bound for the BF program we have stated. This bound is used to show that BF dominates DS when $\epsilon = \vec{0}_m$, lets us compare BF and DS when $\epsilon$ is not all zeros, and gives rates of convergence for consistency. The proof is presented with just the major steps needed to get the bound. In the following sections, each of the major steps are proved.

**Theorem 21.** *Let $g^{bf}$ be the prediction from BF when it's given $P = \{z \in \Delta_k^n : b - \epsilon \leq Az \leq b + \epsilon\}$. Also, fix an arbitrary prediction $g^{ref} = g^{(\theta^{ref})} \in \mathcal{G}_{bf}$ and call it the reference prediction. Then,*

$$d(\eta, g^{bf}) \leq -V(b, \epsilon) + \eta \cdot \log \eta \leq d(\eta, g^{ref}) + 2\epsilon^\top |\theta^{ref}| \leq d(\eta, g^{ref}) + 2\|\epsilon\|_\infty \left\|\theta^{ref}\right\|_1.$$

From this result, we can write the upper bound in terms of model and approximation uncertainty. Let $g^* = g^{(\theta^*)} := \arg\min_{g \in \mathcal{G}_{bf}} d(\eta, g)$. By definition, $g^* \in \mathcal{G}_{bf}$. By also recalling that $d(\eta, g^{bf}) = d(\eta, g^*) + d(g^*, g^{bf})$ (Lemma 20), we get the following.

**Corollary 22** (Theorem 7). *Suppose we had the same assumptions as the above theorem. Then,*

$$d(\eta, g^{bf}) \leq d(\eta, g^*) + 2\epsilon^\top |\theta^*| \leq d(\eta, g^*) + 2\|\epsilon\|_\infty \|\theta^*\|_1$$

*and both $2\epsilon^\top |\theta^*|$, $2\|\epsilon\|_\infty \|\theta^*\|_1$ serve as upper bounds to BF's approximation uncertainty $d(g^*, g^{bf}) = \mathcal{E}_{bf}^{appr}$.*

*Proof of Theorem 21.* We'll first argue why to bound $d(\eta, g^{bf})$, it's sufficient to upper bound $-V(b, \epsilon)$. Recall from Lemma 15

$$-V(b, \epsilon) = -\min_{z \in P} \max_{g \in \Delta_k^n} z^\top \log g = \max_{z \in P} \min_{g \in \Delta_k^n} -z^\top \log g = \max_{z \in P} -z^\top \log g^{bf}.$$

It's clear that the right hand term is larger than or equal to $-\eta^\top \log g^{bf}$, so, $-V(b, \epsilon) + \eta \cdot \log \eta \geq d(\eta, g^{bf})$. Indeed, $-V(b, \epsilon)$ can have its value be computed from a convex program.

We use a sensitivity analysis result to upper bound that convex program's optimal objective value. The program whose objective value we wish to bound is the *arbitrary* program. E.g. the convex program whose optimal value is $-V(b, \epsilon)$ is the arbitrary program. In essense, the result says that if there's another convex program (call it the reference program) with sufficiently similar constraints to the arbitrary program, we can bound the arbitrary program's objective by the reference program's objective plus a weighted sum of the reference program's optimal Lagrange multipliers. We will construct a reference program from our chosen reference prediction $g^{ref}$ and denote its optimal Lagrange multipliers by $\sigma^{ref}$ and $\sigma'^{ref}$. The polytope constraint in the reference program will involve the same matrix $A$, but will have the constraints $Az \leq \widehat{b}$ and $-Az \leq -\widehat{b}$. One can ignore what $\widehat{b}$ is for now. Before we present the bound, define $\epsilon^+ = b + \epsilon - b^*$, the total overestimate of $b^*$ and $\epsilon^- = b^* - b + \epsilon$, the total underestimate of the same quantity. Corollary 25 states

$$-\eta \log g^{bf} \leq -g^{(\theta^{ref})} \cdot \log g^{(\theta^{ref})} + (\sigma'^{ref} - \sigma^{ref})^\top (\widehat{b} - b^*) + \epsilon^{+\top} \sigma^{ref} + \epsilon^{-\top} \sigma'^{ref}.$$

Now, what remains is for us to simplify the terms in the upper bound. Note that by definition, $\theta^{ref} = \sigma'^{ref} - \sigma^{ref}$. Lemma 26 states that

$$\theta^{ref\top}(\widehat{b} - b^*) = (g^{(\theta^{ref})} - \eta)^\top \log g^{(\theta^{ref})}.$$

This means that we can simplify our above bound to

$$-\eta \log g^{bf} \leq -g^{(\theta^{ref})} \cdot \log g^{(\theta^{ref})} + (g^{(\theta^{ref})} - \eta)^\top \log g^{(\theta^{ref})} + \epsilon^{+\top} \sigma^{ref} + \epsilon^{-\top} \sigma'^{ref}$$
$$= -\eta \cdot \log g^{(\theta^{ref})} + \epsilon^{+\top} \sigma^{ref} + \epsilon^{-\top} \sigma'^{ref}.$$

To continue, observe that by definition, $b^* \in [b - \epsilon, b + \epsilon]$. This means that for every element $j$, $|b_j - b_j^*| \leq \epsilon_j$. So, $\epsilon^+, \epsilon^- \leq 2\epsilon$. Thus if we take $|\cdot|$ to be element-wise absolute value,

$$\epsilon^{+\top} \sigma^{ref} + \epsilon^{-\top} \sigma'^{ref} \leq 2\epsilon \cdot |\theta^{ref}| \leq 2\|\epsilon\|_\infty \|\theta^{ref}\|_1.$$

We get the first inequality because we will construct the reference weights $\sigma^{ref}, \sigma'^{ref}$ so that for any element $j$, only one of $\sigma_j^{ref}$ and $\sigma_j'^{ref}$ will be non-zero. Putting everything we have so far together, we have

$$-\eta \cdot \log g^{bf} \leq -V(b, \epsilon) \leq -\eta \cdot \log g^{(\theta^{ref})} + 2\epsilon \cdot |\theta^{ref}| \leq -\eta \cdot \log g^{(\theta^{ref})} + 2\|\epsilon\|_\infty \|\theta^{ref}\|_1.$$

Adding $\eta \cdot \log \eta$ to each of the four sections of the inequality gives us our claim. $\qquad \square$

## I.1 REFERENCE PROGRAM

We now construct a BF program where an arbitrary $g = g^{(\theta)} \in \mathcal{G}_{bf}$ is the optimal solution. This is equivalent to fixing a vector $\theta \in \mathbb{R}^m$.

**Theorem 23.** *Suppose we had some fixed but arbitrary $g = g^{(\theta)} \in \mathcal{G}_{bf}$ and define $\widehat{b} = Ag^{(\theta)}$. Also, define our Lagrange multipliers*

$$\sigma_i = \begin{cases} -\theta_i & if \quad \theta_i < 0 \\ 0 & otherwise \end{cases} \quad and \quad \sigma_i' = \begin{cases} \theta_i & if \quad \theta_i \geq 0 \\ 0 & otherwise. \end{cases}$$

*Then, $g$ and $\sigma, \sigma'$ are jointly optimal for the maximum entropy problem*

$$\min_{z \in P} z \cdot \log z \qquad where \qquad P = \{z \in \Delta_k^n : \widehat{b} \leq Az \leq \widehat{b}\}.$$

*Proof.* We state the KKT conditions and show that they are satisfied. The convex program in question is

$$\min_{\substack{Az-\widehat{b}\leq 0 \\ -Az+\widehat{b}\leq 0 \\ Dz=\vec{1}_n}} z \cdot \log z.$$

Say we associate (like in the proof of Theorem 2) the Lagrange multipliers $\xi$ with constraint $Dz - \vec{1}_n = \vec{0}_n$. Before stating the KKT conditions concretely, note that by construction, Slater's condition is satisfied because we have all affine constraints and the feasible region is non-empty (see Boyd and Vandenberghe [2004], Section 5.2.3). To show that a pair of primal and dual variables $z$, $\sigma$, and $\sigma'$ are jointly optimal, it is necessary and sufficient to satisfy

$$Az - \widehat{b} \leq 0$$
$$-Az + \widehat{b} \leq 0$$
$$Dz - \vec{1}_n = 0$$
$$\sigma \geq 0$$
$$\sigma' \geq 0$$
$$\sigma_i(Az - \widehat{b})_i = 0 \quad i = 1, \dots$$
$$\sigma'_i(-Az + \widehat{b})_i = 0 \quad i = 1, \dots$$
$$\nabla_z \left[ z \cdot \log z + \sigma^\top(Az - \widehat{b}) + \sigma'^\top(-Az + \widehat{b}) + \xi^\top(Dz - \vec{1}_n) \right] = 0.$$

Of course, we will be considering $z = g^{(\theta)}$. The first three requirements are met by construction. To see that the third requirement is met, note that by construction, $g^{(\theta)} \in \Delta_k^n$, meaning that every $k$ elements sum to one, which is what $Dz = \vec{1}_n$ requires. Also, we have constructed $\sigma$ and $\sigma'$ to be non-negative. Now, since $Az = \widehat{b}$, the complementary slackness conditions are trivially satisfied. To see that the zero gradient condition is satisfied, note that in solving for the functional form of the learner's prediction (or adversary's labeling), we took the gradient of the Lagrangian and set the gradient equal to 0 and solved for $z$. In us choosing $\sigma^{ds}$, we use the functional form of $z$, so we automatically satisfy the zero gradient condition. Therefore, all of the KKT conditions are satisfied. □

## I.2  SENSITIVITY ANALYSIS OF AN ARBITRARY BF PROGRAM

We now want to write an arbitrary BF program as an instance of the reference program with perturbed constraints. This will allow us to bound $-V(b, \epsilon)$. Define our perturbed problem as follows.

$$V^*(\widehat{b}, u_1, u_2) := \min_{\substack{Az-\widehat{b}\leq u_1 \\ -Az+\widehat{b}\leq u_2 \\ Dz-\vec{1}_n=\vec{0}_n}} z \cdot \log z.$$

If $u_1 = u_2 = 0$, i.e. $V^*(\widehat{b}, 0, 0)$, then we get our reference program. Recall that we have defined $\epsilon^+ = b + \epsilon - b^*$ and $\epsilon^- = b^* - b + \epsilon$. So, if we choose

$$u_1 = -\widehat{b} + b^* + \epsilon^+ \quad \text{and} \quad u_2 = \widehat{b} - b^* + \epsilon^-,$$

we have

$$V^*(\widehat{b}, -\widehat{b} + b^* + \epsilon^+, \widehat{b} - b^* + \epsilon^-) = \min_{\substack{Az-\widehat{b}\leq -\widehat{b}+b^*+\epsilon^+ \\ -Az+\widehat{b}\leq \widehat{b}-b^*+\epsilon^- \\ Dz-\vec{1}_n=\vec{0}_n}} z \cdot \log z = V(b, \epsilon)$$

which is the arbitrary BF program whose optimal value we have discussed beforehand. In the proof of Theorem 21, we saw that $-V(b, \epsilon) \geq -\eta^\top \log g^{bf}$. So, getting a lower bound for $V^*(\widehat{b}, u_1, u_2)$ (with appropriately chosen $u_1, u_2$) would give an upper bound for $-V(b, \epsilon)$.

Boyd and Vandenberghe [2004] provide a global sensitivity result (Section 5.6.2). We state the result for our program without proof. Indeed, the requirement for the following result is that the reference or unperturbed program satisfy Slater's condition, which we have argued for in Section I.1. We associate $\sigma^{ref}$ with the upper bound for $Az$, i.e. $Az - \widehat{b} \leq 0$, while $\sigma'^{ref}$ is associated with $-Az + \widehat{b} \leq 0$.

**Theorem 24** (Boyd and Vandenberghe [2004] Section 5.6.2, Equation 5.57)**.** *If $\sigma$ and $\sigma'$ are dual optimal for the unperturbed (or reference) problem, then for all $u_1$ and $u_2$, we have*

$$V^*(\widehat{b}, u_1, u_2) \geq V^*(\widehat{b}, 0, 0) - \sigma^{ref\top} u_1 - \sigma'^{ref\top} u_2.$$

The result we use in the proof of the BF bound follows from plugging in our choices for $u_1$ and $u_2$ and rearranging.

**Corollary 25.**

$$-\eta \log g^{bf} \leq -g^{(\sigma^{ref})} \cdot \log g^{(\sigma^{ref})} + (\sigma'^{ref} - \sigma^{ref})^\top (\widehat{b} - b^*) + \epsilon^{+\top} \sigma^{ref} + \epsilon^{-\top} \sigma'^{ref}$$

*Proof.* First, reverse the direction of the inequality in Theorem 24.

$$-V^*(\widehat{b}, u_1, u_2) \leq -V^*(\widehat{b}, 0, 0) + \sigma^{ref\top} u_1 + \sigma'^{ref\top} u_2.$$

Plugging in

$$u_1 = -\widehat{b} + b^* + \epsilon^+ \quad \text{and} \quad u_2 = \widehat{b} - b^* + \epsilon^-$$

as chosen above gives

$$-g^{bf} \cdot \log g^{bf} \leq -g^{ref} \cdot \log g^{ref} + \sigma^{ref\top} u_1 + \sigma'^{ref\top} u_2.$$

because by construction, $V(\widehat{b}, u_1, u_2) = g^{bf} \cdot \log g^{bf}$ and $V^*(\widehat{b}, 0, 0) = V(\widehat{b}, 0) = g^{ref} \cdot \log g^{ref}$. Explicitly expanding $u_1, u_2$ gives

$$-\eta \cdot \log g^{bf} \leq -g^{ref} \cdot \log g^{ref} + (\sigma'^{ref} - \sigma^{ref})^\top (\widehat{b} - b^*) + \epsilon^{+\top} \sigma^{ref} + \epsilon^{-\top} \sigma'^{ref}$$

where we recall that in Theorem 21, we showed that $-\eta \cdot \log g^{bf} \leq -g^{bf} \cdot \log g^{bf}$. $\qquad\square$

## I.3  SIMPLIFICATION OF TERMS IN SENSITIVITY ANALYSIS

We wish to now simplify the term $(\sigma'^{ref} - \sigma^{ref})^\top (\widehat{b} - b^*)$. Recall that by construction of $\sigma^{ref}$ and $\sigma'^{ref}$ that $\sigma'^{ref} - \sigma^{ref} = \theta^{ref}$. So, we analyze $\theta^{ref\top} (\widehat{b} - b^*)$.

**Lemma 26.**

$$\theta^{ref\top} (\widehat{b} - b^*) = (g^{(\theta^{ref})} - \eta)^\top \log g^{(\theta^{ref})}.$$

*Proof.* Observe that by definition and construction respectively, $b^* = A\eta$ and $\widehat{b} = Ag^{(\theta^{ref})}$. Therefore, the left hand side of our claim can be written as

$$\theta^{ref\top} (\widehat{b} - b^*) = \theta^{ref\top} (A(g^{(\theta^{ref})} - \eta)) = (g^{(\theta^{ref})} - \eta)^\top A^\top \theta^{ref}.$$

But by definition, that's equal to

$$(g^{(\theta^{ref})} - \eta)^\top a^{(\theta^{ref})} = \sum_{i=1}^{n} \sum_{\ell=1}^{k} (g_{i\ell}^{(\theta^{ref})} - \eta_{i\ell}) a_{i\ell}^{(\theta^{ref})}.$$

Since $g_i^{(\theta^{ref})}$ and $\eta_i$ are each distributions (in $\Delta_k$), for any constant, especially

$$Z_i^{(\theta^{ref})} := \log \left( \sum_{\ell=1}^{k} a_{i\ell}^{(\theta^{ref})} \right), \quad \text{we have} \quad \sum_{\ell=1}^{k} (g_{i\ell}^{(\theta^{ref})} - \eta_{i\ell}) Z_i^{(\theta^{ref})} = 0.$$

Using this, we can subtract 0 to our above sum to get

$$\sum_{i=1}^{n} \sum_{\ell=1}^{k} (g_{i\ell}^{(\theta^{ref})} - \eta_{i\ell}) \log(\exp(a_{i\ell}^{(\theta^{ref})})) - \sum_{i=1}^{n} \sum_{\ell=1}^{k} (g_{i\ell}^{(\theta^{ref})} - \eta_{i\ell}) Z_i^{(\theta^{ref})}$$

$$= \sum_{i=1}^{n} \sum_{\ell=1}^{k} (g_{i\ell}^{(\theta^{ref})} - \eta_{i\ell}) \log \left( \frac{\exp(a_{i\ell}^{(\theta^{ref})})}{Z_i^{(\theta^{ref})}} \right)$$

But, the term inside the logarithm is exactly $g_{i\ell}^{(\theta^{ref})}$ and our claim is proved. $\qquad\square$

# J PROOF OF THEOREM 6

**Theorem 27** (Theorem 6). *There is only one best approximator to $\eta$ in $\mathcal{G}$, i.e. $\arg\min_{g \in \mathcal{G}} d(\eta, g)$ has only one element. Call that best approximator $g^* = g^{(\theta^*)} \in \mathcal{G}$. When $\epsilon = \vec{0}_m$, the learner's prediction gotten from solving $V(b^*, \vec{0}_m)$, call it $g^{bf*}$, is exactly $g^*$.*

*Proof.* We begin by stating Theorem 21 with $\epsilon = \vec{0}_m$. For any $g \in \mathcal{G}_{bf}$,

$$d(\eta, g^{bf*}) \le d(\eta, g).$$

This implies that

$$g^{bf*} \in \arg\min_{g \in \mathcal{G}} d(\eta, g).$$

To show all claims, we need to show that $g^{bf*}$ is the only element in the set of best approximators to $\eta$. Since $g^{bf*}$ is the optimal solution to $V(b^*, \vec{0}_m)$, it follows that $Ag^{bf*} = b^* = A\eta$. Now, suppose for contradiction that there existed another $g^{(\theta')} \in \mathcal{G}$ where $g^{(\theta')} \ne g^*$ and

$$g^{(\theta')} \in \arg\min_{g \in \mathcal{G}} d(\eta, g).$$

That is, there are two different best approximators to $\eta$ from $\mathcal{G}$. This means that $d(\eta, g^{bf*}) = d(\eta, g^{(\theta')})$. Since $g^{bf*} \in \mathcal{G}$ and $Ag^{bf*} = A\eta$, we can use the Pythagorean theorem (Lemma 19) to see that

$$d(\eta, g^{(\theta')}) = d(\eta, g^{bf*}) + d(g^{bf*}, g^{(\theta')}).$$

But from what we have just said, we have

$$0 = d(g^{bf*}, g^{(\theta')}) = \sum_{i=1}^{n} \sum_{\ell=1}^{k} g_{i\ell}^{bf*\top} \log\left(\frac{g_{i\ell}^{bf*}}{g_{i\ell}^{(\theta')}}\right)$$

It suffices to show that the only way this equality can hold is when $g^{(\theta')} = g^{bf*}$ element-wise. This would contradict our assumption that $g^{bf*} \ne g^{(\theta')}$.

To derive the contradiction, divide the RHS by $\sum_{i=1}^{n} \sum_{\ell=1}^{k} g_{i\ell}^{bf*} = n$ to get

$$-\frac{1}{n} \sum_{i=1}^{n} \sum_{\ell=1}^{k} g_{i\ell}^{bf*\top} \log\left(\frac{g_{i\ell}^{(\theta')}}{g_{i\ell}^{bf*}}\right).$$

Observe that $-\log(\cdot)$ is a convex function. Therefore, by Jensen's inequality,

$$-\frac{1}{n} \sum_{i=1}^{n} \sum_{\ell=1}^{k} g_{i\ell}^{bf*\top} \log\left(\frac{g_{i\ell}^{(\theta')}}{g_{i\ell}^{bf*}}\right) \ge -\frac{1}{n} \sum_{i=1}^{n} \sum_{\ell=1}^{k} \log\left(g_{i\ell}^{bf*} \frac{g_{i\ell}^{(\theta')}}{g_{i\ell}^{bf*}}\right) = 0.$$

The equality condition of Jensen's inequality states that the inequality above is equality if and only if for all $i, i' \in [n], \ell, \ell' \in [k]$,

$$\frac{g_{i\ell}^{bf*}}{g_{i\ell}^{(\theta')}} = \frac{g_{i'\ell'}^{bf*}}{g_{i'\ell'}^{(\theta')}} = R$$

where $R \in \mathbb{R} \setminus \{0\}$ is the common ratio. Recall that by definition, for each $i \in [n]$

$$\sum_{\ell=1}^{k} g_{i\ell}^{bf*} = \sum_{\ell=1}^{k} g_{i\ell}^{(\theta')} = 1$$

For fixed but arbitrary $i \in [n]$, using the common ratio, we have

$$1 = \sum_{\ell=1}^{k} g_{i\ell}^{bf*} = \sum_{\ell=1}^{k} R g_{i\ell}^{(\theta')} = R.$$

Therefore, Jensen's inequality turns into an equality only when $g^{bf*} = g^{(\theta')}$ element-wise. This finishes the argument by contradiction and we conclude that $g^{bf*}$ is the unique best approximator to $\eta$ from the set $\mathcal{G}$. Therefore, $g^{bf*} = g^*$. $\square$

**Corollary 28.** $Ag^* = A\eta$.

*Proof.* Observe that the following equalities hold:

$$A\eta = b^* = Ag^{bf*} = Ag^*.$$

The first equality is by definition. The second and third equalities follow by the previous result (Theorem 27). Namely, $Ag^{bf*} = b^*$ by construction and it was shown above that $g^{bf*} = g^*$. $\qquad\square$

# K ALMOST ALL OCDS PREDICTIONS ARE IN $\mathcal{G}_{bf}$

We show now that almost all DS predictions fall into the exponential family of BF predictions $\mathcal{G}_{bf}$. The main restriction is that elements in the DS prediction (See Equation 8) cannot be 0 or 1. This is to avoid expressions with $\infty$, especially $\infty/\infty$. So, we'll define $\mathcal{G}_{ds}^{\circ}$ to be the set of all OCDS predictions where no class frequency or accuracy is 0 or 1. The set of OCDS predictions $\mathcal{G}_{ds}$ is the closure of $G_{ds}^{\circ}$.

The strategy will be to work backwards from an OCDS prediction and to write it as a softmax, involving the matrix $A$. Then, it suffices to check what the possible weights are. We also show how to go from a prediction's weights to OCDS parameters (class frequencies/rule accuracies). These OCDS parameters can be used to construct that same prediction via E step.

To continue, we need to represent which rules predict what class on datapoints. This is because we allow rules to abstain and want to talk about the rules that don't abstain on a datapoint. Define a function $\rho\colon [n] \times [k] \to 2^{[p]}$ be the function that returns which rules predicted a certain label on a certain datapoint. For example, $\rho(i, \ell)$ returns the rules that predict class $\ell$ on datapoint $x_i$.

Suppose $A$ was the matrix that encoded accuracy constraints for our $p$ rules and class frequency constraints (a la the main paper). Abuse notation and call the OCDS prediction for class $\ell$ on datapoint $i$ with class frequencies $w$ and rule accuracies $b$

$$g_{i\ell}^{ds}(w, b) \propto w_\ell \prod_{j \in \rho(i,\ell)} b_j \prod_{\substack{\ell'=1 \\ \ell' \neq \ell}}^{k} \prod_{j' \in \rho(i,\ell')} \frac{1 - b_{j'}}{k - 1}.$$

This forms a distribution as $\ell$ varies in $[k]$. Let the OCDS prediction with $w, b$ on all $n$ datapoints in $X$ be denoted $g^{ds}(w, b)$. Then for element-wise inequality,

$$\mathcal{G}_{ds}^{\circ} := \{g^{ds}(w, b)\colon 0 < w, b < 1\}.$$

**Lemma 29.** *For accuracy constraints and rules that possibly abstain, $\mathcal{G}_{bf} = \mathcal{G}_{ds}^{\circ}$.*

*Proof.* In Lemma 30 below, we show that for any prediction $g^{ds} \in \mathcal{G}_{ds}^{\circ}$, there are real weights $\theta^{ds}$ such that $g^{(\theta^{ds})} = g^{ds}$. By allowing each of the accuracies and class frequencies to range from $(0, 1)$, all possible real weights are exhibited. As $\mathcal{G}_{bf}$ is the set of predictions $g^{(\theta)}$ attainable by all real weights $\theta \in \mathbb{R}^m$, we have our claim. $\qquad\square$

**Lemma 30.** *For any $g^{ds} \in \mathcal{G}_{ds}^{\circ}$ constructed by one E step on accuracies $0 < b_1, \ldots, b_p < 1$ and class frequencies $0 < w_1, \ldots, w_k < 1$, the weights*

$$\theta_{p+\ell} = \log\left(e^n w_\ell\right) \quad \text{and} \quad \theta_j = \log\left(e^{n_j} \frac{b_j(k-1)}{1 - b_j}\right),$$

*are real and are such that $g^{(\theta)} = g^{ds}$.*

*Proof.* We show this by working backwards, going from writing the DS prediction as a softmax to bringing about the matrix $A$ used in $\mathcal{G}_{bf}$'s definition. In doing this, the claimed weights will show up. Write the quantity that the DS prediction is proportional to (from above) as a softmax:

$$\frac{\exp\left(\log\left(w_\ell\right) + \sum_{j \in \rho(i,\ell)} \log\left(b_j\right) + \sum_{\substack{\ell'=1 \\ \ell' \neq \ell}}^{k} \sum_{j' \in \rho(i,\ell')} \log\left(\frac{1 - b_{j'}}{k - 1}\right)\right)}{\sum_{\ell'=1}^{k} \exp\left(\log\left(w_{\ell'}\right) + \sum_{j \in \rho(i,\ell')} \log\left(b_j\right) + \sum_{\substack{\ell''=1 \\ \ell'' \neq \ell'}}^{k} \sum_{j'' \in \rho(i,\ell'')} \log\left(\frac{1 - b_{j''}}{k - 1}\right)\right)}.$$

Observe that we can write this as

$$\frac{\exp\left(\log(w_\ell) + \sum_{j\in\rho(i,\ell)}\log\left(\frac{b_j(k-1)}{1-b_j}\right) + \sum_{j'\in\cup_{\ell'=1}^k\rho(i,\ell')}\log\left(\frac{1-b_{j'}}{k-1}\right)\right)}{\sum_{\ell'=1}^k\exp\left(\log(w_{\ell'}) + \sum_{j\in\rho(i,\ell')}\log\left(\frac{b_j(k-1)}{1-b_j}\right) + \sum_{j''\in\cup_{\ell''=1}^k\rho(i,\ell'')}\log\left(\frac{1-b_{j''}}{k-1}\right)\right)}.$$

Note that we can't make the last sum range over $[p]$ because we allow for specialists, so not all rules necessarily have to make a prediction. Before moving forward, notice that the last term in every exponent is the same. Therefore, those make no contribution to the softmax. The softmax can be written as

$$\frac{\exp\left(\log(w_\ell) + \sum_{j\in\rho(i,\ell)}\log\left(\frac{b_j(k-1)}{1-b_j}\right)\right)}{\sum_{\ell'=1}^k\exp\left(\log(w_{\ell'}) + \sum_{j\in\rho(i,\ell')}\log\left(\frac{b_j(k-1)}{1-b_j}\right)\right)}. \tag{9}$$

To continue, recall the weights we presented in the claim,

$$\theta_{p+\ell} = \log(e^n w_\ell) \quad \text{and} \quad \theta_j = \log\left(e^{n_j}\frac{b_j(k-1)}{1-b_j}\right)$$

where once again $j \in [p]$ and $\ell \in [k]$. We would like to show that the linear combination in the numerator is $a_{i\ell}^{(\theta)}$ (the $(k(i-1)+\ell)^{th}$ element of the vector $A^\top\theta$). Indeed, observe (from the definition of $A$ in Subsection B.3) that

$$a_{i\ell}^{(\theta)} = \sum_{j=1}^p \frac{1}{n_j}h_{i\ell}^{(j)}\theta_j + \sum_{\ell'=1}^k \frac{1}{n}[\vec{e}_{\ell'}^{\,n}]_{i\ell}\theta_{p+\ell'}.$$

By transcribing the definitions of our matrix entries, we have that

$$a_{i\ell}^{(\theta)} = \sum_{j=1}^p \frac{1}{n_j}\theta_j\mathbf{1}(h^{(j)}(x_i) = \ell) + \sum_{\ell'=1}^k \frac{1}{n}\theta_{p+\ell'}\mathbf{1}(\ell' = \ell).$$

So, the only $\theta_j$'s that appear are associated with the rules that predict label $\ell$ on datapoint $x_i$. This is exactly the set of rules that $\rho(i,\ell)$ represents. Also, there's only one value of $\ell'$ that makes the indicator non-zero. We can now write

$$a_{i\ell}^{(\theta)} = \sum_{j\in\rho(i,\ell)} \frac{1}{n_j}\theta_j + \frac{1}{n}\theta_{p+\ell} = \log(w_\ell) + \sum_{j\in\rho(i,\ell)}\log\left(\frac{b_j(k-1)}{1-b_j}\right),$$

after plugging in the claimed weights $\theta$. This is exactly the argument of the numerator in Equation 9. So, we have exhibited a set of weights $\theta$ such that the DS prediction is recovered. $\qquad\square$

We can now go backwards and solve for the accuracies and class frequencies.

**Lemma 31.** *An arbitrary prediction $g^{(\theta)} \in \mathcal{G}_{bf}$ is a one-coin DS prediction with*

$$b_j = \frac{1}{1 + e^{n_j}(k-1)\exp(-\theta_j)} \quad \text{and} \quad w_\ell = \frac{\exp(\theta_{p+\ell})}{\sum_{\ell'=1}^k\exp(\theta_{p+\ell'})}$$

*Moreover, the computed $b_j$ and $w_\ell$ all reside in $(0,1)$ and the $w_\ell$'s form a distribution.*

*Proof.* For rule weight $\theta_j$ where $j \in [p]$, it suffices to solve for $b_j$ in

$$\theta_j = \log\left(e^{n_j}\frac{b_j(k-1)}{1-b_j}\right).$$

Doing some quick algebra, we have

$$\exp\left(\theta_j\right)\left(1 - b_j\right) = e^{n_j} b_j \left(k - 1\right) \quad \text{meaning} \quad \exp\left(\theta_j\right) = b_j(e^{n_j}\left(k - 1\right) + \exp\left(\theta_j\right))$$

We conclude that

$$b_j = \frac{1}{1 + e^{n_j}\left(k - 1\right)\exp\left(-\theta_j\right)}.$$

As $\theta_j \to -\infty$, the right hand side tends to 0, while $\theta \to \infty$ means the right hand side tends to 1. Since we require the $\theta$ value to be strictly real, we will never get $b_j \in \{0, 1\}$.

Now, we handle the class frequency weights, i.e. weights $\theta_{p+1}, \ldots, \theta_{p+k}$. Since the class frequencies are a distribution, the class frequency that we choose must be such that $\sum_{\ell=1}^{k} w_\ell = 1$. Observe also that a class frequency weight appears in every argument of the prediction softmax (see proof of Lemma 30). Therefore, we can add $\log(\widehat{w})$ to every argument of the softmax without changing its value. Taking that into the class frequency weight, we can write

$$\theta_{p+\ell} = \log\left(e^n \frac{w_\ell}{\widehat{w}}\right) \quad \text{meaning} \quad w_\ell = \frac{\exp\left(\theta_{p+\ell}\right)\widehat{w}}{e^n}.$$

Since we require $\sum_{\ell=1}^{k} w_\ell = 1$,

$$\frac{\widehat{w}}{e^n} \sum_{\ell=1}^{k} \exp\left(\theta_{p+\ell}\right) = 1 \quad \text{we infer that} \quad \widehat{w} = \frac{e^n}{\sum_{\ell=1}^{k} \exp\left(\theta_{p+\ell}\right)}.$$

Plugging in this $\widehat{w}$ and recognizing the result is a softmax gives us our claim. This means that arbitrary distributions from the exponential family are actually one-coin DS predictions whose accuracies and class frequencies are in $(0, 1)$. $\qquad\square$

## L   DAWID SKENE ERROR EXPRESSION

We now decompose the OCDS loss into terms corresponding to model and approximation uncertainty. Even though the BF and OCDS prediction sets are technically different, they're sufficiently similar so that OCDS has the same model uncertainty as BF, which we'll show. We use $g^\dagger$ to represent a best OCDS approximator to $\eta$, compared to $g^*$ for BF's best approximator ($g^\dagger \in \arg\min_{g \in \mathcal{G}_{ds}} d(\eta, g)$). We'll not endeavor to show that $g^\dagger$ is unique.

We show a general decomposition which can be simplified by virtue of dealing with OCDS predictions.

### L.1   DS LOSS DECOMPOSITION

**Lemma 32.** *Let $g^\dagger \in \mathcal{G}_{ds}$ be a best approximator to $\eta$, $g^{ds}$ be a OCDS prediction with estimated parameters (w, b), and $g^{ds*}$ be the DS prediction using empirical parameters ($w^*, b^*$). Then,*

$$\underbrace{d(\eta, g^{ds})}_{\mathcal{E}_{ds}} = \underbrace{d(\eta, g^\dagger)}_{\mathcal{E}_{ds}^{mod}} + \mathcal{E}_{ds}^{appr}$$

$$\mathcal{E}_{ds}^{appr} = \underbrace{d(\eta, g^{ds*}) - d(\eta, g^\dagger)}_{\mathcal{E}_{ds,1}^{appr}} + \underbrace{\sum_{i=1}^{n}\sum_{\ell=1}^{k} \eta_{i\ell} \log\left(\frac{g_{i\ell}^{ds*}}{g_{i\ell}^{ds}}\right)}_{\mathcal{E}_{ds,2}^{appr}}.$$

*Moreover, $d(\eta, g^\dagger) = d(\eta, g^*)$.*

*Proof.* The first claim is by definition. Then, by adding 0 twice, one immediately gets

$$d(\eta, g^{ds}) = d(\eta, g^\dagger) + d(\eta, g^{ds*}) - d(\eta, g^\dagger) + d(\eta, g^{ds}) - d(\eta, g^{ds*}).$$

The first term after the equals sign corresponds with the model uncertainty while the remaining terms are the approximation uncertainty. One can simplify the last two terms as follows:

$$d(\eta, g^{ds}) - d(\eta, g^{ds*}) = \sum_{i=1}^{n} \sum_{\ell=1}^{k} \eta_{i\ell} \left[ \log\left(\frac{\eta_{i\ell}}{g_{i\ell}^{ds}}\right) - \log\left(\frac{\eta_{i\ell}}{g_{i\ell}^{ds*}}\right) \right] = \sum_{i=1}^{n} \sum_{\ell=1}^{k} \eta_{i\ell} \log\left(\frac{g_{i\ell}^{ds*}}{g_{i\ell}^{ds}}\right).$$

For our very last claim, recall that Lemma 29 shows that $\mathcal{G}_{bf} = \mathcal{G}_{ds}^{\circ}$. The BF predictions missing from $\mathcal{G}_{ds} \setminus \mathcal{G}_{bf}$ are exactly the limit points of $\mathcal{G}_{bf}$. Specifically, the DS predictions where a rule accuracy $b_j$ or class frequency $w_\ell$ is in the set $\{0, 1\}$. Thus if $g^\dagger \notin \mathcal{G}_{bf}$, the best approximator in $\mathcal{G}_{bf}$ is arbitrarily close to $g^\dagger$. Therefore, the BF and DS model uncertainties are arbitrarily close. $\qquad\square$

Now, in our analysis of the last term, we will encounter the normalizing constant for the DS prediction. For the one-coin DS model, it is

$$Z_i^{ds} = \sum_{\ell=1}^{k} w_\ell \prod_{j\in\rho(i,\ell)} b_j \prod_{\substack{\ell'=1\\\ell'\neq\ell}}^{k} \prod_{j'\in\rho(i,\ell')} \frac{1 - b_{j'}}{k-1}. \tag{10}$$

We'll also consider

$$Z_i^* = \mathrm{Pr}_n(h^{(j)}(x) = h^{(j)}(x_i), \forall j \in [p]) = \frac{1}{n}|\{x \in X : h^{(j)}(x_i) = h^{(j)}(x), \forall j \in [p]\}|,$$

or the fraction of datapoints where the ensemble's predictions are the same as the ones on datapoint $x_i$. This quantity is known when one has the ensemble predictions. However, the DS model predicts this implicitly when it normalizes, e.g. the denominator on the right hand side of Equation 8. Thus, it comes into play.

We will consider the distribution of *unique* ways the ensemble can predict on datapoints. Since $Z_i^*$ is defined in terms of datapoint index $i$, there may be duplicates. For example, if all rules predict class 1 on the first three datapoints, $Z_1^* = Z_2^* = Z_3^*$. More generally, if we have $p$ rules that predict on all points and $k$ classes, there are $k^p$ possible ways for the ensemble to predict. I.e. each rule can predict any of the $k$ classes. However, it's often the case that not all $k^p$ ways to predict obtain, for usually $n < k^p$ or the ensemble makes the same prediction on different datapoints. So, we let $Z^*, Z^{ds*}, Z^{ds}$ be the empirical, empirical OCDS, and OCDS distributions of unique ways for the ensemble to predict. To be clear, the next two quantities are in reference to datapoint $x_i$ and not the $i^{th}$ "unique way for the ensemble to predict". For example, $Z_i^{ds}$ is from Equation 10. One can get $Z_i^{ds*}$ by substituting in the empirical class frequencies and accuracies.

## L.2 ONE-COIN ERROR EXPRESSION (LEMMA 10)

We can actually simplify the last term in the OCDS error expression. While we have used $w \in \Delta_k$ to represent a class frequency distribution, we'll use $\vec{w}$ to emphasize that it is a vector.

**Lemma 33** (Lemma 10). *Fix OCDS prediction $g^{ds}$ gotten from applying one E step to fixed class frequencies $\vec{w}$ and fixed rule accuracies $b$. For $g^{ds}$ and $g^{ds*}$,*

$$d(\eta, g^{ds}) = d(\eta, g^*) + d(\eta, g^{ds*}) - d(\eta, g^*) + n\left( KL(\vec{w}^*, \vec{w}) + \sum_{j=1}^{p} \frac{n_j}{n} KL(\vec{b_j^*}, \vec{b_j}) \right.$$

$$\left. + KL(Z^*, Z^{ds*}) - KL(Z^*, Z^{ds}) \right)$$

*where $\vec{w}^*$ is the vector of empirical class frequencies, while $\vec{b_j}$ and $\vec{b_j^*}$ are the distributions $(b_j, 1 - b_j)^\top$ of the fixed and empirical accuracies for rule $j$.*

*Proof.* We show this by simplifying the last sum in Lemma 32. (By that same Lemma, we can replace $d(\eta, g^\dagger)$ by $d(\eta, g^*)$.) For a one-coin prediction, it is equal to

$$\sum_{i=1}^{n} \sum_{\ell=1}^{k} \eta_{i\ell} \left[ \log\left( \frac{w_\ell^* \prod_{j\in\rho(i,\ell)} b_j^* \prod_{\substack{\ell'=1\\\ell'\neq\ell}}^{k} \prod_{j'\in\rho(i,\ell')} \frac{1-b_{j'}^*}{k-1}}{w_\ell \prod_{j\in\rho(i,\ell)} b_j \prod_{\substack{\ell'=1\\\ell'\neq\ell}}^{k} \prod_{j'\in\rho(i,\ell')} \frac{1-b_{j'}}{k-1}} \right) + \log\left(\frac{Z_i^{ds}}{Z_i^{ds*}}\right) \right].$$

We have used our above definition of $Z$. Observe that because the last logarithm term doesn't depend on $\ell$, it becomes

$$\sum_{i=1}^{n} \left[ \log \left( Z_i^{ds} \right) - \log \left( Z_i^{ds*} \right) \right].$$

Before simplifying further, define $\tau$ as the total of number of unique ways the ensemble predicts on a datapoint. $\tau \leq (k+1)^p$ because each of the $p$ rules can predict any of the $k$ classes or abstain. For each $t \in [\tau]$, define $i_t \in [n]$ to be the first datapoint where the ensemble predicts like the $t^{th}$ unique way. Now, define $Z_{i_t}^*$ as the fraction of points where the ensemble predicts as the $t^{th}$ unique way. We may understand this another way: suppose we fix some datapoint $x_{i_t}$ and randomly select one out of the $n$ datapoints $x$. $Z_{i_t}^*$ is the probability that the rule predictions on $x$ match the rule predictions on $x_{i_t}$. Then,

$$\sum_{t=1}^{\tau} Z_{i_t}^* = 1.$$

Thus, our sum of differences of logarithms becomes

$$\sum_{t=1}^{\tau} n Z_{i_t}^* \left[ \log \left( Z_{i_t}^{ds} \right) - \log \left( Z_{i_t}^{ds*} \right) \right].$$

By adding 0, this is equal to

$$\sum_{t=1}^{\tau} n Z_{i_t}^* \left[ \log \left( \frac{Z_{i_t}^{ds}}{Z_{i_t}^*} \right) + \log \left( \frac{Z_{i_t}^*}{Z_{i_t}^{ds*}} \right) \right] = n(KL(Z^*, Z^{ds*}) - KL(Z^*, Z^{ds})).$$

This is equal to the difference of KL divergences between the empirical and predicted distributions of ensemble predictions. $Z_{i_t}^*$ is the probability that for each rule $h^{(j)}$, $h^{(j)}(x_{i_t}) = h^{(j)}(x)$. Observe that $Z_{i_t}^*$ in general does not equal either $Z_{i_t}^{ds}$ or $Z_{i_t}^{ds*}$, meaning this quantity is non-zero.

Now, we take on the first logarithm term. To get rid of the fraction, define

$$\xi_j = \log \left( \frac{b_j^*}{b_j} \right), \quad \widehat{\xi}_j = \log \left( \frac{1 - b_j^*}{1 - b_j} \right), \quad \text{and} \quad \xi_{p+\ell} = \log \left( \frac{w_\ell^*}{w_\ell} \right).$$

Also, by turning the products in the logarithm into sums outside the logarithm, we get

$$\sum_{i=1}^{n} \sum_{\ell=1}^{k} \eta_{i\ell} \left[ \xi_{p+\ell} + \sum_{j \in \rho(i,\ell)} \xi_j + \sum_{\substack{\ell'=1 \\ \ell' \neq \ell}}^{k} \sum_{j' \in \rho(i,\ell')} \widehat{\xi}_{j'} \right].$$

Lets first simplify

$$\sum_{\ell=1}^{k} \sum_{i=1}^{n} \eta_{i\ell} \xi_{p+\ell}.$$

Observe that by summing over $i$ first, we are computing $n w_\ell^*$. Thus, the above is equal to

$$\sum_{\ell=1}^{k} n w_\ell^* \log \left( \frac{w_\ell^*}{w_\ell} \right) = n KL(\vec{w}^*, \vec{w}).$$

We deal with the remaining terms now. Rather than summing over specific $j$ and $j'$, we can use indicators as follows.

$$\sum_{i=1}^{n} \sum_{\ell=1}^{k} \eta_{i\ell} \left[ \sum_{j=1}^{p} \xi_j \mathbf{1}(h^{(j)}(x_i) = \ell) + \sum_{\substack{\ell'=1 \\ \ell' \neq \ell}}^{k} \sum_{j'=1}^{p} \widehat{\xi}_{j'} \mathbf{1}(h^{(j')}(x_i) = \ell') \right].$$

Now, recall that

$$\sum_{i=1}^{n} \sum_{\ell=1}^{k} \eta_{i\ell} \mathbf{1}(h^{(j)}(x_i) = \ell) = n_j b_j^* \quad \text{so that} \quad n_j(1 - b_j^*) = \sum_{i=1}^{n} \sum_{\ell=1}^{k} \sum_{\substack{\ell'=1 \\ \ell' \neq \ell}}^{k} \eta_{i\ell} \mathbf{1}(h^{(j)}(x_i) = \ell').$$

This means our above sum is actually equal to

$$\sum_{j=1}^{p} n_j b_j^* \log\left(\frac{b_j^*}{b_j}\right) + \sum_{j=1}^{p} n_j (1-b_j^*) \log\left(\frac{1-b_j^*}{1-b_j}\right).$$

Defining $\vec{b_j} = (b_j, 1-b_j)^\top$ and respectively for $b_j^*$, the above becomes

$$\sum_{j=1}^{p} n_j KL(\vec{b_j^*}, \vec{b_j}).$$

Putting everything together,

$$\sum_{i=1}^{n}\sum_{\ell=1}^{k} \eta_{i\ell} \log\left(\frac{g_{i\ell}^{(\theta^{ds*})}}{g_{i\ell}^{(\theta^{ds})}}\right) = n\left(KL(\vec{w}^*, \vec{w}) + \sum_{j=1}^{p} \frac{n_j}{n} KL(\vec{b_j^*}, \vec{b_j}) + KL(Z^*, Z^{ds*}) - KL(Z^*, Z^{ds})\right).$$

$\square$

## M  BF AND DS ERROR COMPARISON

We now have everything needed to be able to compare BF and DS by looking at their model and approximation uncertainties. The first case considered is when BF gets the empirical parameters, i.e. $\epsilon = \vec{0}_m$.

**Theorem 34.** *For any set of rule predictions, if the empirical rule accuracies $b^*$ and empirical class frequencies $w^*$ are given to BF (so it predicts $g^{bf} = g^*$), the learner's best-play is always better/no worse than any DS prediction. That is, for any OCDS prediction $g^{ds}$,*

$$d(\eta, g^{bf}) \le d(\eta, g^{ds}).$$

*Proof.* From Lemma 29, $\mathcal{G}_{bf} = \mathcal{G}_{ds}^\circ$. Since BF is given $b^*, w^*$, we know that the BF prediction $g^{bf}$, is equal to $g^*$, the best approximator of $\eta$ from $\mathcal{G}_{bf}$ (Theorem 27). With these two facts, $g^*$ is better than any prediction in $\mathcal{G}_{ds}^\circ$. However, there are also DS predictions outside of $\mathcal{G}_{ds}$. So, take $g^{ds} \in \mathcal{G}_{ds} \setminus \mathcal{G}_{bf}$. Since $g^{ds}$ is a limit point of $\mathcal{G}_{ds}^\circ$, there is a sequence $\{g_i^{ds+}\}_{i=1}^{n}$ whose limit is $g^{ds}$. (Note that $g_i^{ds+} \in \mathcal{G}_{ds}^\circ$ for all $i$ and that we have abused notation with the subscript index.) We have already established that for any $i$,

$$d(\eta, g^{bf}) \le d(\eta, g_i^{ds^+}) = d(\eta, g^{ds}) + d(\eta, g_i^{ds+}) - d(\eta, g^{ds}).$$

Because $g_i^{ds+} \to g^{ds}$, the last difference is arbitrarily small and

$$d(\eta, g^{bf}) \le d(\eta, g^{ds}),$$

which is what we wanted to show. $\square$

### M.1  LEMMA 12

We now have the error bound presented in the paper in full detail.

**Lemma 35** (Lemma 12). *If $g^{bf}$ from $V(b, \epsilon)$, $g^{(\theta^*)}$ the best approximator from $\mathcal{G}_{bf}$, and $\epsilon$ satisfies*

$$\|\epsilon\|_\infty \le \frac{1}{2\|\theta^*\|_1}\left(d(\eta, g^{ds*}) - d(\eta, g^{(\theta^*)}) + n\left(KL(\vec{w}^*, \vec{w}^{ds}) + \sum_{j=1}^{p} \frac{n_j}{n} KL(\vec{b_j^*}, \vec{b_j^{ds}})\right)\right.$$

$$\left. + n\left(KL(Z^*, Z^{ds*}) - KL(Z^*, Z^{ds})\right)\right),$$

*then $g^{bf}$ is better than the one-coin DS prediction $g^{ds}$, i.e. $d(\eta, g^{bf}) \le d(\eta, g^{ds})$.*

*Proof.* Theorem 21 states

$$d(\eta, g^{bf}) \leq d(\eta, g^{(\theta^*)}) + 2\|\epsilon\|_\infty \|\theta^*\|_1,$$

bounding the total epistemic error of the BF prediction via the exact model uncertainty and an upper bound on the approximation uncertainty. From Lemma 33, we see that the model uncertainty for one-coin DS matches. Therefore, it suffices for our upper bound of BF's approximation uncertainty to be smaller than DS' approximation uncertainty. Written out, we require $\epsilon$ sufficiently small that the following inequality holds.

$$2\|\epsilon\|_\infty \|\theta^*\|_1 \leq \mathcal{E}_{ds}^{appr}$$

But, this just boils down to

$$\|\epsilon\|_\infty \leq \frac{1}{2\|\theta^*\|_1}\left( d(\eta, g^{ds*}) - d(\eta, g^{(\theta^*)}) + n\left( KL(\vec{w}^*||\vec{w}) + \sum_{j=1}^{p} \frac{n_j}{n} KL(\vec{b_j^*}||\vec{b_j}) \right.\right.$$

$$\left.\left. + KL(Z^*||Z^{ds*}) - KL(Z^*||Z^{ds}) \right) \right)$$

where we've expanded the OCDS approximation uncertainty via Lemma 32. $\qquad\square$

# N  OCDS WITH EM IS INCONSISTENT

We end the theoretical portion of the appendix by formally describing a class of problems where OCDS equipped with EM can easily be shown to be inconsistent. In essence, we compute the unique optimal approximator $g^*$ to $\eta$ (Theorem 27) and show that applying the M Step to it, and then applying the E Step will result in the OCDS prediction not equalling $g^*$. Recall that EM is said to converge when repeated applications of the E and M step bring no change to the prediction. Thus, if we give EM a prediction (namely $g^*$) and and the resulting prediction from the E-Step (after applying an M-Step) is different from $g^*$, then that means EM never converges to $g^*$.

Our class of problems will have $p = 2$ rules in the ensemble and $k = 2$ classes. Moreover, the rules will be generalists and predict on all $n$ datapoints. Observe that there are eights ways for the ensemble to predict and for the labels to be assigned (Table 4).

Table 4: All Rule Prediction/Ground Truth Label Combinations

| true label$\rightarrow$ | 1 | 2 | 1 | 2 | 1 | 2 | 1 | 2 |
|---|---|---|---|---|---|---|---|---|
| $h^{(1)}(x)$ | 1 | 1 | 2 | 2 | 1 | 1 | 2 | 2 |
| $h^{(2)}(x)$ | 1 | 1 | 2 | 2 | 2 | 2 | 1 | 1 |

Call $c_{11}$ the number of points (out of $n$) where both rules in the ensemble predict 1. Going by this, $c_{21}$ is the number of points where the first rule predicts class 2 while the second rule predicts class 1. Formally,

$$c_{11} = n\Pr_n(h^{(1)}(x) = 1, h^{(2)}(x) = 1) \quad \text{and} \quad c_{21} = n\Pr_n(h^{(1)}(x) = 2, h^{(2)}(x) = 1).$$

One quickly infers that

$$c_{11} + c_{22} + c_{12} + c_{21} = n.$$

Similarly, call $r_{1,12}$ the number of points where the true label is 1, and the first rule predicts class 1 while the second rule erroneously predicts class 2.

$$r_{1,12} = n\Pr_n(y = 1, h^{(1)}(x) = 1, h^{(2)}(x) = 2)$$

One sees that $r_{1,12} + r_{2,12} = c_{12}$. Moreover, $nb_1^* = r_{1,11} + r_{2,22} + r_{1,12} + r_{2,21}$ or writing out the probabilities only,

$$b_1^* = \Pr_n(h^{(1)}(x) = y) = \Pr_n(y = 1, h^{(1)}(x) = 1, h^{(2)}(x) = 1)$$

$$+ \Pr_n(y = 2, h^{(1)}(x) = 2, h^{(2)}(x) = 2) + \Pr_n(y = 1, h^{(1)}(x) = 1, h^{(2)}(x) = 2)$$

$$+ \Pr_n(y = 2, h^{(1)}(x) = 2, h^{(2)}(x) = 1).$$

Similarly for the class frequencies, $w_1^* = r_{1,11} + r_{1,22} + r_{1,12} + r_{1,21}$.

Suppose for datapoint $x_i$ that the rules both predict 1. Then,

$$\Pr_n(y = \ell \mid h^{(1)}(x) = 1, h^{(2)}(x) = 1) = \frac{\Pr_n(y = \ell, h^{(1)}(x) = 1, h^{(2)}(x) = 1)}{\Pr_n(h^{(1)}(x) = 1, h^{(2)}(x) = 1)} = \frac{r_{\ell,11}}{c_{11}}.$$

If a prediction (from BF/OCDS) predicts the above probability for label $\ell$ when the rules each predict 1, we'll say the prediction infers the correct proportions of labels for $c_{11}$. If the prediction can do this for all $c$ values, we'll just say it infers the correct proportion of labels. Formally, say $g'$ infers the correct proportion of labels. Suppose that for datapoint $x_i$, $h^{(1)}(x_i) = \ell'$ and $h^{(2)}(x_i) = \ell''$. When we say $g'$ infers the correct proportion of labels, we mean

$$g'_{i\ell} = \frac{r_{\ell,\ell'\ell''}}{c_{\ell'\ell''}} = \frac{\Pr_n\left(y = \ell, h^{(1)}(x_i) = \ell', h^{(2)}(x_i) = \ell''\right)}{\Pr_n\left(h^{(1)}(x_i) = \ell', h^{(2)}(x_i) = \ell''\right)}.$$

We now show that under a certain restriction on the $r$ and $c$ values, the BF prediction infers the correct proportion of labels.

**Lemma 36.** *Suppose for a dataset that $c_{11} = c_{22}$, $c_{12} = c_{21}$, $r_{1,11} = r_{2,22}$, $r_{1,12} = r_{2,21}$, and $r_{2,12} = r_{1,21}$. Then, class frequencies for each class are equal. Now, suppose we have an ensemble of two rules that predict on each of the $n$ datapoints. If BF is given the empirical accuracies and empirical class frequencies, it correctly infers the correct proportion of labels. Furthermore, a set of weights that are optimal for BF are as follows. The class frequency weights are both equal to $n$. The learner's weights for rules $h^{(1)}$ and $h^{(2)}$ are*

$$n \log\left(\sqrt{\frac{r_{1,11}}{r_{2,11}}\frac{r_{1,12}}{r_{2,12}}}\right) \quad \text{and} \quad n \log\left(\sqrt{\frac{r_{1,11}}{r_{2,11}}\frac{r_{2,12}}{r_{1,12}}}\right)$$

*respectively.*

*Proof.* We now write out the dual problem for BF. We will make the following simplification: defining real variables representing $\sigma' - \sigma$. Let $n\theta_1$ and $n\theta_2$ represent the weights for the rules, while $n\tau_1$ and $n\tau_2$ represent the weights for the class frequencies. They are allowed to be real by definition. We include the $n$ explicitly on the outside so that the log-sum-exp term in the BF dual does not contain $n$. (Each row of $A$ in this case is defined by $h^{(j)}/n$ or $\vec{e}_\ell^{\,n}/n$, so we want to get rid of the $n$.) Expanding the terms from Theorem 2, our objective is

$$\max_{\theta_1,\theta_2,\tau_1,\tau_2} [nb_1\theta_1 + nb_2\theta_2 + nw_1\tau_1 + nw_2\tau_2 - c_{11}\log\left(\exp\left(\tau_1 + \theta_1 + \theta_2\right) + \exp\left(\tau_2\right)\right)$$
$$- c_{22}\log\left(\exp\left(\tau_1\right) + \exp\left(\tau_2 + \theta_1 + \theta_2\right)\right)$$
$$- c_{12}\log\left(\exp\left(\tau_1 + \theta_1\right) + \exp\left(\tau_2 + \theta_2\right)\right)$$
$$- c_{21}\log\left(\exp\left(\tau_1 + \theta_2\right) + \exp\left(\tau_2 + \theta_1\right)\right)].$$

Note that the term inside the logarithm is the sum of all the exponential of all possible predictions given that the ensemble predicts in a certain way. For example, $c_{11}$ is the case where the rules both predict label 1. Therefore, the two possible predictions are $\tau_1 + \theta_1 + \theta_2$ for class 1 and $\tau_2$ for class 2. Since this objective is convex, we can take the partial derivatives with respect to each variable and exhibit $\theta_1, \theta_2, \tau_1, \tau_2$ such that the partial derivatives are 0. The partial derivatives set to 0 are

$$nb_1 = c_{11}\frac{e^{\tau_1+\theta_1+\theta_2}}{e^{\tau_1+\theta_1+\theta_2}+e^{\tau_2}} + c_{22}\frac{e^{\tau_2+\theta_1+\theta_2}}{e^{\tau_2+\theta_1+\theta_2}+e^{\tau_1}} + c_{12}\frac{e^{\tau_1+\theta_1}}{e^{\tau_1+\theta_1}+e^{\tau_2+\theta_2}} + c_{21}\frac{e^{\tau_2+\theta_1}}{e^{\tau_2+\theta_1}+e^{\tau_1+\theta_2}}$$

$$nb_2 = c_{11}\frac{e^{\tau_1+\theta_1+\theta_2}}{e^{\tau_1+\theta_1+\theta_2}+e^{\tau_2}} + c_{22}\frac{e^{\tau_2+\theta_1+\theta_2}}{e^{\tau_2+\theta_1+\theta_2}+e^{\tau_1}} + c_{12}\frac{e^{\tau_2+\theta_2}}{e^{\tau_2+\theta_2}+e^{\tau_1+\theta_1}} + c_{21}\frac{e^{\tau_1+\theta_2}}{e^{\tau_1+\theta_2}+e^{\tau_2+\theta_1}}$$

$$nw_1 = c_{11}\frac{e^{\tau_1+\theta_1+\theta_2}}{e^{\tau_1+\theta_1+\theta_2}+e^{\tau_2}} + c_{22}\frac{e^{\tau_1}}{e^{\tau_1}+e^{\tau_2+\theta_1+\theta_2}} + c_{12}\frac{e^{\tau_1+\theta_1}}{e^{\tau_1+\theta_1}+e^{\tau_2+\theta_2}} + c_{21}\frac{e^{\tau_1+\theta_2}}{e^{\tau_1+\theta_2}+e^{\tau_2+\theta_1}}$$

$$nw_2 = c_{11}\frac{e^{\tau_2}}{e^{\tau_2}+e^{\tau_1+\theta_1+\theta_2}} + c_{22}\frac{e^{\tau_2+\theta_1+\theta_2}}{e^{\tau_2+\theta_1+\theta_2}+e^{\tau_1}} + c_{12}\frac{e^{\tau_2+\theta_2}}{e^{\tau_2+\theta_2}+e^{\tau_1+\theta_1}} + c_{21}\frac{e^{\tau_2+\theta_1}}{e^{\tau_2+\theta_1}+e^{\tau_1+\theta_2}}$$

Now, choose $\tau_1 = \tau_2 = 1$ and

$$\theta_1 = \log\left(\sqrt{\frac{r_{1,11}}{r_{2,11}}\frac{r_{1,12}}{r_{2,12}}}\right) \quad \text{and} \quad \theta_2 = \log\left(\sqrt{\frac{r_{1,11}}{r_{2,11}}\frac{r_{2,12}}{r_{1,12}}}\right).$$

By our choice of $\tau$'s, we can disregard the appearance of $\tau_1$ and $\tau_2$ in the softmaxes. Observe that the accuracies and class frequencies can be written in terms of the $r$ terms.

$$nb_1 = r_{1,11} + r_{2,22} + r_{1,12} + r_{2,21}, \quad nb_2 = r_{1,11} + r_{2,22} + r_{2,12} + r_{1,21}$$

$$nw_1 = r_{1,11} + r_{1,22} + r_{1,12} + r_{1,21}, \quad \text{and} \quad nw_2 = r_{2,11} + r_{2,22} + r_{2,12} + r_{2,21}.$$

We briefly digress to show that $nw_1 = nw_2$. Since

$$c_{11} = c_{22}, \quad \text{it follows that} \quad r_{1,11} + r_{2,11} = r_{1,22} + r_{2,22}.$$

As we have assumed $r_{1,11} = r_{2,22}$, we see the above equality implies $r_{2,11} = r_{1,22}$. Now by our assumptions of equality of the $r$ terms, $nw_1 = nw_2$.

If we can establish correspondences between the softmaxes and those equations involving the $r$'s using our claimed weights, that means those weights are optimal and the learner's optimal prediction infers the correct proportions of labels. (The learner's predictions are exactly those softmaxes.) For the first partial derivative,

$$c_{11} \frac{e^{\theta_1 + \theta_2}}{e^{\theta_1 + \theta_2} + 1} = c_{11} \frac{e^{\log\left(\frac{r_{1,11}}{r_{2,11}}\right)}}{e^{\log\left(\frac{r_{1,11}}{r_{2,11}}\right)} + 1} = c_{11} \frac{r_{1,11}}{r_{1,11} + r_{2,11}} = r_{1,11}$$

because $r_{1,11} + r_{2,11} = c_{11}$. The softmax term for $c_{22}$ works out the same way. Recall that by assumption that $r_{1,11} = r_{2,22}$, the softmax equals $r_{2,22}/c_{22}$ because we also assumed that $c_{11} = c_{22}$. For the last two terms in the first partial derivative, one can see that

$$c_{12} \frac{e^{\theta_1}}{e^{\theta_1} + e^{\theta_2}} = c_{12} \frac{\sqrt{\frac{r_{1,11}}{r_{2,11}} \frac{r_{1,12}}{r_{2,12}}}}{\sqrt{\frac{r_{1,11}}{r_{2,11}} \frac{r_{1,12}}{r_{2,12}}} + \sqrt{\frac{r_{1,11}}{r_{2,11}} \frac{r_{2,12}}{r_{1,12}}}} = c_{12} \frac{r_{1,12}}{c_{12}} = c_{21} \frac{r_{2,21}}{c_{21}}$$

where for the second to last equality, we used the fact that $c_{12} = r_{1,12} + r_{2,12}$ whereas the last equality used our assumptions that $r_{1,12=r_{2,21}}$ and $c_{12} = c_{21}$. These calculations show that the first partial derivative is $0$ with respect to the weights we have chosen. In fact, all of the partial derivatives evaluate to zero when our weights are used.

To summarize, we chose weights of $n$ for each of the class frequency constraints, and we chose

$$n \log \left( \sqrt{\frac{r_{1,11}}{r_{2,11}} \frac{r_{1,12}}{r_{2,12}}} \right)$$

for the first rule's constraint, and

$$n \log \left( \sqrt{\frac{r_{1,11}}{r_{2,11}} \frac{r_{2,12}}{r_{1,12}}} \right)$$

for the second rule's constraint. $\qquad\square$

Because we gave the empirical accuracies and class frequencies to BF, its prediction is also the best approximator. Since the optimal approximator $g^*$ is such that $Ag^* = b^*$, the M-step of OCDS' EM algorithm (Algorithm 1) will return the empirical accuracies and class frequencies. Then, the E-Step will return OCDS predictions using the aforementioned empirical quantities. To see that the resulting prediction from OCDS after an M-Step and E-Step is not optimal (doesn't necessarily match BF's prediction), observe that the OCDS prediction for class 1 on a datapoint where both rules predict label 1 is

$$\frac{w_1^* b_1^* b_2^*}{w_1^* b_1^* b_2^* + w_2^* (1 - b_1^*)(1 - b_2^*)}, \quad \text{not necessarily equal to} \quad \frac{r_{1,11}}{r_{1,11} + r_{2,11}},$$

which is the optimal prediction.

For completeness, following Lemma 30, the OCDS weights are

$$n \log (w_1^*) \quad \text{and} \quad n \log (w_2^*)$$

for the class frequencies while assigning

$$n \log \left( \frac{b_1^* (k - 1)}{1 - b_1^*} \right) \quad \text{and} \quad n \log \left( \frac{b_2^* (k - 1)}{1 - b_2^*} \right)$$

for the rules.

### N.1 A CONCRETE EXAMPLE

We now instantiate an instance of the above problem and show that one does not get $g^*$ after applying one M step followed by one E step to $g^*$. This means that EM does not converge at $g^*$.

Define $w_1^* = w_2^* = 0.5$ and say we have $n = 22$ datapoints. Also, say

$$r_{1,11} = r_{2,22} = 5, \quad r_{1,12} = r_{2,21} = 3, \quad r_{2,11} = r_{1,22} = 2, \quad r_{2,12} = r_{1,21} = 1,$$

which means that

$$c_{11} = c_{22} = 7, \quad c_{12} = c_{21} = 4, \quad w_1^* = w_2^* = \frac{11}{22} = 0.5, \quad b_1^* = \frac{16}{22}, \quad b_2^* = \frac{12}{22}.$$

The previous section shows that $g^*$ computed from $V(b^*, \vec{0}_4)$ (we're abusing notation so that $b^*$ has both the empirical rule accuracies and class frequencies) is such that $Ag^* = b^*$. Note that $g^*$ is unique (Theorem 27) so that we only have to consider what we computed from $V(b^*, \vec{0}_4)$. This means that the M step with $g^*$ will return $b^*$. Now, we check the result of the E step. We compute one prediction from the E step, and omit the others for brevity. With the empirical accuracies and empirical class frequencies (from the M step), DS predicts that the probability that the label is 1 when both rules predict 1 (which has actual value $r_{1,11}/c_{11}$) is

$$\frac{w_1^* b_1^* b_2^*}{w_1^* b_1^* b_2^* + w_2^*(1 - b_1^*)(1 - b_2^*)} = \frac{\frac{1}{2}\frac{16}{22}\frac{12}{22}}{\frac{1}{2}\frac{16}{22}\frac{12}{22} + \frac{1}{2}\frac{6}{22}\frac{10}{22}} = \frac{1}{1 + \frac{6 \cdot 10}{16 \cdot 12}} \approx 0.76.$$

Since there are $c_{11} = 7$ points where both rules predict 1, DS predicts that $0.76 \cdot 7 = 5.33$ out of 7 many points have label 1. This is in contrast to the optimal approximator, which correctly predicts that 5 out of 7 points have label 1 (when both rules predict 1).

The following table shows the optimal approximator's predictions, and DS' predictions for the $r$ values. It suffices to note that the DS predictions do not match the optimal approximator. Therefore, EM cannot converge at $g^*$ for this problem

Table 5: Predicted $r$ values from BF and OCDS

| Pred. | $r_{1,11}$ | $r_{2,11}$ | $r_{1,22}$ | $r_{2,22}$ | $r_{1,12}$ | $r_{2,12}$ | $r_{1,21}$ | $r_{2,21}$ |
|---|---|---|---|---|---|---|---|---|
| $g^*$ | 5 | 2 | 2 | 5 | 1 | 3 | 1 | 3 |
| $g^{ds*}$ | 5.33 | 1.67 | 1.67 | 5.33 | 1.71 | 2.29 | 1.71 | 2.29 |

as applying an M step followed by an E step results in the prediction moving away from $g^*$. We conclude that DS is inconsistent.

## O  EXPERIMENTAL RESULTS

To end the appendix, we now present the complete experimental results. The results of the main paper are presented again, but with more detailed commentary. The loss decomposition for every dataset is also shown.

### O.1  DATASETS

The first citation is the source of the raw data, while the second citation is the source of the rules-of-thumb. We note that many of the datasets and rules have been compiled in the WRENCH repository [Zhang et al., 2021]. If the validation sets were not provided, we construct it by doing a random split on the training set. The resulting size of the training/validation sets are chosen to match the cited author's choices. The datasets are also included in the supplementary material.

1. Animals with Attributes (AwA) [Xian et al., 2019], [Mazzetto et al., 2021]. We wish to distinguish images of animals. Two classes (chimpanzee and seal) are selected to create a binary task. The rules are created by fine-tuning a pretrained ResNet-18 using labeled data from other classes.

2. Basketball [Fu et al., 2020] for both data and rules. We wish to identify which videos are about basketball in a set of videos. The rules are heuristics that are given information such as presence of certain objects, their size, and distance from each other.

3. Breast Cancer [Wolberg et al., 1995], [Arachie and Huang, 2018]. We wish to determine whether breast cell nuclei are positive or negative for breast cancer. The rules are single dimension logistic regression classifiers. The features chosen are mean radius of the nucleus, radius standard error, and worst radius of the cell nucleus.

4. Cardiotocography [de Campos and Bernardes, 2010], [Arachie and Huang, 2018]. We wish to use cardiotocograms to predict fetal heart rate. Out of the ten possible classes, the two most common are selected. The rules are single dimension logistic regression classifiers. The features chosen are accelerations per second, mean value of long-term variability, and histogram median.

5. DomainNet [Peng et al., 2019], [Mazzetto et al., 2021]. We wish to classify images that come from different domains. 5 classes are randomly selected from the 25 classes that appear most frequently. To generate the rules, a pre-trained ResNet-18 network is fine-tuned using 60 labeled examples from each domain.

6. IMDB [Maas et al., 2011], [Ren et al., 2020]. We wish to determine whether or not an IMDB review is positive or negative. The rules look for specific keywords or patterns present in the text.

7. OBS Network [Rajab, 2017], [Arachie and Huang, 2018]. We wish to detect and block network nodes which may have potentially malicious behavior. The rules are single dimension logistic regression classifiers. The features chosen are percentage flood per node, average packet drop rate, and utilized bandwidth.

8. SMS [Almeida and Hidalgo, 2012], [Awasthi et al., 2020]. We wish to determine whether a text message (SMS) is spam or not. Rules check for specific keywords or phrases.

9. Yelp [Zhang et al., 2015], [Ren et al., 2020]. We wish to classify whether a Yelp review is positive or not. The rules look for specific keywords or patterns present in the text.

10. Youtube [Alberto and Lochter, 2017], [Snorkel AI Inc., 2022]. We wish to determine whether comments on five youtube videos were spam or not. The rules check for certain keywords to determine if the comment is spam.

For datasets where the rules predict probabilities, we convert the probabilities to hard predictions by taking the label with the highest probability. Also, for methods that cannot handle abstaining rules, we have a version of the dataset where a random label replaces a rule abstention. This "filled out" dataset is only used for the methods that can't handle rule abstentions.

Table 6 summarizes some relevant statistics.

Table 6: Dataset Statistics

| Name | # Class ($k$) | # Rules ($p$) | # Train ($n$) | # Valid | Rules Abstain? |
|------|------|------|------|------|------|
| AwA | 2 | 36 | 1372 | 172 | N |
| Basketball | 2 | 4 | 17970 | 1064 | Y |
| Cancer | 2 | 3 | 171 | 227 | N |
| Cardio | 2 | 3 | 289 | 385 | N |
| Domain | 5 | 5 | 2587 | 323 | N |
| IMDB | 2 | 8 | 20000 | 2500 | Y |
| OBS | 2 | 3 | 239 | 317 | N |
| SMS | 2 | 73 | 4571 | 500 | Y |
| Yelp | 2 | 8 | 30400 | 3800 | Y |
| Youtube | 2 | 10 | 1586 | 120 | Y |

## O.2 METHODS

We consider a seven methods that serve as representatives for various label estimation strategies used in weak supervision. They are briefly described here, along with relevant details about initialization. See the attached code for the complete implementation of each method.

1. Majority Vote (MV), but we normalize the counts for each datapoint so a distribution is predicted. Namely, its prediction $g_{i\ell}^{mv}$ is

$$g_{i\ell}^{mv} := \frac{|\{j \in [p] \colon h^{(j)}(x_i) = \ell\}|}{|\{j \in [p] \colon h^{(j)}(x_i) \neq \, ?\}|}$$

2. Dawid-Skene, one-coin variant (OCDS) equipped with EM [Dawid and Skene, 1979], [Li and Yu, 2014]. Our nominal representative for the probabilistic approach, described in Subsection B.2. Assumes conditional independence of rule predictions given the label. EM is initialized with the majority vote prediction and is run until convergence.

3. Data Programming (DP) [Ratner et al., 2016], a generalization of DS that relaxes the independence assumptions made by use of a factor graph. Run with the default hyperparameters provided in the WRENCH implementation [Zhang et al., 2021].

4. Enhanced Bayesian Classifier Combination (EBCC) [Li et al., 2019], a Bayesian method that generalizes DS, interpretable as having multiple confusion matrices per rule. Posterior label distribution is estimated via mean field variational inference with suggested hyperparameters from aforementioned authors.

5. Hyper Label Model (HyperLM) [Wu et al., 2023], a graph neural network that predicts a label given rule predictions.

6. Adversarial Multi-Class Learning, Convex Combination variant (AMCL CC) [Mazzetto et al., 2021], another adversarial weak supervision method. The scaling factor is set to $0.4$. We note that this method cannot handle rules that abstain. Thus, if any rule abstains on a datapoint, "fill-in" that rule's prediction by selecting a label uniformly at random. This method requires a linear program solver, for which we use Gurobi [Gurobi Optimization, 2021].

7. Balsubramani-Freund with log-loss and accuracy constraints (BF), our method in consideration, implemented with CVXPY [Diamond and Boyd, 2016], run with MOSEK [MOSEK ApS, 2022], an off the shelf convex solver.

DP, EBCC, AMCL CC, and BF each have sources of randomness, so we run each mentioned method 10 times. DP's and EBCC's randomness comes from the intialization, while AMCL CC and BF's randomness comes from the fact that those two methods use labeled data. For the table, 100 labeled points are randomly drawn from the validation set. Wilson's interval with failure probability $0.05$ (as suggested by Brown et al. [2001]) is used to bound the rule accuracies and class frequencies for BF.

The mean log loss/0-1 loss/Brier score is reported for each method on each dataset. Bolded entries ones where the respective means could not be distinguished by a paired two tailed t-test with $p = 0.05$.

## O.3 COMPARISON WITH SOTA

We consider three losses for each method. Let the ground truth be $\eta$ and the prediction from some fixed method be $g$. For the experiments, the labels will be deterministic. If the label for datapoint $x_i$, denoted $y_i$ is $\ell$, then $\eta_{i\ell} = 1$.

1. Log-loss: $\frac{1}{n} \sum_{i=1}^{n} \sum_{\ell=1}^{k} -\eta_{i\ell} \log g_{i\ell}$
2. 0-1 loss: $\frac{1}{n} \sum_{i=1}^{n} \mathbf{1}(\arg\max_{\ell \in [k]} g_{i\ell} = y_i)$
3. Brier Score: $\frac{1}{n} \sum_{i=1}^{n} \sum_{\ell=1}^{k} (g_{i\ell} - \eta_{i\ell})^2$

In Table 7, we see that BF does very well when judged by its prediction's log loss. This may not be surprising because that is its the minimax game objective. Even when BF does not have the best log-loss (compared to other methods on Yelp, Youtube), its loss was very close to HyperLM's.

Now, the model uncertainty in the last row $\frac{1}{n} d(\eta, g^*)$ allows us to determine how much of BF's loss is from its approximation uncertainty. (Recall $g^*$ is gotten by solving the BF dual with empirical rule accuracies/class frequencies, Theorem 21.) In other words, how much of BF's error is theoretically reducible without getting more rules of thumb? For almost all of these datasets considered, the answer is not very much (in terms of log loss). We see that the BF log loss with 100 labels is decently close to the lowest possible log loss BF can attain with the rules-of-thumb used. In the loss visualizations below, one can see that for a lot of the datasets, BF is limited by the lack of rules available to it. Thus, the presence of additional labeled data (up to 300 labeled points) does not bring a big gain in performance. Also in those graphs is the breakdown of DS error. That will be discussed when the graphs are presented.

Table 8 shows that the BF prediction is pretty good when measured with 0-1 loss and Brier score. While it is less dominant, it still performs very well – being the method that has the best result on the largest number of datasets. When it does not

Table 7: Comparison of BF Against Other WS Methods Using Average Log Loss

| Method | AwA | Basketball | Cancer | Cardio | Domain | IMDB | OBS | SMS | Yelp | Youtube |
|---|---|---|---|---|---|---|---|---|---|---|
| MV | 0.31 | 2.40 | 14.87 | 0.66 | 5.48 | 6.39 | 8.73 | 0.79 | 5.90 | 1.27 |
| OCDS | 0.24 | 3.75 | 4.46 | 13.74 | 22.32 | 2.91 | 6.28 | 0.78 | 1.73 | 17.63 |
| DP | 0.42 | 1.31 | 6.14 | 7.01 | 9.21 | 0.68 | 3.98 | 0.53 | 2.61 | 0.72 |
| EBCC | **0.13** | 0.45 | 4.25 | 0.90 | 1.80 | 0.73 | 2.23 | 0.43 | 0.81 | 0.69 |
| HyperLM | 0.21 | 1.31 | 6.93 | 0.60 | 1.29 | 0.62 | 2.66 | 0.68 | **0.60** | **0.42** |
| AMCL CC | **0.14** | 1.26 | 14.86 | 0.42 | 5.42 | 1.46 | 8.73 | 0.69 | 0.85 | 0.70 |
| BF | **0.13** | **0.39** | **0.68** | **0.20** | **1.12** | **0.59** | **0.61** | **0.42** | 0.64 | 0.50 |
| $\frac{1}{n}d(\eta, g^*)$ | 0.01 | 0.32 | 0.65 | 0.13 | 1.01 | 0.57 | 0.59 | 0.25 | 0.54 | 0.21 |

Table 8: Comparison of BF Against Other WS Methods Using Average 0-1 Loss and Average Brier Score

| Method | AwA | | Basketball | | Cancer | | Cardio | | Domain | | IMDB | | OBS | | SMS | | Yelp | | Youtube | |
|---|---|---|---|---|---|---|---|---|---|---|---|---|---|---|---|---|---|---|---|---|
| | 0-1 | BS | 0-1 | BS | 0-1 | BS | 0-1 | BS | 0-1 | BS | 0-1 | BS | 0-1 | BS | 0-1 | BS | 0-1 | BS | 0-1 | BS |
| MV | **1.31** | 0.15 | 24.54 | 0.31 | 52.05 | 0.95 | 34.95 | 0.35 | 45.73 | 0.62 | 29.40 | 0.47 | **27.62** | 0.54 | 31.92 | 0.32 | **31.84** | 0.49 | **18.79** | **0.23** |
| OCDS | 2.11 | 0.04 | **11.29** | **0.23** | 52.05 | 1.02 | 39.79 | 0.80 | 80.17 | 1.60 | 49.81 | 0.95 | **27.62** | 0.55 | 9.67 | **0.18** | 46.74 | 0.72 | 52.40 | 1.05 |
| DP | 3.15 | 0.06 | **11.29** | **0.23** | 50.88 | 1.01 | 39.79 | 0.80 | 72.51 | 1.36 | 30.48 | 0.45 | **27.62** | 0.55 | 32.19 | 0.36 | 46.78 | 0.71 | 34.75 | 0.40 |
| EBCC | **1.57** | **0.03** | 36.33 | 0.29 | 52.05 | 1.03 | 39.79 | 0.62 | 48.23 | 0.74 | 28.26 | 0.45 | **27.62** | 0.55 | **8.16** | 0.25 | 36.02 | 0.51 | 52.40 | 0.50 |
| HyperLM | 2.55 | 0.10 | 36.36 | 0.45 | 52.05 | 0.94 | 7.96 | 0.31 | 41.98 | 0.65 | **27.74** | 0.41 | **27.62** | 0.45 | 53.73 | 0.50 | 32.92 | **0.41** | 20.37 | 0.26 |
| AMCL CC | 2.00 | 0.06 | 12.14 | **0.23** | 49.18 | 0.93 | **3.11** | **0.06** | 36.82 | 0.54 | 31.74 | 0.46 | **27.62** | 0.54 | 45.04 | 0.49 | 37.39 | 0.48 | 38.88 | 0.47 |
| BF | 3.67 | 0.06 | **11.40** | 0.22 | **40.47** | **0.49** | 3.11 | 0.08 | 36.75 | 0.55 | 29.33 | **0.41** | **27.62** | **0.42** | 13.50 | 0.25 | **34.42** | 0.45 | 24.34 | 0.33 |
| $g^*$ | 0.58 | 0.01 | 11.27 | 0.19 | 36.26 | 0.46 | 3.11 | 0.06 | 37.26 | 0.51 | 28.74 | 0.38 | 27.62 | 0.40 | 8.09 | 0.14 | 26.54 | 0.36 | 7.31 | 0.12 |

have the best result, it is competitive with the other methods shown. Like for log-loss, we are also able to evaluate the best approximator $g^*$ to $\eta$ on these losses. Note that $g^*$ is the best approximator in terms of KL divergence. Except for Domain with 0-1 Loss and IMDB with 0-1 Loss, $g^*$ had loss no bigger (and often smaller) than even the best methods. This shows that the prediction gotten from BF with log loss is good even when evaluated under other losses.

## O.4 CONSISTENCY

To demonstrate the consistency of BF, we show that it is consistent under the DS generative assumption. For us, a method being consistent means it can attain 0 approximation uncertainty for every problem. And specifically, BF produces a prediction that has 0 approximation uncertainty when it is given $b^*$. In the literature, consistency can mean the ability of a method to infer the underlying generative distribution as the number of (unlabeled) datapoints $n \to \infty$. When BF is used in that setting, it will also infer the underlying generative distribution.

We will consider the one-coin BF model, with rule accuracy and class frequency constraints. The data will be generated under the one-coin DS assumption with $k = 2$ classes, $p = 3$ rules and $n$ datapoints. Our label space will be $\{-1, 1\}$ for convenience, and distributions over two elements will be other those labels.

1. Draw the underlying label distribution $w^\star \sim Dirichlet(1, 1)$.
2. For each $j \in [p]$, draw underlying accuracy $b_j^\star \sim Beta(2, 4/3)$.
3. For each $i \in [n]$:
   (a) Draw label $y_i \sim Categorical(w)$
   (b) For each $j \in [p]$, draw rule $j$'s prediction, $y_i(-1)^s, s \sim Bernoulli(b_j)$.

If we fix $n$, we can compute the empirical class frequencies and rule accuracies, $w^*$ and $b^*$ respectively. Those are the quantities given to BF. To simulate the case where one gets more data generated by the same underlying distribution, we generate a total of $n = 10^5$ datapoints, and give BF the first $10^2, 10^3, \ldots$ datapoints. A total of 10 datasets are generated via this process and the resulting KL divergence between the BF prediction and the underlying distribution is averaged.

The underlying label distribution $\eta$ in this case can be represented easily. If we fix the underlying label distribution and the underlying rule accuracies, then for any set of rule predictions, the underlying distribution $\eta$ is in the form of the RHS of

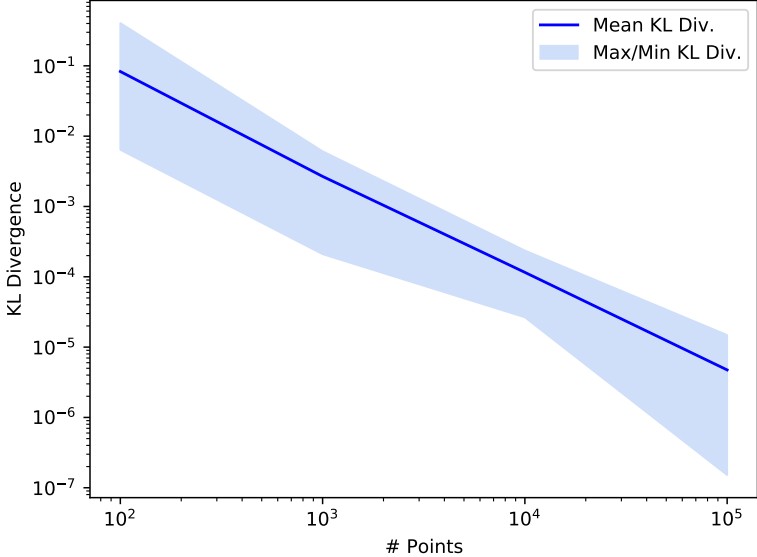

Figure 6: Synthetic Data Convergence

Equation 8. By how the rule accuracies/class frequencies are generated, one can easily show that $\eta \in \mathcal{G}_{bf}$ (Lemma 29). We measure the KL divergence of the BF prediction when it gets $10^2, 10^3, \ldots$ datapoints to the underlying label distribution $\eta$ for those $10^2, 10^3 \ldots$ datapoints.

The figure shown in the main paper is reproduced here (Figure 6). Once again, the graph is on a log-log scale and the divergence between the BF prediction and the underlying label distribution ($\frac{1}{n} d(\eta, g^*)$) decreases exponentially fast. This means that $g^* \to \eta$ as $n \to \infty$.

## O.5  ERROR DECOMPOSITION VISUALIZATION

To finish the appendix, we present the error decomposition visualizations for all ten datasets. We want to remind the reader that even though we write $d(\mu, \nu)$, what we have plotted is $\frac{1}{n} d(\mu, \nu)$. Practically speaking, this makes no difference to us as the vertical axis values have just been scaled. Because of this, the values on the vertical axes match the values in Table 7. If the OCDS loss $d(\eta, g^{ds})$ is not shown, it is because the value is so large that it would otherwise compress all the other values displayed.

Recall that we are framing the discussion on approximation uncertainty in terms of how it changes as a function of how well the empirical parameters are estimated. Specifically for our experiments, we are concerned with how well the empirical rule accuracies and class frequencies are estimated. For (one-coin) BF, these estimates are given directly to BF while for OCDS, those quantities are estimated via its EM algorithm.

For a fixed quantity of labeled points, we randomly sample that quantity 10 times from the validation set for BF. The resulting minimum/average/maximum approximation uncertainties are plotted.

We will take $\eta$ to be the ground truth labeling. To compute the BF loss decomposition, we take advantage of Theorem 6, which says that if we give BF the empirical parameters, we get the best approximator $g^*$ for $\eta$. Calling the BF prediction $g^{bf}$, we know $d(\eta, g^{bf})$ and $d(\eta, g^*)$. By Lemma 5, those two quantities are enough to compute $d(g^*, g^{bf})$. We point out that the blue section (the minimum BF approximation uncertainty $d(g^*, g^{bf})$) often doesn't decrease because the rule accuracies/class frequency on the labeled dataset (the validation set) does not in general equal the rule accuracies/class frequencies on the training set (which is what we're labeling and measuring loss on). For OCDS, we are interested in $\mathcal{E}_{ds,1}^{appr} = d(\eta, g^{ds*}) - d(\eta, g^*)$ and $\mathcal{E}_{ds,2}^{appr} = d(\eta, g^{ds}) - d(\eta, g^{ds*})$. These two quantities taken together represent the OCDS approximation uncertainty, but only the latter depends on how well EM estimates the empirical rule accuracies and class frequencies. We plot lines such that the gap between the lines denotes a specific contribution of loss. The (green) gap between the horizontal axis and the solid black line is the model uncertainty. The gap between the solid and dashed lines

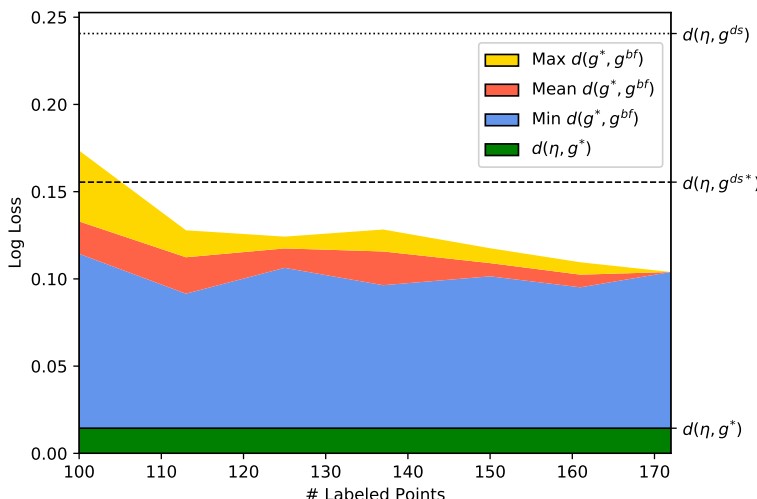

Figure 7: AwA BF/OCDS Loss Decomposition

is $\mathcal{E}_{ds,1}^{appr} + d(\eta, g^*) = d(\eta, g^{ds*})$, the loss OCDS has if EM perfectly estimated the empirical rule accuracies and class frequencies. (Recall that $g^{ds*}$ is the OCDS prediction gotten from doing one E-Step with the empirical parameters.) The gap between the dashed and dotted lines is $\mathcal{E}_{ds,2}^{appr} = d(\eta, g^{ds}) - d(\eta, g^{ds*})$, the loss incurred from imperfect estimation of the empirical rule accuracies/class frequencies by EM. We remind the reader that while $\mathcal{E}_{ds,2}^{appr}$ can be negative, that was not observed in our experiments.

To be terse, we will reference the figure number after the dataset and will not explicitly say "Figure". For Cancer (9), Cardio (10), IMDB (12), SMS (14), and Yelp (15), $\mathcal{E}_{ds,1}^{appr}$ is very low as the dashed line ($d(\eta, g^{ds*})$) is not far above the solid line $d(\eta, g^*)$. Thus, the reducible portion of OCDS' error is mainly from EM's failure to estimate the empirical rule accuracies/class frequencies well. For AwA (7), Basketball (8), and OBS (13), EM perfectly estimating the empirical rule accuracies/class frequencies would give a loss close to the BF loss $d(\eta, g^{bf})$ because $\mathcal{E}_{ds,1}^{appr}$ is so large. Domain (11) and Youtube (16) are the last two datasets and is the in-between case.

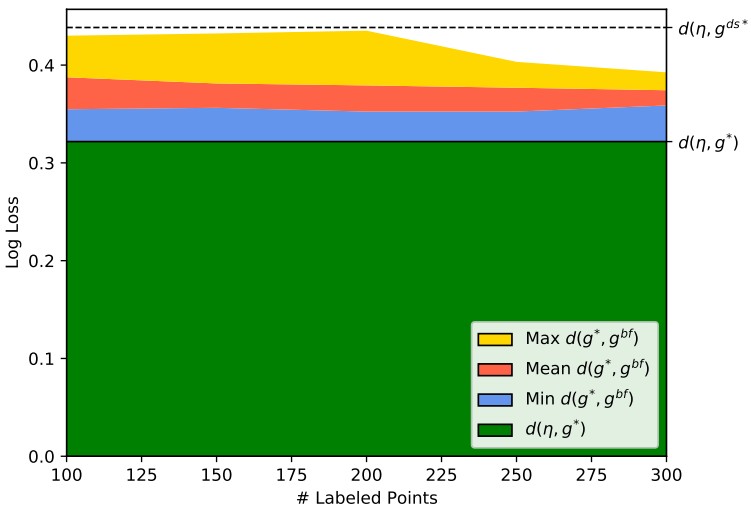

Figure 8: Basketball BF/OCDS Loss Decomposition

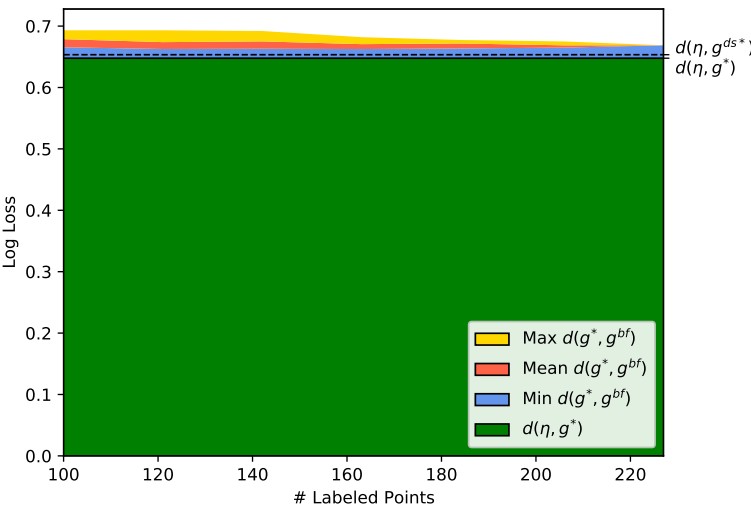

Figure 9: Cancer BF/OCDS Loss Decomposition

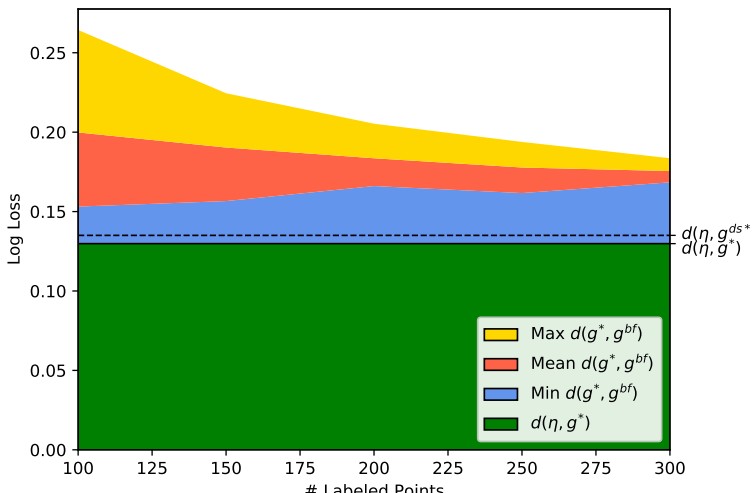

Figure 10: Cardio BF/OCDS Loss Decomposition

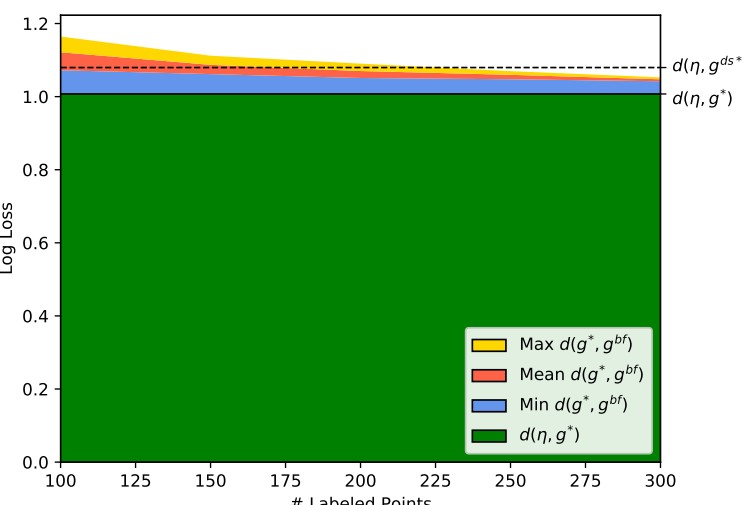

Figure 11: Domain BF/OCDS Loss Decomposition

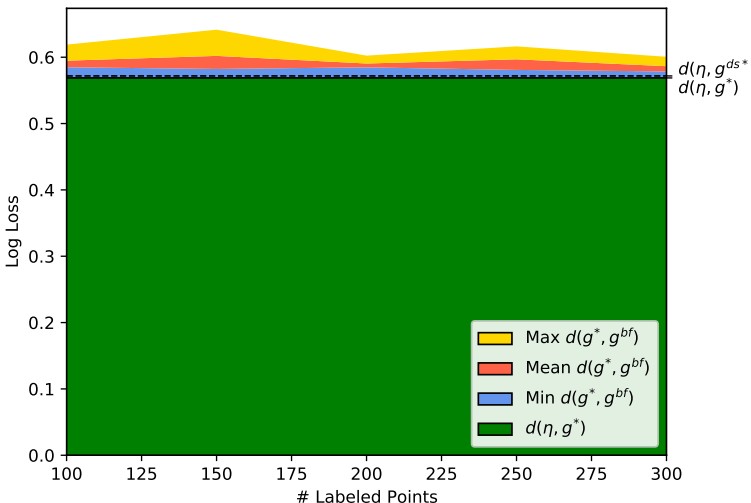

Figure 12: IMDB BF/OCDS Loss Decomposition

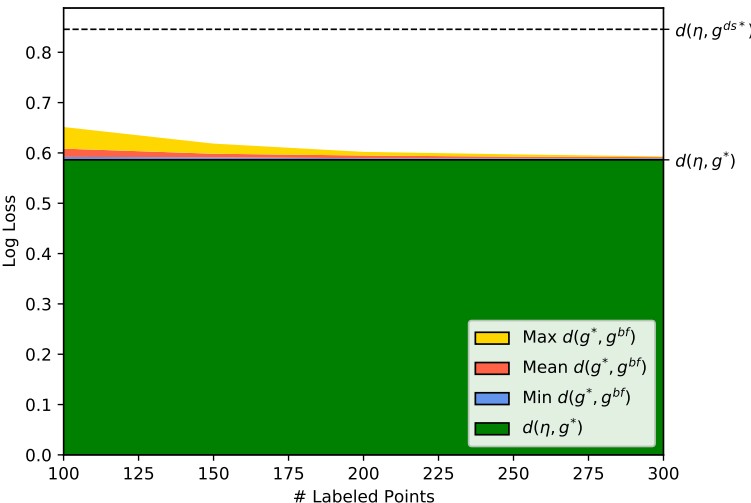

Figure 13: OBS BF/OCDS Loss Decomposition

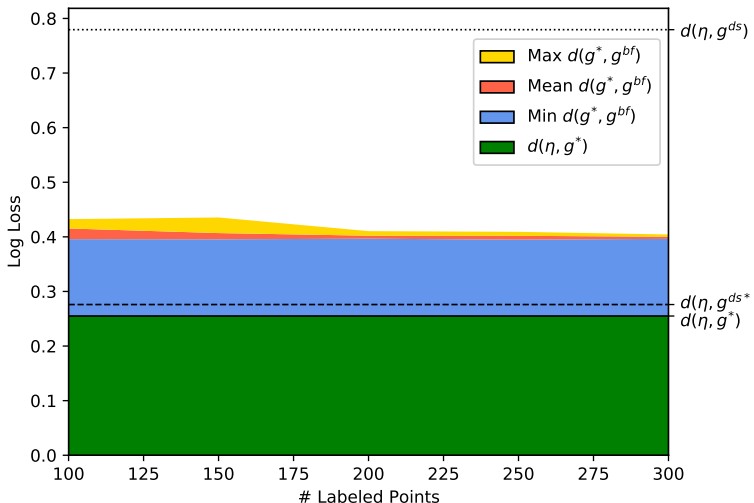

Figure 14: SMS BF/OCDS Loss Decomposition

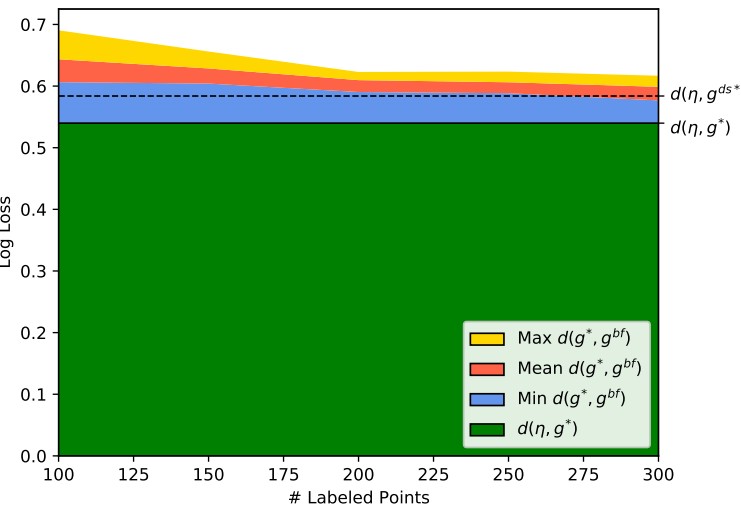

Figure 15: Yelp BF/OCDS Loss Decomposition

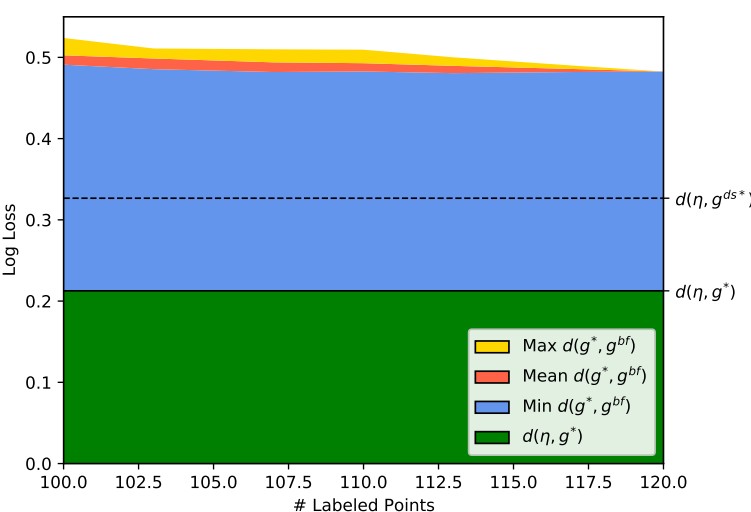

Figure 16: Youtube BF/OCDS Loss Decomposition