# OpenReview forum: "Convergence Behavior of an Adversarial Weak Supervision Method"
_auai.org/UAI/2024/Conference — UAI 2024 spotlight_

### Official Review · Reviewer_8jRK · 2024-03-20

**Q2-1 Originality-Novelty:** 3
**Q2-2 Correctness-Technical Quality:** 3
**Q2-5 Clarity Of Writing:** 3

**Q1 Summary And Contributions:**

The motivation of this paper is to explore the comparison between the Bayesian Rule Ensemble Method (BF) and Consistent Optimization for Data Augmentation (OCDS) in weakly supervised learning, as well as to conduct an in-depth analysis of the properties and performance of BF.
The main contributions include proving the consistency of BF and validating its performance through experiments. Additionally, the paper analyzes the relationship between the model and approximation uncertainty of BF and OCDS, revealing the advantages of BF over OCDS and its potential applications.

**Q2-3 Extent To Which Claims Are Supported By Evidence:**

3: Good: the main claims are supported by convincing evidence (in the form of adequate experimental evaluation, proofs, (pseudo-)code, references, assumptions).

**Q2-4 Reproducibility:**

3: Good: key resources (e.g. proofs, code, data) are available and key details (e.g. proofs, experimental setup) are sufficiently well-described for competent researchers to confidently reproduce the main results.

**Q3 Main Strengths:**

This paper introduces a novel comparison between the Bayesian Rule Integration method (BF) and OCDS in weakly supervised learning, offering unique insights into their performance and properties. It provides detailed experimental evaluations, rigorous mathematical proofs, and comparisons with the current state-of-the-art, ensuring the reliability of the research findings. Additionally, the paper includes information on the datasets and methods used in the experiments, facilitating reproducibility by other researchers.

**Q4 Main Weakness:**

There are still some aspects of this paper that need improvement. For example, in Figure 2 and Figure 3, the legends obscure the dashed lines in the figures, and the dashed lines extend beyond the borders. Additionally, I believe it would be helpful to provide a brief introduction below each figure and table, despite the detailed explanations provided in the text. Lastly, it would be beneficial to include the settings of some hyperparameters used in the experiments, facilitating better reproducibility of your results by future researchers.

**Q5 Detailed Comments To The Authors:**

The motivation of this paper is to explore the comparison between the Bayesian Rule Ensemble Method (BF) and Consistent Optimization for Data Augmentation (OCDS) in weakly supervised learning, as well as to conduct an in-depth analysis of the properties and performance of BF.
The main contributions include proving the consistency of BF and validating its performance through experiments. Additionally, the paper analyzes the relationship between the model and approximation uncertainty of BF and OCDS, revealing the advantages of BF over OCDS and its potential applications.
This paper introduces a novel comparison between the Bayesian Rule Integration method (BF) and OCDS in weakly supervised learning, offering unique insights into their performance and properties. It provides detailed experimental evaluations, rigorous mathematical proofs, and comparisons with the current state-of-the-art, ensuring the reliability of the research findings. Additionally, the paper includes information on the datasets and methods used in the experiments, facilitating reproducibility by other researchers.
The authors provided a comprehensive theoretical analysis, which makes the conclusion very convincing.
A minor issue is that the presentation of the paper could be further improved.

**Q9 Complying With Reviewing Instructions:**

Yes

---

> ### Author Rebuttal · Authors · 2024-04-03
>
> Thank you for your time and detailed comments. We appreciate your suggestions to improve the paper’s presentation.

---

### Official Review · Reviewer_kFQz · 2024-03-20

**Q2-1 Originality-Novelty:** 3
**Q2-2 Correctness-Technical Quality:** 3
**Q2-5 Clarity Of Writing:** 3

**Q1 Summary And Contributions:**

This paper investigates methods for estimating the (probabilities of) labels for data using rules-of-thumb and minimal label supervision. The authors analyze the properties of an adversarial approach under weak supervision, and compare it to the Dawid-Skene (DS) probabilistic methods. The research provides theoretical insights, statistical analyses, and experimental validations to shed light on the behavior of the adversarial approach under log-loss. Overall, this paper contributes to a new understanding of weak supervision methodologies.

**Q2-3 Extent To Which Claims Are Supported By Evidence:**

3: Good: the main claims are supported by convincing evidence (in the form of adequate experimental evaluation, proofs, (pseudo-)code, references, assumptions).

**Q2-4 Reproducibility:**

3: Good: key resources (e.g. proofs, code, data) are available and key details (e.g. proofs, experimental setup) are sufficiently well-described for competent researchers to confidently reproduce the main results.

**Q3 Main Strengths:**

1, The weak supervision problem is important and the motivation is clear. The adversarial method investigated in this paper is reasonable and plausible.

2, The paper demonstrates a strong theoretical foundation by providing detailed statistical analyses of the adversarial approach, offering valuable insights into its convergence properties.

3, The paper illuminates the strengths and weaknesses of each approach by comparing the adversarial method with probabilistic methods, such as the Dawid-Skene model.

4, The effectiveness of the analysis are evaluated experimentally by comparing the adversarial approach with other Weak Supervision methods.

**Q4 Main Weakness:**

It seems to be a good paper on ensemble learning, but I don't fully understand some details, especially the proofs (which is out of my expertise). So I cannot give any main weakness on this paper.

**Q5 Detailed Comments To The Authors:**

1, The authors need to clarify the underlying assumptions about the weak supervision sources clearer, as these might limit applicability in certain scenarios.

2, The authors should provide additional information about the practical challenges and considerations of implementing the adversarial weak supervision method. This information could improve understanding of the Theorems' application value.

**Q9 Complying With Reviewing Instructions:**

Yes

---

> ### Author Rebuttal · Authors · 2024-04-03
>
> Thank you for your time and comments.  In this reply, we'd like to add information and discuss the points you bring up in your comments.
>
> With regards to underlying assumptions about weak supervision sources, we mean to study the combination of weak supervision sources/rules of thumb without any assumptions on their underlying structure or interactions.  This does not pose a problem for the adversarial method because the worst case is assumed, but it can cause the independence assumptions for generative methods to be violated.  This does not stop these methods, e.g. Dawid-Skene, Data Programming, Firebolt [1], etc. from being applied in practice.  One approach is to try and learn/model the dependencies so the generative model is less misspecified (e.g. [2]).  This is not easy though [3].  Since the generative approach is the one mostly adopted in practice, e.g. Snorkel AI (https://snorkel.ai/), we study and compare the adversarial to generative/probabilistic approaches in this paper.
>
> With respect to the practical considerations for the adversarial methods presented, we see two main ones.  First is constructing the polytope of labelings $P$.  The consistency/rate of convergence results require that the polytope contain $\eta$, the true conditional label probabilities $Pr(y=\ell\mid x)$. In practice, we’d hope that $P$ contains $\eta$ with probability at least $1-\delta$. To get the inequalities (e.g. Equation 2) needed to construct the polytope, we use Wilson score intervals.  However, we have $p+k$ score intervals.  If each interval holds with probability at least $0.95$, the probability that they all hold simultaneously is basically always lower than $1-\delta = 0.95$ if one uses a union bound.  Since the quality of our prediction $g^{bf}$ depends on the size of the polytope, i.e. $\epsilon$ in Equation 3, requiring that each score interval hold with probability at least $1-\delta/(p+k)$ would make $\epsilon$ big, and likely make $g^{bf}$ suffer in quality.  To our knowledge adversarial methods which construct polytopes in similar ways suffer from this problem [4], [5]. Empirically, it doesn’t seem like having each score interval hold with probability at least 0.95 is a problem.  E.g. our experiments. Some theoretical work has been done on this issue.  See [4], the paragraph after Lemma 9.   Second is optimizing the dual objective in Theorem 2. That maximization problem can be optimized via SGD where the number of variables is $p+k$, dependent only on the rule and class count only.  This would put it in line with a method like Data Programming which has been deployed in industry (c.f. Snorkel AI) and likely uses some form of gradient descent.  The Data Programming implementation we used does a form of gradient descent.
>
> [1] https://proceedings.mlr.press/v151/kuang22a/kuang22a.pdf
>
> [2] https://proceedings.mlr.press/v97/varma19a/varma19a.pdf
>
> [3] https://arxiv.org/pdf/2106.10302.pdf
>
> [4] https://proceedings.mlr.press/v139/mazzetto21a/mazzetto21a.pdf
>
> [5] https://jmlr.org/papers/volume24/22-0339/22-0339.pdf

---

### Official Review · Reviewer_3pCP · 2024-03-22

**Q2-1 Originality-Novelty:** 2
**Q2-2 Correctness-Technical Quality:** 3
**Q2-5 Clarity Of Writing:** 3

**Q1 Summary And Contributions:**

Weak supervision methods use labeling functions/rules-of-thumb to label training data. Such methods can be split into two categories: the Dawid-Skene (DS) model, which assumes a probabilistic graphical model that is then fit based on the observable data, and the Balsubramani-Freund (BF) model which is an adversarial model that uses estimates of the rule accuracies and class balances as constraints for an optimization problem to produce the most feasible set of true labels. This paper studies an adversarial model with a log loss objective and derives its optimal solution. DS and BF models are compared using an error decomposition of model uncertainty + approximation uncertainty (as KL divergences), and the consistency of these methods are compared. Empirical results show that log-loss BF attains lowest log loss and best accuracy on most standard WS benchmarks, and error decomposition is measured.

**Q2-3 Extent To Which Claims Are Supported By Evidence:**

3: Good: the main claims are supported by convincing evidence (in the form of adequate experimental evaluation, proofs, (pseudo-)code, references, assumptions).

**Q2-4 Reproducibility:**

3: Good: key resources (e.g. proofs, code, data) are available and key details (e.g. proofs, experimental setup) are sufficiently well-described for competent researchers to confidently reproduce the main results.

**Q3 Main Strengths:**

* Empirical results show that log-loss BF outperforms DS-style WS methods on a variety of datasets, in terms of log-loss, 0-1 loss and Brier score
* log-loss BF is theoretically grounded with measurable error decomposition analysis provided, and the optimal solution is simple to compute
* This paper is practically relevant; while it is difficult for practitioners to label an entire dataset, it is simple to label part of the dataset, with which they can predict a rule's accuracy. With an estimate of the rule's accuracy available, this paper suggests that the class of adversarial WS methods should be considered more closely and has merits over DS methods.

**Q4 Main Weakness:**

* The argument about the inconsistency of DS methods is limited to the EM algorithm, which is known to have poor convergence guarantees. I was expecting a more general argument for comparing BF and DS consistency.
* Many weak supervision methods of the DS type do not explicitly use labeled data to compute accuracies and class balances. For instance, MV is truly label free and DP/Snorkel sometimes uses labeled data to estimate class balance, but otherwise defaults to a uniform prior or estimates class balance via tensor decomposition (see https://arxiv.org/abs/1810.02840). Therefore, it is unclear to me if BF outperforms these methods purely due to utilization of labeled data. One way to test this could be to reduce the amount of data used to estimate accuracies and study what happens as $\epsilon$ gets larger. Another option is to integrate labeled data into WS baselines more explicitly by taking the average of the accuracy estimated via a WS method and the accuracy estimated using labeled data (such as in https://arxiv.org/abs/2103.02761)

**Q5 Detailed Comments To The Authors:**

Major:
* Can you comment on how the WS baselines considered utilize labeled data? What happens when you use even less labeled data or when you explicitly incorporate labeled data into the WS estimates?

Minor:
* Are the results regarding the BF solution related to an application of the principle of maximum entropy?
* Can you provide more information about AMCL CC? Why is there sometimes a big performance gap between this and BF if they are both adversarial?
* Clarity: it was unclear how $\epsilon$ is computed until when section 8 mentions using the Wilson score interval. Since we only can estimate $b_j$ and do not know the true $b_j^\star$, $\epsilon$ needs to be approximated, which was not clear in section 4 and 5.
* Are there theoretical guarantees and similar analysis done for the 0-1 loss BF approach?
* How is $a_il$ defined in Theorem 2?
* Is $g^{log}$ equal to $g^{(\theta^{log})}$?

**Q9 Complying With Reviewing Instructions:**

Yes

---

> ### Author Rebuttal · Authors · 2024-04-04
>
> Thank you for your time and your thorough comments on our work. We first respond to the minor comments, then the major comment.  Finally, we discuss what it would take to give a more general argument for OCDS’ inconsistency.
>
> For the minor comments, in the order they are stated,
>
> — Yes, one can interpret it that way. Related is [1] for supervised learning.
>
> — AMCL CC’s objective and rule accuracy constraints are in terms of Brier score (compared to log loss and 0-1 loss for BF).  In Table 6 of the appendix, BF and AMCL CC’s Brier scores are quite similar.
>
> — Our theoretical results are in terms of estimates for rules accuracies/class frequencies $b$ and the interval width $\epsilon$. One may choose a different method to estimate those quantities, e.g. $\epsilon$ by Chernoff bounds. An interesting problem is determining how to generate unsupervised class frequency/rule accuracy estimates suitable for BF.  [2] could be a starting point.
>
> — For the transductive setting, we are not aware of consistency/rates of convergence for BF/related adversarial methods with 0-1 loss.  A difficulty we foresee is the non-differentiability of the loss function.
>
> — $a_{i\ell}^{\theta}$ is meant to be element $k(i-1) + \ell$ of $A^{\top} \theta \in \mathbb{R}^{kn}$.  $a_{i\ell}^{\theta}$ is the score from WMV for datapoint $x_i$, class $\ell$.  $g_{i\ell}$ is the resulting probability after taking the softmax of  $a_{i\ell}^{\theta}$, varying $\ell \in [k]$.
>
> — Yes, $g^{log} = g^{(\theta^{log})}$, though we meant to change the notation in Figures 2,3 to say $g^{bf}$ rather than $g^{log}$.
>
> ---
>
> To preface our response to the major comment, we want to give our point of view for the experiments comparing BF to other WS methods.  We believe that showing BF (with labeled data) is comparable to unsupervised WS methods is a necessary step in arguing that adversarial methods in WS deserve to be studied.  I.e. it would be bad if BF with labeled data was worse than unsupervised WS methods.
>
> When even less data is used to estimate the class frequencies/rule accuracies for BF and AMCL CC, the quality of their respective predictions drop.  For BF, when the adversary chooses from $\Delta_k^n$, the learner’s loss can be arbitrarily large. If one is in the setting of Theorem 6, the adversary must choose $g^{*}$.  Having labeled data puts one between the two extremes.  In keeping with the discussion on model and approximation uncertainty, BF’s error on half the datasets (Cancer, Domain, IMDB, OBS, Yelp) is dominated by the model uncertainty, something unaffected by labeled data.  See Appendix pp 51-55.  We note that on Basketball, IMDB, SMS, Yelp, Youtube, the average number of labeled points out of 100 used to estimate a rule’s accuracy was approximately $77, 32, 3, 19, 17$ respectively.
>
> While using labeled data can improve the unsupervised WS baselines we see, it is not clear how much improvement there will be.  Concretely, one can use labeled data with OCDS by estimating the class frequencies/rule accuracies and taking one E step. $g^{ds*}$ can serve as a proxy should those estimates be close to the empirical rule accuracies/class frequencies $b^*, w^*$. We see that $d(\eta, g^{ds*})$ is often bigger than OCDS’ model uncertainty $d(\eta, g^{\*})$, sometimes by a lot.  It’s because of OCDS’ parameterization of weights (Lemma 31, Appendix) that labeled data will not be as helpful as with BF.  For AwA, Basketball, OBS, $g^{ds*}$ is worse or comparable to $g^{bf}$. For Cancer, Cardio, IMDB, SMS, $g^{ds*}$ is similar in quality to $g^{\*}$, meaning labeled data could bring a large improvement. For Domain, Yelp, Youtube, $d(\eta, g^{ds*})$ is about $d(\eta, g^{*})$ plus half of BF’s approximation uncertainty, i.e. there could be improvement.
>
> ---
>
> Finally, we want to discuss the subtleties of giving a more general argument for DS’ inconsistency. A more general argument comparing the consistency of BF and DS, or specifically one coin Dawid-Skene (OCDS), would entail the analysis of more sophisticated algorithms used estimate the underlying rule accuracies $Pr(h^{(j)}(x) = y)$ and the underlying class frequencies $Pr(y=\ell)$.  This is because using the conditional independence assumption for OCDS does not completely determine inconsistency.  This independence assumption determines both the set of predictions that OCDS can make (Lemma 7, essentially the same set as BF) but also the form of the weights. The OCDS weight for rule $h^{(j)}$ is $\log (e^{n_j} b_j(k-1)/(1-b_j))$ for some $b_j \in (0,1)$ and $\log(e^{n}w_{\ell})$ for class $\ell$. (Lemma 31, Appendix).  However, we need to choose an algorithm like EM to estimate the rule accuracies $b_j$ and class frequencies $w_\ell$. Only with the choice of an algorithm can we discuss consistency.  So, we believe that evaluating OCDS' inconsistency must be done case by case. E.g. EM or [2]'s algorithm.
>
> [1] https://arxiv.org/pdf/2007.05447.pdf
>
> [2] https://jmlr.org/papers/volume17/14-511/14-511.pdf

---

### Official Review · Reviewer_wWwt · 2024-03-26

**Q2-1 Originality-Novelty:** 3
**Q2-2 Correctness-Technical Quality:** 3
**Q2-5 Clarity Of Writing:** 3

**Q1 Summary And Contributions:**

Summary:
This paper discusses an important subfield of machine learning, weakly supervised learning.
By utilizing the weak supervision of the given data, two different approaches Dawid-Skene model (the probabilistic approach) and Balsubramani-Freund model (the adversarial approach) are discussed. The authors provide a variety of statistical results for the adversarial approach under log-loss. Also, they prove that the probabilistic approaches for the same model class can fail to be consistent.

Contributions:
1.	The authors provide a variety of statistical results (e.g., relation with logistic regression, consistency and convergence rate) for the adversarial approach under log-loss;
2.	This paper provides an analysis of the difference between the probabilistic approach and the adversarial approach, which is helpful to better understand this problem.

**Q2-3 Extent To Which Claims Are Supported By Evidence:**

2: Fair: the main claims are somewhat supported by evidence (but the experimental evaluation may be weak, or does not match entirely with the claims, important baselines may be missing, proofs contain important ideas but lack rigor, algorithmic details are only discussed superficially, references are imprecise, assumptions are not sufficiently motivated or explicated, etc.).

**Q2-4 Reproducibility:**

3: Good: key resources (e.g. proofs, code, data) are available and key details (e.g. proofs, experimental setup) are sufficiently well-described for competent researchers to confidently reproduce the main results.

**Q3 Main Strengths:**

1.	 This paper is well-organized and mostly clearly written.
2.	The discussion between two kinds of approaches is clear and easy to understand.
3.	This paper is technically sound, with a variety of statistical results (e.g., relation with logistic regression, consistency, and convergence rate) for the adversarial approach under log-loss.

**Q4 Main Weakness:**

The only concern I have is that the experiment result is not solid enough. All experiments are based on a simple dataset, while a large number of methods in the weakly supervised learning field are not compared. The performance of the proposed method remains unclear in the realistic tasks.

**Q5 Detailed Comments To The Authors:**

Although the main contribution of this paper lies in the theoretical aspects, I think adding some more realistic datasets may improve this work from a practical perspective.

Practical applications of these findings to real-world scenarios would greatly enhance the paper's impact, offering readers insight into how these theoretical advantages play out in practice.

**Q9 Complying With Reviewing Instructions:**

Yes

---

> ### Author Rebuttal · Authors · 2024-04-03
>
> Thank you for your time and the practical perspective you bring in your comments.
>
> We would like to expand a bit on the experiments conducted in the main paper.  While our experiment for consistency was from a simple synthetic dataset, the results presented in Table 1 are from a comparison on ten real world datasets.  These datasets have diverse domains and involve ensemble sizes $3\leq p \leq 73$, datapoints counts $171\leq n \leq 30400$. We defer to the appendix section Q.1 for a more detailed exposition about the dataset domains/what the rules of thumb are.

---

### Official Review · Reviewer_WVfz · 2024-03-31

**Q2-1 Originality-Novelty:** 3
**Q2-2 Correctness-Technical Quality:** 3
**Q2-5 Clarity Of Writing:** 2

**Q1 Summary And Contributions:**

This paper examines the aggregation of multiple labeling functions by incorporating confidence regions based on their predictive accuracy and label proportion. The primary contribution is an analysis of a "log loss" variant of the Balsubramani-Freund (BF) methods. The authors demonstrate the consistency of the BF methods' solution and establish the convergence rate. Additionally, theoretical analysis and experiments are conducted to compare the BF methods with the Dawid-Skene methods.

**Q2-3 Extent To Which Claims Are Supported By Evidence:**

3: Good: the main claims are supported by convincing evidence (in the form of adequate experimental evaluation, proofs, (pseudo-)code, references, assumptions).

**Q2-4 Reproducibility:**

3: Good: key resources (e.g. proofs, code, data) are available and key details (e.g. proofs, experimental setup) are sufficiently well-described for competent researchers to confidently reproduce the main results.

**Q3 Main Strengths:**

Strengths:

+ This paper provides a thorough analysis of the adversarial approach (Balsubramani-Freund method), including the form of the solution, its consistency, and the convergence rate.
+ The paper also theoretically and empirically compares the Balsubramani-Freund method with the Dawid-Skene method, highlighting the advantages of the BF method.

**Q4 Main Weakness:**

Regarding the results, I do not find major weaknesses in this paper, as it offers a comprehensive analysis of the BF methods (though my perspective might be influenced by my limited familiarity with the literature). I believe the readability of this paper could be further enhanced, as discussed in Q5.

**Q5 Detailed Comments To The Authors:**

- The purpose of Section 5.4 remains somewhat unclear to me. While it is interesting to see that the max-min problem can be transformed into a logistic regression problem, the benefits or implications of such a conversion are not clearly explained.

- Additionally, it would be beneficial to discuss the computational complexity of solving Equation (4). Specifically, it is important to discuss whether the optimization procedure is more time-consuming than the Dawid-Skene estimator.

**Q9 Complying With Reviewing Instructions:**

Yes

---

> ### Author Rebuttal · Authors · 2024-04-03
>
> Thank you for your time and your comments.
>
> The goal of Section 5.4 was to provide a backstop in terms of arguing that BF is not too “pessimistic”.  In showing the transformation of BF to logistic regression, we establish the following conditional: If one claims BF is too pessimistic in practice, then they also claim that logistic regression is too pessimistic in practice. So, if one does not accept the claim that logistic regression is too pessimistic, they must reject the claim that BF is too pessimistic.  I.e. modus tollens. (By pessimistic, we mean too preoccupied with the worst case scenario.)
>
> We agree that computational complexity is an important aspect.  One may use SGD to optimize the dual problem presented in Theorem 2.  The maximization objective is concave, and the number of variables to optimize over is small, $p+k$, the number of rules plus number of classes.  Note that it does not scale with the number of datapoints $n$. This would bring the method in line with the Data Programming/Snorkel implementation which also uses gradient descent.  We note that for our experiments the total time needed to construct and solve the convex optimization problem (Theorem 2) was often less than a second and at worst less than 3 seconds.  (There were $3\leq p \leq 73$ rules of thumb and $171 \leq n\leq 30400$ datapoints in the 10 real datasets we considered.)  This was comparable/faster to running EM to convergence for Dawid-Skene.

---

### Meta-Review · Area_Chair_bNUT · 2024-04-16

This paper addresses the problem of label aggregation in the weakly supervised learning setting. In contrast to the popular line of solutions based on probabilistic modeling (e..g, under the Dawid-Skene model), this paper anlayzed an adversarial approach based the Balsubramani-Freund  methods. The core idea of this approach is via the estimates of the labeling function’s accuracies and class balances as constraints for an optimization problem to produce the most feasible set of true labels.
The authors anlaiyzed the consistency and convergence of this adversarial approach and showed its advantage over the probabiblistic methods.

The paper presents a promising alternative to improving label quality for the weakly supervised learning setting. The focus of log-loss based BF solution is theoretically sound and well explained. All reviewers agreed on its technical contributions.